# Face detection in untrained deep neural networks

Seungdae Baek [1,4], Min Song [2,4], Jaeson Jang [1], Gwangsu Kim [3] & Se-Bum Paik [1,2✉]

Face-selective neurons are observed in the primate visual pathway and are considered as the basis of face detection in the brain. However, it has been debated as to whether this neuronal selectivity can arise innately or whether it requires training from visual experience. Here, using a hierarchical deep neural network model of the ventral visual stream, we suggest a mechanism in which face-selectivity arises in the complete absence of training. We found that units selective to faces emerge robustly in randomly initialized networks and that these units reproduce many characteristics observed in monkeys. This innate selectivity also enables the untrained network to perform face-detection tasks. Intriguingly, we observed that units selective to various non-face objects can also arise innately in untrained networks. Our results imply that the random feedforward connections in early, untrained deep neural networks may be sufficient for initializing primitive visual selectivity.

[1] Department of Bio and Brain Engineering, Korea Advanced Institute of Science and Technology, Daejeon 34141, Republic of Korea. [2] Program of Brain and Cognitive Engineering, Korea Advanced Institute of Science and Technology, Daejeon 34141, Republic of Korea. [3] Department of Physics, Korea Advanced Institute of Science and Technology, Daejeon 34141, Republic of Korea. [4] These authors contributed equally: Seungdae Baek, Min Song. ✉email: sbpaik@kaist.ac.kr

The ability to identify and recognize faces is a crucial function for social behavior, and this ability is thought to originate from neuronal tuning at the single or multi-neuronal level[1–20]. Neurons that selectively respond to faces (face-selective neurons) are observed in various species[21–23], and they have been considered as the building blocks of face detection[18]. The observation of this type of intriguing neuronal tuning in the brain has inspired neuroscientists, raising important questions about its developmental mechanism—whether face-selective neurons can arise innately in the brain or require visual experience, and whether neuronal tuning to faces is a special type of function distinctive from tunings to other visual objects.

Regarding the emergence of neuronal face-selectivity, previous studies have suggested a scenario in which visual experience develops face-selective neurons[24–26]. The experience-dependent characteristics of face-selective neurons imply that visual experience plays a critical role in developing face-selectivity in the brain. It was observed that the preferred feature images of face-selective neurons in adult monkeys are those that resemble animals or familiar people depending on individual experiences[25]. Another study of the inferior temporal cortex (IT) in monkeys reported that robust tuning of face-selective neurons is not observed until 1 year after birth[6] and that face-selectivity relies on experience during the early infant years. It was also reported that monkeys raised without face exposure did not develop normal face-selective domains[26]. However, another view suggests that face-selectivity can innately arise without visual experience[27–34]. Although visual experience is critical for refining the development of face-selective neurons, several lines of research have demonstrated that primitive face-selectivity is observed even before visual experience[27–31]. Primate infants behaviorally prefer to look at face-like objects as opposed to non-face objects[32–34], implying that face-encoding units may already exist in infants. Moreover, category-selective domains, including those for faces, are observed in the ventral stream of adult humans who have been blind since birth[28,29]. Furthermore, a recent study reported that face-selective neurons are observed in infant animals and that the spatial organization of such early face-selective regions appeared similar to that observed in adults[6]. These results taken together imply that face-selective neurons can arise before visual experience, in contradiction to the first scenario.

There has been another important debate as to whether face-selectivity is a special type of visual function distinguished from other processes of object recognition, the developmental mechanism of which needs to be considered and examined distinctively from other visual neural tunings. After early observations of the face-selective responses of single neurons in the IT, face detection has been considered one of the most important visual functions necessary for the survival of social animals[35–38]. Observation of the fusiform face area (FFA), which is specialized for face recognition, also reinforced the idea that face-selectivity is a specialized neuronal tuning, which may develop differentially from cognition of other general visual objects[12,39–42]. However, more recent studies have reported that selectivity to objects such as a car or a bird can also develop in the FFA from visual experience[43,44], implying that faces may not be a special, distinct type of object class for visual function and that neuronal tuning to various visual objects can also arise similarly to selectivity to faces.

The argument concerning these issues, which reveals our incomplete understanding of face-selectivity, likely stems from limitations regarding the control of the experimental conditions, as it is impossible to control the amount of visual experience for a particular category, such as the face, in individual subjects. Even if the subjects are visually deprived such that they are prevented from having a visual experience, the portion of category-selective neurons and their degree of tuning may vary across subjects and cannot easily be predicted. These various factors make it difficult to investigate the developmental mechanism of face- and other object-selective neurons in the brain.

A model study using biologically inspired artificial neural networks, such as deep neural networks (DNNs)[45,46], may offer an effective approach to the problem in this case[47–50]. Recently, DNNs, a stack of biologically inspired feedforward projections with a linear–nonlinear neural motif, have provided insight into the underlying mechanisms of brain functions, particularly with regard to the development of various functions for visual perception[47,48,51]. For example, a recent model study reported that the neural response of the monkey IT cortex could not only be predicted by the responses of DNNs trained to natural images[47,48] but could also be controlled by the preferred feature image generated by the DNN model[52]. Notably, previous studies using random hierarchical networks provide important clues about the origin of innate face-selectivity in untrained neural networks. It was reported that untrained feedforward networks can initiate various cognitive functions with random weights and that a random network can perform image classification tasks in that way as well[53–56]. It was also reported that a randomly initialized convolutional neural network could reconstruct corrupted images without any training, which implies that a random network can provide a priori information about the low-level statistics in natural images[57]. Overall, such observations suggest the possibility of the emergence of innate cognitive functions, such as primitive face-selectivity in untrained, random hierarchical networks. However, the details of how this innate function emerges in untrained neural networks are not yet understood.

Herein, we show that face-selective units (model neurons) can arise in completely untrained hierarchical neural networks. Using AlexNet[45], a model that captures properties of the ventral stream of the visual cortex, we found that face-selectivity can emerge robustly across different conditions of randomly initialized DNNs. We found that their face-selectivity indices (FSI) are comparable to those observed with face-selective neurons in the brain. The preferred feature images obtained from the reverse-correlation (RC) method and the generative adversarial network show that face-selective units are selective for a face-like configuration, distinct from units with no selectivity. Furthermore, we found that face-selective units enable the network to perform face detection. Intriguingly, we found that units selective to various non-face objects can also arise innately in untrained neural networks, implying that face-selectivity may not be a special type of visual tuning and that selectivity to various objects classes can arise innately in untrained DNNs, spontaneously from random feedforward wirings. Overall, our results imply a possible scenario in which the random feedforward connections that develop in early, untrained networks may be sufficient for initializing primitive face-selectivity as well as selectivity to other visual objects in general.

## Results

**The emergence of face-selectivity in untrained DNNs.** To simulate the emergence of face-selective neurons (Fig. 1a), we measured the responses of a biologically inspired DNN model, AlexNet[45] (Fig. 1b), to a similarity-controlled face stimulus set (Fig. 1c). A standard AlexNet model is composed of five convolutional layers (feature extraction network) and three fully connected layers (classification network), which together reproduce the structure of the ventral stream of the visual pathway (Supplementary Table 1). To investigate the selective responses of individual units rather than the performance of a trained system, we discarded the classification layers and examined activity in the

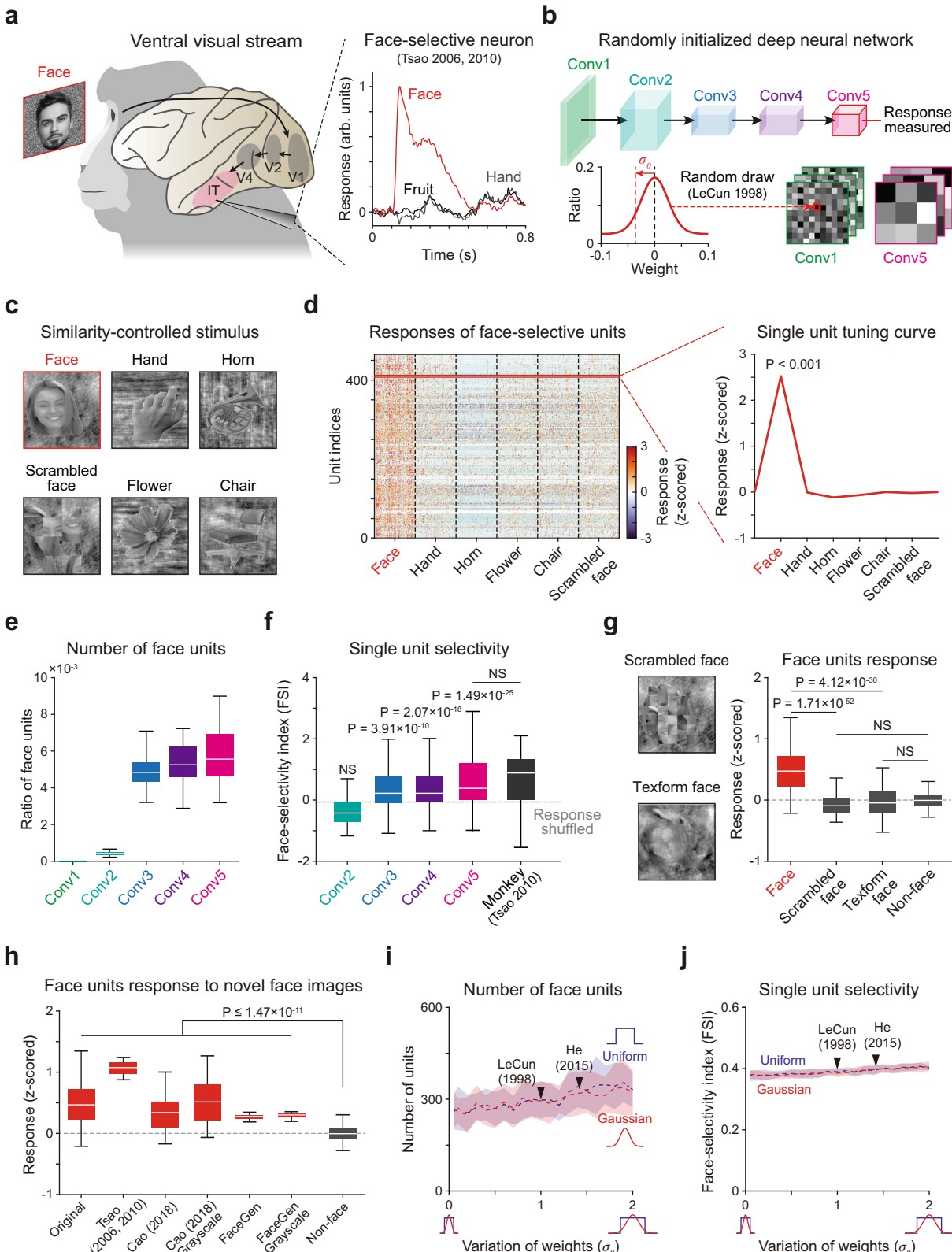

**Fig. 1**

final layer (Fig. 1b, top, Conv5) of the feature extraction network. To examine whether face tuning of units can arise even in completely untrained DNNs, we devised an untrained AlexNet by randomly initializing the weights of filters in each convolutional layer (Fig. 1b, bottom). For this, we used a standardized network initialization method[58], by which the weights of kernels in each convolutional layer were randomly drawn from a Gaussian

distribution with parameters set to control the strength of the input signals across the layers. The stimulus set consisted of grayscale images in six different categories (Fig. 1c), specifically the face, a scrambled face, and four non-face objects, as previously done in monkey experiments[59]. The images in each class were designed to control the low-level features of the luminance, contrast, object size, and object location, and they have

**Fig. 1 Spontaneous emergence of face-selectivity in untrained networks. a** Face-selective neurons and their response observed in monkey experiments. The response was normalized to the maximum value as 1. The face image shown is not the original stimulus set due to copyright. The image shown is available at [https://www.shutterstock.com] (see Methods for details). **b** The architecture of the untrained AlexNet[45]. The untrained AlexNet was devised using a random initialization method[58], for which the values in each weight kernel were randomly sampled from a Gaussian distribution. **c** A stimulus set was designed to control the degree of intra-class image similarity. Stimulus images were selected and modified from a publicly available dataset that has been used in human fMRI study[59]. The original images are available at [http://vpnl.stanford.edu/fLoc/]. **d** Responses of individual face-selective units in the untrained AlexNet ($P < 0.001$, two-sided rank-sum test, uncorrected). **e** The number of face-selective units in each convolutional layer in untrained networks ($n = 100$). **f** Face-selectivity index (FSI) of face-selective neurons in the primate IT[7] ($n = 158$), and face units in each convolutional layer in the untrained AlexNet. The control FSI was measured according to the shuffled responses of face-selective units in the untrained network. **g** (Left) Examples of texform and scrambled face images. (Right) Responses of face-selective units to the original face ($n = 200$), the scrambled face ($n = 200$) and texform face images ($n = 100$). **h** Responses of face-selective units to four different sets of novel face images: (1) 50 face images from our original dataset (images not used for finding face-selective units), (2) 16 images used in Tsao et al.[5,7], (3) 50 images used in Cao et al.[62] in color and gray scale, and (4) 50 face images artificially generated by the FaceGen simulator (singular inversions) in color and gray scale. **i** The number of face-selective units, where the weight variation was changed from 5 to 200% of the original value using two different initialization methods with a Gaussian (red) and a uniform distribution (blue). **j** FSI of face-selective units across changes in the weight. Dashed lines indicate the mean and shaded areas indicate the standard deviation of 30 random networks. All box plots indicate the inter-quartile range (IQR between Q1 and Q3) of the dataset, the horizontal line depicts the median and the whiskers correspond to the rest of the distribution (Q1 − 1.5*IQR, Q3 + 1.5*IQR).

statistically comparable intra- and inter-class image levels of similarity (Supplementary Fig. 1).

Surprisingly, we observed a group of face-selective units ($n = 250 \pm 63$ in 100 random networks, mean ± s.d.) that show significantly higher responses to face images than to non-face images emerging in the untrained networks (Fig. 1d, $P < 0.001$, two-sided rank-sum test). Here, a unit is defined as a unit component at each position of the channel in an activation map of the network. For example, there are 43,264 units ($= 13 \times 13 \times 256$, $N_{x\text{-position}} \times N_{y\text{-position}} \times N_{channel}$) in Conv5. We considered each unit (of the same filter) at different spatial locations as different ones, as the selectivity of units at different locations appears to be distinct despite the fact that they share the same filter (Supplementary Fig. 2a–f). We also investigated the layer-specific emergence of face-selective units in untrained networks. We found that face-selective units are also observed in earlier layers, Conv3 to 5 but are scarcely found in Conv1 and 2 (Fig. 1e, Conv1: $n = 0.008 \pm 0.002\%$, Conv2: $n = 0.047 \pm 0.009\%$, Conv3: $n = 0.491 \pm 0.089\%$, Conv4: $n = 0.534 \pm 0.103\%$, Conv5: $n = 0.579 \pm 0.146\%$). We found that the number of face-selective units and the face-selectivity index (FSI) of each unit increased through the layer hierarchy (Fig. 1e, f). Notably, the number of face-selective units did not show significant differences across the convolutional group or filters within each layer (Supplementary Fig. 2g, h). This suggests that face-selective units are not dominantly generated by a particular filter. The number of observed face-selective units was highest in the mid- and high-level layers, similar to observations in the ventral visual pathway of monkeys[5,6]. These results suggest that the development of face-selectivity requires a hierarchical structure of the network along with random feedforward weights, which enables multiple linear-nonlinear computations.

We also found that the observed face-selective units in the untrained networks (Conv5) show a value of the averaged FSI[5,7] comparable to the index associated with monkey IT neurons[7] (Fig. 1f, $n_{untrained} = 465$, $n_{monkey} = 158$, NS, two-sided rank-sum test, $P = 7.69 \times 10^{-2}$, $r_{rbc} = 9.25 \times 10^{-2}$, two-sided Kolmogorov–Smirnov test, $P = 2.49 \times 10^{-4}$, $d = 2.32 \times 10^{-2}$) and a significantly higher value than those measured from a shuffled response (Fig. 1f, $n_{untrained} = 465$, $n_{shuffled} = 465$, two-sided rank-sum test, $P = 1.49 \times 10^{-25}$, $r_{rbc} = 5.09 \times 10^{-1}$) for various definitions of the FSI[5,17,60] (Supplementary Fig. 3). These results suggest that face-selective units, highly tuned to the face as observed in the brain, can emerge in DNNs even in the complete absence of learning.

One possible scenario for the emergence of such face-selectivity in random networks is that the observed face-selective units are simply sensitive for local face parts common to facial images. To investigate this possibility, we measured the responses of the face-selective units to a local feature of the face using two types of control images in which global face features are disrupted but local face features are preserved. These were (1) scrambled faces, in which small patches of the local face components were spatially scrambled, and (2) texform faces[61], in which global face features are disrupted but the statistics of the local face texture is preserved (Fig. 1g, left). We confirmed that face-selective units show significantly higher responses to the original face images compared to the corresponding control images (Fig. 1g, right; Face vs. Scrambled face, $n = 200$, one-sided rank-sum test, $P = 1.71 \times 10^{-52}$, $r_{rbc} = 7.69 \times 10^{-1}$; Face vs. Texform face, $n = 100$, one-sided rank-sum test, $P = 4.12 \times 10^{-30}$, $r_{rbc} = 6.56 \times 10^{-1}$). In addition, the responses of face units to these control images were not greater than those to other non-face images, implying that face-selective units are selective to the global context of faces instead of the local components (Fig. 1g, right; Scrambled face vs. Non-face, $n = 200$, one-sided rank-sum test, NS, $P = 1.00$, $r_{rbc} = -2.42 \times 10^{-1}$, one-sided Kolmogorov–Smirnov test, $P = 8.31 \times 10^{-1}$, $d = 3.00 \times 10^{-3}$; Texform face vs. Non-face, $n = 100$, one-sided rank-sum test, NS, $P = 9.40 \times 10^{-1}$, $r_{rbc} = -9.00 \times 10^{-2}$, one-sided Kolmogorov–Smirnov test, $P = 4.51 \times 10^{-2}$, $d = 1.84 \times 10^{-2}$). These results suggest that the observed face units in the untrained network are not particularly selective to local face parts, but are instead selective to a whole face.

Next, we investigated the responses of face-selective units to four different novel stimulus sets that were not used to find face-selective units. These were (1) 50 face images from our original data set, not used for finding the face-selective units; (2) 16 face images used in Tsao et al. (2006, 2010)[5,7]; (3) 50 face images used in Cao et al.[62]; and (4) 50 face images artificially generated by the FaceGen simulator (singular inversions) in color and grayscale (Fig. 1h). We found that face-selective units in the untrained network show significantly higher responses to novel face images compared to the responses to non-face images under all conditions (Fig. 1h, Novel face vs. Non-face, one-sided rank-sum test, $P \leq 1.47 \times 10^{-11}$, $r_{rbc} \geq 4.54 \times 10^{-1}$). These results suggest that the observed face-selectivity in an untrained network defined by one specific dataset can be generalized to other novel sets of faces.

To confirm that the emergence of face-selective units is not due to the specific initial parameter set but is rather generally observed in an untrained network, we varied the width of the weight distribution for random network initialization (Gaussian and uniform) from 5 to 200% of the original standard deviation of the standardized random initialization[58] and examined if face-selective units consistently emerge (Fig. 1i, j). We found that face-selective units consistently arise in the untrained networks across the variation of the parameter. The number (Fig. 1i) and selectivity index (Fig. 1j) of observed face units were largely unchanged across a wide range of weight variations and across the wide variation in the width of the weight distribution. This implies that the emergence of face-selective units in untrained networks is highly robust to variations of the wiring strength.

**Preferred feature images of face-selective units in an untrained network**. Next, to characterize the feature-selective responses of these face-selective units qualitatively, we reconstructed preferred feature images (PFI) of individual units using a reverse-correlation (RC) method[63] and a generative adversarial network algorithm (X-Dream)[25] (see Methods). In the RC analysis, we presented 2500 images of bright and dark 2D Gaussian filters at random positions as input stimuli to an untrained network (Fig. 2a). By adding stimuli weighted according to the corresponding neural response with 100 repeated iterations, we obtained the preferred feature images of the target units. In the X-Dream analysis, a deep generative adversarial neural network[64] was trained to ImageNet datasets[65] to synthesize preferred feature images from the image codes scored by a genetic algorithm using the responses of units from repeated iterations (Fig. 2b). We found that the response of target units induced by the PFI was increased, being higher than those induced by face stimulus images, as the iteration number of the genetic algorithm exceeds a certain value (Fig. 2a, right, $n = 465$, RC PFI vs. face stimulus, two-sided rank-sum test, $P = 1.51 \times 10^{-6}$, $r_{rbc} = 1.58 \times 10^{-1}$; RC PFI vs. non-face stimulus, two-sided rank-sum test, $P = 3.12 \times 10^{-2}$, $r_{rbc} = 7.07 \times 10^{-2}$, Fig. 2b, right, $n = 465$, X-Dream PFI vs. face stimulus, two-sided rank-sum test, $P = 1.92 \times 10^{-119}$, $r_{rbc} = 7.63 \times 10^{-1}$; X-Dream PFI vs. non-face stimulus, two-sided rank-sum test, $P = 2.68 \times 10^{-129}$, $r_{rbc} = 7.94 \times 10^{-1}$). These results indicate that our PFI generation methods successfully find the most preferred input feature of face-selective units. As a result, using both methods, we obtained the PFI of face-selective units, units selective to non-face classes, and units without selective responses to any image classes (Fig. 2c, Supplementary Fig. 4). We found that the PFI of the face-selective units presents distinguishable features from those of other objects. We observed that the PFIs of face-selective units represent face-like configurations, whereas the PFIs of units selective to non-face classes show noticeable configurations of each object class (i.e., flowers in an RC and X-Dream).

To quantify the structural similarity of the PFIs to face images, we defined the face-configuration index as the averaged pixel-wise correlation between a PFI and the 200 face images used for face unit selection (Fig. 2d). We estimated the face-configuration index of each PFI of face-selective and non-face-selective units generated by the RC method. We found that the estimated PFIs of face-selective units (with a visually observable face-like configuration) have a significantly higher average value of the index than that from the units selective to non-face objects (Fig. 2e, Face PFI vs. Non-face PFI, $n_{Face} = 465$, $n_{Hand} = 7$, $n_{Horn} = 772$, $n_{Flower} = 107$, $n_{Chair} = 63$, one-sided rank-sum test, $P \leq 5.63 \times 10^{-4}$, $r_{rbc} \geq 1.56 \times 10^{-1}$; f, Face PFI vs. Non-face PFI, $n_{Face} = 465$, $n_{Hand} = 7$, $n_{Horn} = 772$, $n_{Flower} = 107$, $n_{Chair} = 63$,

one-sided rank-sum test, $P \leq 1.93 \times 10^{-5}$, $r_{rbc} \geq 2.35 \times 10^{-1}$). Notably, the average pairwise correlation estimated between each face stimulus image shows a significantly lower value of nearly zero, implying that the observed index of face-selective units reflects structural similarity to the averaged (or a prototype, abstract) face image rather than similarly to particular face images accidentally observable.

Next, we hypothesized that the observed face-selective units may encode invariant representations of the prototype face images, as some types of intrinsic invariance are a basic property of CNNs. A number of previous studies suggested that various types of invariance (e.g., translation, scaling, and rotation) over a wide range of image transformations can be implemented in a CNN[66–70], mostly due to three key components—the convolutional layer, the pooling layer, and the hierarchical structure—in CNN models. Thus, to investigate whether the observed face-selective units show invariant representations of face images regardless of the corresponding image condition, we measured the responses of face-selective units to face and non-face object images with various positions, sizes, and rotation angles. First, we observed that single face units show constant face tuning under a fairly wide range of size/position/rotation variations and that range was comparable with those of face-selective neurons in IT[71] (Supplementary Figs. 5 and 6). Notably, we found that our model units also show the inversion effect observed in monkeys[5,27,72]. We found that the responses of face-selective units to inverted face images are significantly lower than those to upright faces (Supplementary Fig. 7).

Similarly, we found viewpoint-invariant face-selective units (Supplementary Figs. 8 and 9) and mirror-symmetric viewpoint-specific units (Supplementary Fig. 10) in a random network. Interestingly, the number of viewpoint-invariant face-selective units increased along the network hierarchy, similar to previous observations in the brain[7]. These results show that the observed invariances can arise from the hierarchical structure with convolutional filtering without the contribution of structured spatial filters. Notably, the current result also suggests a possible scenario through which to understand how viewpoint invariant selectivity can arise in infant animals. From the similarities between the CNN and the biological brain models, i.e., that the fundamentals of both CNNs and sensory cortices are based on the hierarchical feedforward structure and that the process of convolution via weight sharing in CNNs can be approximated by a biological model[73–76] of periodic functional maps with hypercolumns in the visual cortex, our result may inspire insight into how innate invariance can arise in infant animals.

**Detection of face images using the responses of face-selective units**. We tested whether the selective responses of these face units could provide reliable information with which to detect between faces and non-face objects. During this task, face ($n = 40$) or non-face ($n = 40$) images were randomly presented to the networks, and the observed response of the final layer was used to train a support vector machine (SVM) to classify whether the given image was a face or not (Fig. 3a). First, we compared the detection performance of the SVM using a single unit randomly sampled from face-selective units and using units without selective responses to any image classes. We confirmed that the SVM trained with a single face-selective unit shows noticeably higher performance than those measured from shuffled responses, whereas the SVM trained with units without selectivity does not (Fig. 3b, Face unit vs. Response shuffled, $n_{face} = 465$, two-sided rank-sum test, $P = 2.97 \times 10^{-121}$, $r_{rbc} = 7.68 \times 10^{-1}$; Response shuffled vs. Non-selective unit, $n_{non-selective} = 7776$, two-sided rank-sum test, NS, $P = 1.10 \times 10^{-1}$, $r_{rbc} = 4.52 \times 10^{-2}$, two-sided

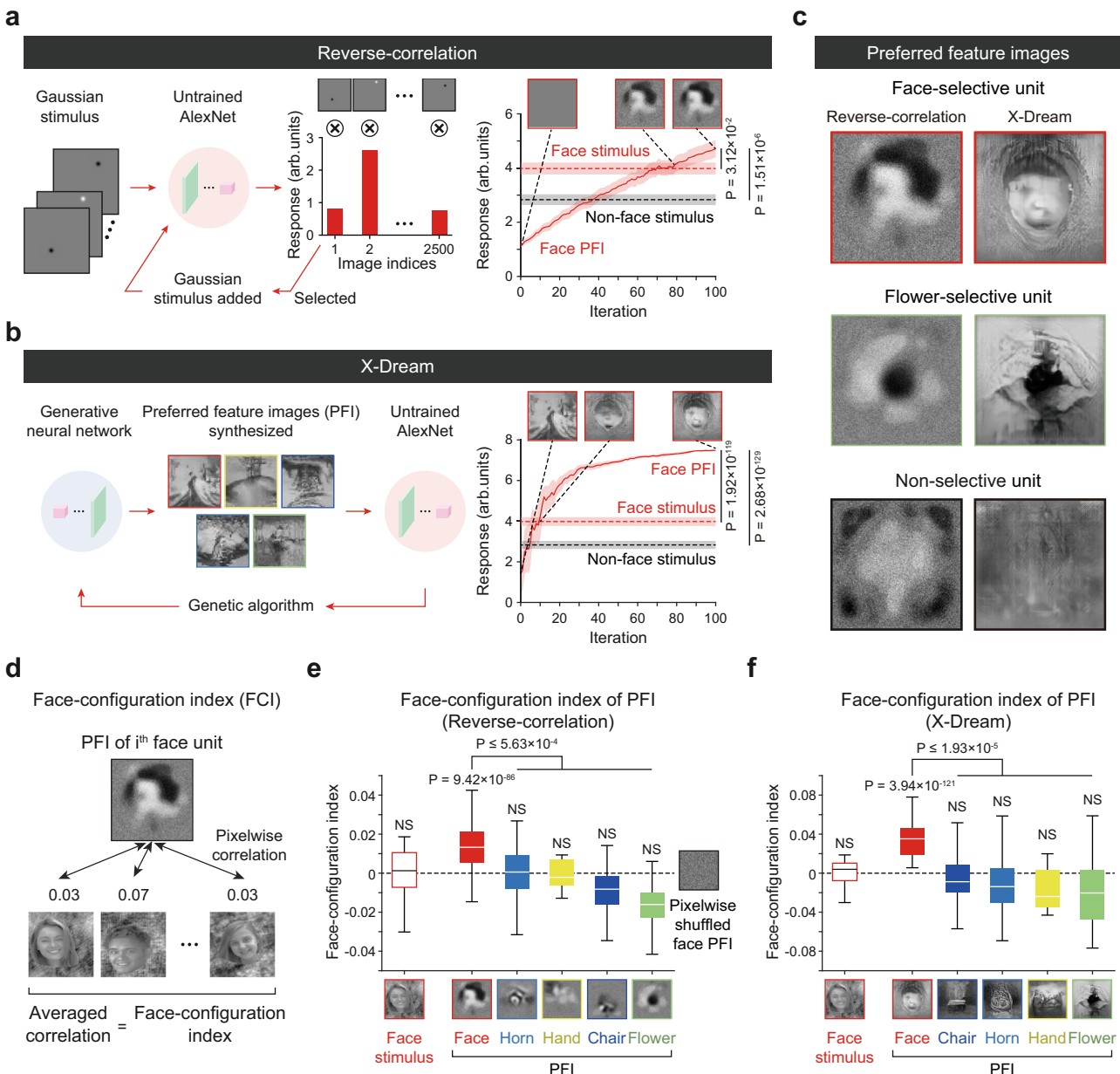

**Fig. 2 Preferred feature images of face-selective units in untrained networks. a** Measurements of preferred feature images (PFI) of target units in Conv5 from the reverse-correlation analysis[63]. Bright and dark 2D Gaussian filters were generated at a random position as an input stimulus set. The PFI was obtained as the summation of stimuli weighted by the corresponding responses. The initial preferred feature image was calculated from the local Gaussian stimulus set by the classical reverse-correlation method[63]. Then, a new stimulus set was generated as the summation of the obtained PFI and local Gaussian stimuli, with the second preferred feature image then obtained from a new stimulus set. This procedure was repeated to obtain the preferred feature image. **b** Schematics of the process used to achieve a preferred feature image (PFI) using a generative neural network (GAN) and a genetic algorithm (X-Dream)[25]. Synthesized images are generated by the GAN with image codes and are fed into an untrained network as input. The genetic algorithm finds a new image code that maximizes the response of the target unit. The PFI of a target unit is achieved after 100 iterations of this procedure. **c** The obtained preferred feature images, using the reverse-correlation method and X-Dream, of the face-selective unit, selective units to non-face class (flower), and units selective to none of the class. **d** Illustration of the face-configuration index (FCI) of a face unit's PFI. The FCI was defined as the pixel-wise correlation between the original face stimuli and the generated PFIs. **e** FCI of PFI, using the reverse correlation method, of units selective to each class ($n_{Face} = 465$, $n_{Hand} = 7$, $n_{Horn} = 772$, $n_{Flower} = 107$, $n_{Chair} = 63$). **f** FCI of PFI, using X-Dream, of the same units as the units used in (**e**). All box plots indicate the inter-quartile range (IQR between Q1 and Q3) of the dataset, the horizontal line depicts the median and the whiskers correspond to the rest of the distribution (Q1 − 1.5*IQR, Q3 + 1.5*IQR). The face images shown in panels (**d**)–(**f**) are selected examples from the publicly available dataset[59]. The original images are available at [http://vpnl.stanford.edu/fLoc/].

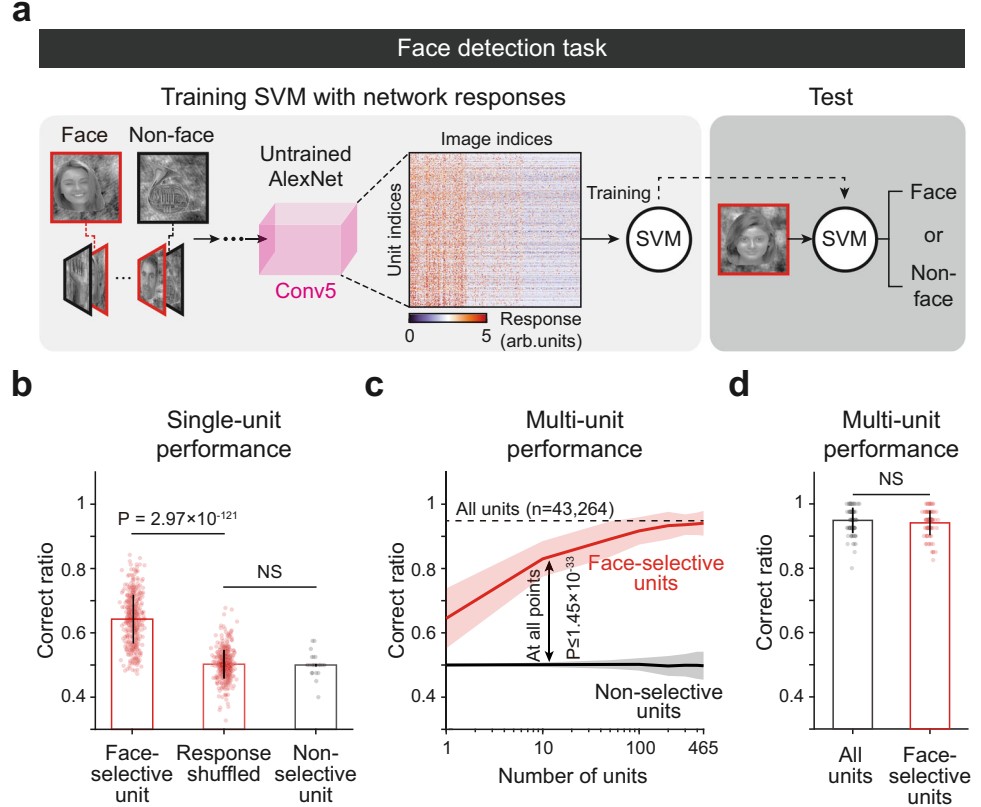

**Fig. 3 Detection of face images using the response of face units in untrained networks. a** Design of the face detection task and SVM classifier using the responses of the untrained AlexNet. During this task, face or non-face images were randomly presented to the networks and the observed response of the final layer was used to train a support vector machine (SVM) to classify whether the given image was a face or not. Among 60 images from each class (face, hand, horn, flower, chair, and scrambled face) that were not used for face unit selection, 40 images were randomly sampled for the training of the SVM, and the other 20 images were used for testing. The images shown are selected examples from the publicly available dataset[59]. The original images are available at [http://vpnl.stanford.edu/fLoc/]. **b** Performance on the face detection task using a single unit randomly sampled from face-selective units ($n = 465$) and units without selective responses to any image classes ($n = 7776$). The chance level was measured by the shuffled responses of face-selective units in the untrained network. The error bar indicates the standard deviation of each unit. Each bar indicates the mean and the error bar indicates the standard deviation of performance of each unit. **c** Performance of the face detection task using face-selective units and non-selective units when varying the number of units from 1 to 456. The dashed line indicates the detection performance using all units in Conv5 ($n = 43,264$). Each line indicates the mean and the shaded area indicates the standard deviation for 100 repeated trials of the random sampling of units. **d** Performance on the face detection task using face-selective units ($n = 465$) and then using all units in Conv5 ($n = 43,264$). Each bar indicates the mean and the error bar indicates the standard deviation for 100 repeated trials of the random sampling of units.

Kolmogorov–Smirnov test, $P = 1.93 \times 10^{-1}$, $d = 2.12 \times 10^{-2}$). Then, extending the test to various numbers of units, we compared the detection performance of this SVM using face-selective units with the performance when using the same number of randomly sampled non-selective units. We confirmed that the SVM trained with multiple face-selective units shows noticeably better performance than that trained with the same number of non-selective units, as the number of units used in each condition was varied from $n = 1$ to 465 (total number of face units in untrained networks) (Fig. 3c, Face vs. Non-selective units, $n_{trial} = 100$, two-sided rank-sum test, $P \leq 1.45 \times 10^{-33}$, $r_{rbc} \geq 8.74 \times 10^{-1}$). We also found that the SVM using face units ($n = 465$) nearly matches the performance of the SVM using all units in the final layer ($n = 43,264$) (Fig. 3d, Face vs. All units, $n_{trial} = 100$, two-sided rank-sum test, NS, $P = 1.90 \times 10^{-1}$, $r_{rbc} = 9.29 \times 10^{-2}$, two-sided Kolmogorov–Smirnov test, $P = 1.90 \times 10^{-1}$, $d = 9.20 \times 10^{-3}$). Furthermore, we found that face units enable the networks to detect faces with various sizes, positions, and rotations even when such image conditions were held constant when training the SVM classifier (Supplementary Fig. 11).

Notably, we also found that the SVM can successfully detect faces when it is trained with the responses of units selective to non-face classes, similar to the results in a previous experiment in human[77], whereas it failed to detect faces with units not selective to any of the classes (Supplementary Fig. 12). To compare our results with the experiment condition of the previous human experiment[77], we first trained the SVM using the responses of four distinct populations: (1) all of the units selective to each class (All-selective), (2) units selective to non-face classes (Non-face-selective), (3) face-selective units only (Face-selective), and (4) units not selective to any of the classes (Non-selective) (Supplementary Fig. 12a). As a result, we found that the SVM trained with non-face-selective units showed a performance comparable with the results of Haxby et al.[77] (Supplementary Fig. 12b, c). Interestingly, the performance was also comparable with those with all-selective units (Supplementary Fig. 12b) and those with face-selective units only, similar to the results in a previous experiment in human[77]. This result is understandable considering that there are only five image classes; thus, even non-face-selective units can provide information for discriminating face and non-face images by generating different levels of activities for each class. Taken together, these results imply

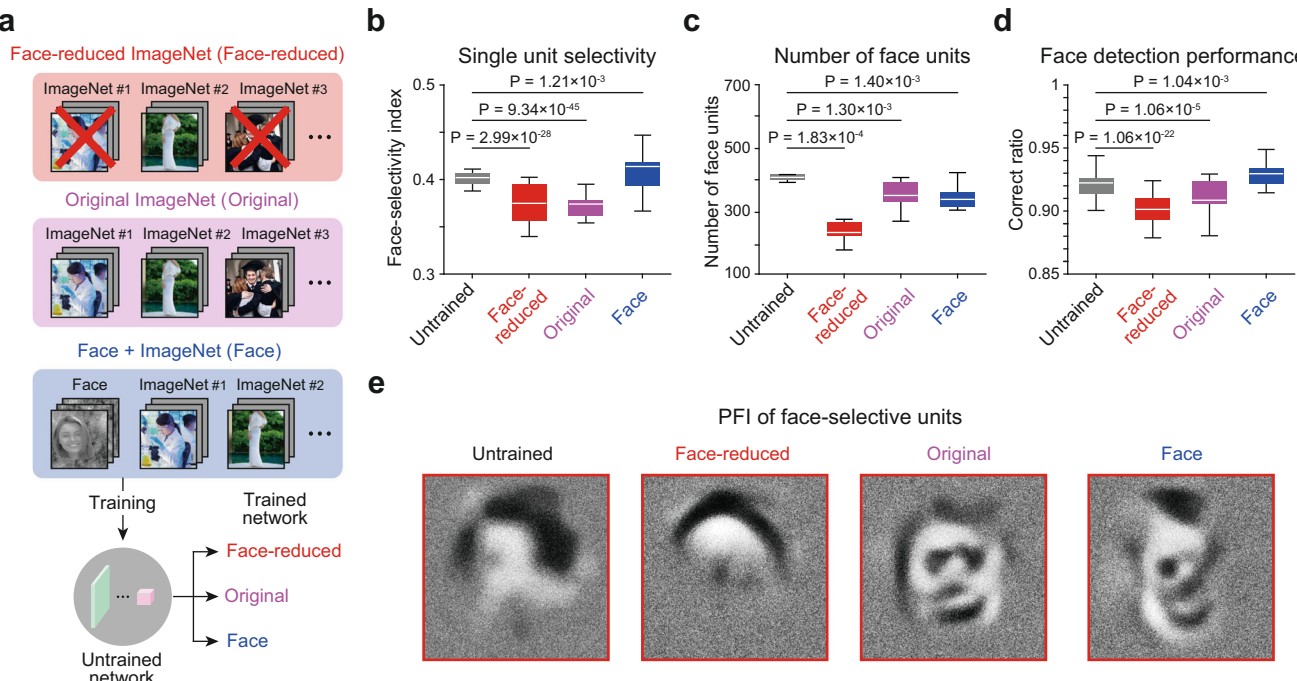

**Fig. 4 Effect of training on face-selectivity in untrained networks. a** Three different datasets modified from publicly available ImageNet[65] were used for the training of the network for image classification: (1) face-reduced ImageNet, (2) original ImageNet, and (3) ImageNet with added face images. For copyright reasons, the face image shown here is not the actual image used in the experiments. The original images are replaced with images with similar contents for display purposes. The original images are available at [https://www.image-net.org/download]. Images shown are available at [https://www.shutterstock.com, http://vpnl.stanford.edu/fLoc/[59]]. See Methods for details). **b** Face-selectivity index of face-selective units in untrained networks and in networks trained with the three datasets ($n_{\text{Untrained}} = 4267$, $n_{\text{Reduced}} = 2452$, $n_{\text{Original}} = 3561$, $n_{\text{Face}} = 3585$). **c** The number of face-selective units in untrained networks and in networks trained with the three datasets ($n_{\text{Net}} = 10$). **d** Face detection performance of untrained networks and of networks trained with three different datasets ($n_{\text{trial}} = 1000$). **e** The obtained preferred feature images (PFI), using the reverse-correlation method of the face-selective unit on each network. All box plots indicate the inter-quartile range (IQR between Q1 and Q3) of the dataset, the horizontal line depicts the median and the whiskers correspond to the rest of the distribution ($Q1 - 1.5*IQR$, $Q3 + 1.5*IQR$).

that the information provided by selective units that emerge in the untrained networks is sufficient to detect between faces and non-face objects.

**The emergence of face-selectivity in trained DNNs.** We tested a scenario in which our current model can corroborate the conflicting observations regarding the role of visual experience for the development of face-selectivity. A previous report suggested that visual experience is necessary for the emergence of face-selectivity by showing that monkeys raised without exposure to faces lack face-selective domains[26]. On the other hand, another recent study showed that the face-selective area develops robustly in congenitally blind humans, suggesting that visual experience is not necessary for face-selectivity[26]. Regarding these conflicting results, we examined how face-selective units in untrained networks can be affected by training with visual inputs.

To investigate the effect of training on a face image set, we prepared the following three different stimulus sets: (1) face-reduced ImageNet: 500 classes including no recognizable face images were manually curated from the ILSVRC 2010 dataset according to a visual inspection by the authors, (2) the original ImageNet, and (3) the original ImageNet with added face images used in the current study (Fig. 4a). Then, the network was trained with each of these image sets. First, we found that the FSI of the face-selective units was significantly decreased after being trained to the face-reduced image set (Fig. 4b, Untrained vs. Face-reduced, $n_{\text{Untrained}} = 4267$, $n_{\text{Reduced}} = 2452$, two-sided rank-sum test, $P = 2.99 \times 10^{-28}$, $r_{\text{rbc}} = 6.76 \times 10^{-1}$), whereas it was

increased after being trained to the face-including image sets (Untrained vs. Face-included, $n_{\text{Face}} = 3585$, two-sided rank-sum test, $P = 1.21 \times 10^{-3}$, $r_{\text{rbc}} = 2.60 \times 10^{-1}$). Notably, the FSI was significantly decreased after being trained to the original ImageNet dataset that contains images of faces but has no group labeled as face (Fig. 4b, Untrained vs. Original, $n_{\text{Original}} = 3561$, two-sided rank-sum test, $P = 9.34 \times 10^{-45}$, $r_{\text{rbc}} = 8.15 \times 10^{-1}$). This suggests that the tuning of face-selective units could either be sharpened or weakened by training with distinct stimulus sets.

Next, we found that the number of face-selective units observed was greater in the network trained with face-including image set compared to that trained to face-reduced images (Fig. 4c, Untrained vs. Trained, $n_{\text{Net}} = 10$, two-sided rank-sum test, $P \leq 1.40 \times 10^{-3}$, $r_{\text{rbc}} \geq 5.72 \times 10^{-1}$). Interestingly, however, we found that the number of face-selective units, when trained to face-including images, appeared to be smaller than that of untrained networks. These results imply that the training process of the network to face-including images selectively sharpens the tuning of face units so that the selectivity of strongly tuned units is sharpened while the weakly tuned units are pruned. In this condition, the face detection performance of the networks would improve in face-trained networks even if the number of face units decreased compared to the initial, untrained condition. To validate this scenario, we trained the SVM using the response of face-selective units for a face detection task in an untrained network and in the three networks trained to each type of data set. As predicted, we found that the face detection performance was significantly increased in the networks trained to the face-

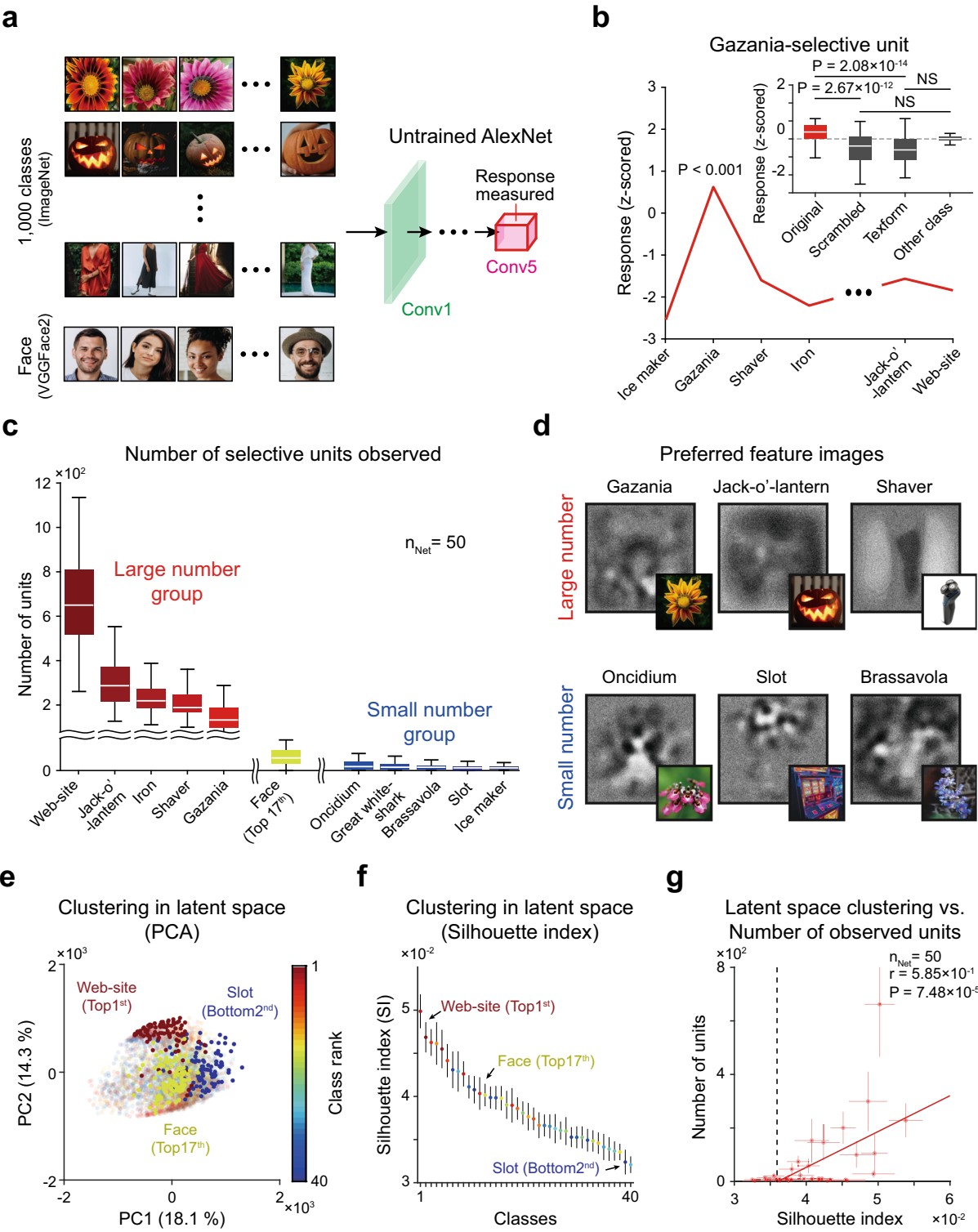

including image set compared to that of the untrained network (Fig. 4d, Untrained vs. Face included, $n_{trial} = 1000$, two-sided rank-sum test, $P = 1.04 \times 10^{-3}$, $r_{rbc} = 2.60 \times 10^{-1}$), whereas the face detection performance of the network trained to the face-reduced image set was significantly decreased compared to the untrained network (Fig. 4d, Untrained vs. Face-reduced, $n_{trial} = 1000$, two-sided rank-sum test, $P = 1.06 \times 10^{-22}$, $r_{rbc} = 6.42 \times 10^{-1}$). Furthermore, we found that the PFI of the face-selective unit shows a clear face configuration in the network trained to face-including natural images, whereas the face

configuration is disrupted in network trained to face-reduced dataset (Fig. 4e). This result is consistent with the previous observation of decreased face-selectivity in face-deprived monkeys[26].

**The emergence of selectivity to various objects in untrained DNNs.** Lastly, we investigated the possibility that units selective to various objects other than faces also emerge similarly in untrained neural networks. For this, we measured the responses

**Fig. 5 ImageNet category-selective units in untrained networks. a** The responses of units in untrained networks to the images of 1000 ImageNet[65] classes and to face images (VGGFace2)[62]. **b** Average tuning curve of gazania selective units. (Inset) Responses of gazania-selective units to the original gazania ($n = 100$), the scrambled gazania ($n = 100$) and texform gazania images ($n = 100$). **c** The number of selective units for 39 classes in which selective units are observed. The error bar indicates the standard deviation of 50 random networks. **d** Sample preferred feature images achieved by reverse-correlation analysis and stimulus images (inset). **e** Visualization of the PCA (principal component analysis)[78] analysis results (only two principal components (PC) are shown) using the Conv5 unit responses to each class in untrained networks. The analysis was performed using 3999 principal components, and the top $140 \pm 32$ components contained 75% of the variance. **f** The silhouette index[79] of the Conv5 unit responses was measured using all principal components to estimate the consistency of data clustering. Each dot indicates the mean and the error bar indicates the standard deviation of 50 simulations of randomly initialized networks. The error bar indicates the standard deviation of 50 simulations of randomly initialized networks. **g** Correlation between the silhouette index and the number of selective units observed (Pearson correlation). Each dot indicates the mean and the error bar indicates the standard deviation of 50 random networks. All box plots indicate the inter-quartile range (IQR between Q1 and Q3) of the dataset, the horizontal line depicts the median and the whiskers correspond to the rest of the distribution (Q1 − 1.5*IQR, Q3 + 1.5*IQR). For copyright reasons, the images in panels (**a**) and (**d**) are not the actual images used in the experiments. The original images are replaced with images with similar contents for display purposes. The original images are available at [https://www.image-net.org/download, https://arxiv.org/abs/1710.08092]. Images shown are available at [https://www.shutterstock.com] (see Methods for details).

of units in random networks to a stimulus dataset of ImageNet containing 1000 classes of objects (Fig. 5a). As a result, we found that selective units are observed in 39 classes among the 1000 classes (Fig. 5b, c). From the analysis using scrambled and texform control images, we confirmed that these object-selective units are not particularly selective to local image parts but are selective to a whole object, similar with units selective to faces (Fig. 5b, inset, Gazania vs. Scrambled gazania, $n = 100$, one-sided rank-sum test, $P = 2.67 \times 10^{-12}$, $r_{rbc} = 4.89 \times 10^{-1}$; Gazania vs. Texform gazania, $n = 100$, one-sided rank-sum test, $P = 2.08 \times 10^{-14}$, $r_{rbc} = 5.36 \times 10^{-1}$, Scrambled gazania vs. other-class, $n = 100$, one-sided rank-sum test, NS, $P = 1.00$, $r_{rbc} = 3.17 \times 10^{-1}$, one-sided Kolmogorov–Smirnov test, $P = 1.02 \times 10^{-2}$, $d = 2.97 \times 10^{-2}$; Texform gazania vs. other-class, $n = 100$, one-sided rank-sum test, NS, $P = 1.00$, $r_{rbc} = 4.20 \times 10^{-1}$, one-sided Kolmogorov–Smirnov test, $P = 3.45 \times 10^{-2}$, $d = 2.55 \times 10^{-2}$). This result suggests that units selective innately to various objects such as faces can emerge in untrained neural networks.

Next, to investigate the emergent condition of units selective to each object further, we sorted those 39 classes according to the number of selective units observed and computed the PFI using the RC method (Fig. 5c, d). In general, we observed a tendency in which the PFIs of the large number group showed a relatively simple configuration of each preferred object class that was visually observable (Fig. 5d, top), whereas those of the small number classes represented a more complicated structure of the PFI (Fig. 5d, bottom). From this result, we hypothesized that objects with a simple configuration, such as faces, can induce stronger clustering in the latent space representation than those of other object classes and therefore may have a greater likelihood of generating units selective to them. To validate our hypothesis, we used the dimension-reduction method[78] to compare a clustered representation of each object class in terms of the raw pixel values and in the responses of Conv5. For quantification of the representational clustering of each class, we measured the silhouette index[79] to estimate the consistency of data clustering. We found that there is a strong correlation between the silhouette index in the Conv5 latent space and the number of selective units observed (Fig. 5e–g, Pearson correlation coefficient, $n_{Net} = 50$, $r = 5.85 \times 10^{-1}$, $P = 7.48 \times 10^{-5}$). This result demonstrates that objects with a simple profile, readily distinguishable from those of other objects statistically, lead to a strong clustering of abstracted responses in the DNN and are more likely to generate units selective to it. Furthermore, the relationship between the silhouette index and the number of units observed shows that the number of units increases as the silhouette index increases,

with no selective units observed when the silhouette index is below 0.036 (Fig. 5g, black dashed line). This result implies that there may be a threshold of the clustering level in the response embedding space, by which a unit selective to that object class can be defined and observed. In neuroscience, this may provide a possible explanation of why face-selective neurons are observed in various experiments while neurons selective to other objects are not observed as readily. Thus, the observed face-selectivity may not be a special case of tuning, whereas selectivity to other visual objects can also arise in random networks simply due to the relatively simple configuration of the corresponding geometric components.

## Discussion
We showed that a biologically inspired DNN develops face-selective units without training, solely from statistical variations of the feedforward projections. These results suggest that the statistical complexity embedded in the structure of the hierarchical circuit[73,74,80] can initialize primitive cognitive functions such as face-selectivity in untrained neural networks. Although the performance of a DNN appears to be similar to that of the brain on certain visual tasks, there are critical differences between biological brains and DNN models, such as convolutional filtering in DNN models, which is not biologically plausible. However, although DNNs are not an impeccable model of the visual pathway of the brain, the current results provide a possible scenario for understanding how primitive visual functions such as face detection can initially arise in early brains before learning begins with sensory inputs.

State-of-the-art studies using random networks provide important clues regarding how these selective units can arise spontaneously in untrained model networks. Recently, it was reported that a network can classify an untrained image class by combining pre-trained readout units[81], a process known as zero-shot learning. Our results suggest that such zero-shot learning is even possible without any pre-trained readout units. It was also reported that an artificial network that learns visual features with random, untrained weights can perform image classification tasks[53,54]. Jarrett et al. showed that features from a randomly initialized one-layer convolutional network could classify the Caltech 101 dataset with a performance level similar to that of a fine-tuned network, consistent with the mathematical notion that a combined convolutional and pooling architecture could develop spatial frequency selectivity and translation invariance[82]. Overall, these results suggest that the initial structure of random networks plays an important role in visual feature extraction before the training process. They also imply that complex types of feature

selectivity, such as face-selectivity, may arise innately from the structure of the random feedforward circuitry.

Furthermore, recent model studies using the lottery ticket hypothesis[55,56] showed that randomly weighted neural networks contain subnetworks that can perform tasks without modifying the initial random weight values, implying that functional architectures can emerge from random initial wirings. This model and our current model have a common aspect in that functional structures can emerge without any modification to the initial random weights. However, the lottery ticket hypothesis showed that a random subnetwork can perform tasks without learning, while our model demonstrates that the functional tuning of single units can arise spontaneously. Indeed, our model not only suggests a possible mechanism of the origin of the functional tuning of single neurons in the early developmental process in the brain but also provides insight into the mechanism underlying the emergence of functional subnetworks, possibly from tuned individual units that emerge in random artificial neural networks.

Similarly, the theory of reservoir computing suggests that the circuits required for higher-order cognitive functions, such as image classification, may already exist in untrained, random recurrent neural networks. In this scenario, higher-order cognitive functions can be achieved only by training a read-out network, as suggested by the lottery ticket hypothesis[83,84]. Interestingly, our results in the face-detection task performed with the SVM are comparable to the concept of reservoir computing, as the training of the SVM with the responses of untrained networks is consistent with the procedure of training a read-out projection from random networks in reservoir computing. It is notable that recent studies suggest that the random network can perform this task if the read-out units are selected via a prior understanding of the system. For example, object classification can be performed by a random network if read-outs are chosen by a synaptic rule observed in the brain[85,86]. While these models focus on the innate functions of networks in that a high dimensional space generated by a random network can perform various tasks without learning, our current results demonstrate that the functional tuning of single units (comparable to neuronal tuning in biological brains) can arise in random networks without any further training of the read-out process, which is distinguished from the main idea of the reservoir computing model.

It is important to note that the current results do not necessarily mean that innate face-selectivity is the tuning observed in adult animals. There is a wealth of evidence showing that higher areas of the visual cortex are immature in the early development stage and that the corresponding functional circuit is modulated by visual experience[87–90]. There is also strong evidence that the IT region, where face-selective neurons are observed, can be altered by early experience[91,92]. Considering the anatomical and physiological changes that occur over the first postnatal year, the innate template of face-selective neurons in very early developmental stages must be refined by later visual experience, including both bottom-up and top-down processes[93,94]. This scenario may be supported by recent observations of the existence of the proto-retinotopic organization and rough face-patches in higher regions of the visual cortex[6,92,95]. Moreover, observations of the early development of cortical circuits may provide further support to our scenario. Retino-thalamic feedforward projections are composed of noisy local samplings that result in unrefined receptive fields in individual thalamus neurons[96]. This is comparable to randomly initialized convolutional kernels before training. Inborn feature-selectivity generated in this early cortex may provide an initial basis for various visual functions.

Importantly, our results imply that innate face-selectivity may arise spontaneously from feedforward wirings, but this requires further evidence and examinations before one can argue whether the mechanism can indeed be considered spontaneous. Specifically, our model suggests that the random weights in CNNs (comparable with random feedforward projections in biological networks) can generate selective responses of units to face images. Then, considering that the development of random weights does not require any training or experience, this could be considered as innate face-selectivity. These results imply a possible scenario in which the random feedforward connections that develop in young animals may be sufficient for initializing primitive face-selectivity. Arguably, this face-selectivity can be considered to arise spontaneously under the assumption that random feedforward wirings can arise without a complicated process. In this scenario, the emergence of face-selectivity may not require genetically programmed innateness but may simply originate from the statistical complexity of random wirings. However, it must be noted that an alternative interpretation is also possible: the observed face-selectivity is innate but may not be considered spontaneous because the initial development of random feedforward wiring requires a certain mechanism programmed in one's genes. Particularly, regarding the question of why face-selective neurons are observed consistently in the same brain regions, the role of programmed genes may be critical—face-selective neurons can simply originate from random feedforward wirings in hierarchical networks, but this neuronal selectivity can only be refined and reserved in a particular face-patch region in the brain, most likely controlled by a blueprint of the brain circuitry programmed in genes. Detailed arguments pertaining to this issue may become possible when further evidence of the developmental mechanism of random wirings and the corresponding dependence on gene coding becomes available.

Our findings suggest a scenario in which proto-organization for cognitive functions may be spontaneously generated, after which training with data can sharpen and specify the selectivity of networks. A recent fMRI study of the inferior temporal cortex in infant and adult monkeys also shows a biological example of this scenario[6]. Livingstone et al. show that neurons broadly tuned to faces are already observed in infant monkeys (~1 month old) and that the region where these neurons are observed is identical to where the face neurons of adult monkeys are observed. This result implies that the innate template of face-selective neurons in infant monkeys may develop spontaneously and be later fine-tuned during the early visual experience. This is consistent with the model scenario of our study, which shows that face-selective neurons emerge in the early cortex provide a basis for early face detection, with this neuronal tuning refined further when learning begins with visual experience with both bottom-up and top-down processes[93,94].

In summary, our results suggest that face-selectivity can arise in a completely untrained neural network with the random initial wiring of hierarchical feedforward projections. These findings may provide insight into the origin of innate cognitive functions in both biological and artificial neural networks.

## Methods

**Neural network model**. We used AlexNet[45] as a representative model of the convolutional neural network. This network consists of feature extraction and classification networks. The feature extraction network consists of five convolutional layers with rectified linear unit (ReLU) activation and a pooling layer, while the classification network has three fully connected layers. The detailed parameters of the architecture were sourced from Krizhevsky et al.[45], which provided the models for V4 and IT[48].

To determine the origin of face-selective neurons, the randomly initialized networks were examined. For the untrained AlexNet, the weights and biases of each convolutional layer were initialized from a Gaussian distribution or a uniform distribution with a zero mean and the standard deviation set to the square root of one over the number of units in the previous layer. This was done to balance the strength of the input signals across the layers, and this approach follows previous research on efficient network initialization processes[58].

**Stimulus dataset.** Seven types of visual stimuli from six datasets were used. (1) A low-level feature-controlled stimulus set[59] was selected and modified from publicly available images in human fMRI study[59] (http://vpnl.stanford.edu/fLoc/) and was used to find units that responded selectively to face images. Specifically, 260 images were prepared for each class (face, hand, horn, flower, chair, and scrambled face). Then, 200 images were randomly sampled from each class and used for face unit selection. Among the 60 remaining images in each class, 40 images were used for the training of the SVM, and the other 20 images used for testing. Each item was overlaid on a group of phase-scrambled background images. This was designed to reduce inter-class differences across various low-level properties, in this case, luminance, contrast, size, position (Supplementary Fig. 1b–f), and the degree of intra-/inter-class image similarity (Supplementary Fig. 1g, h). To validate the face-selective response to novel face stimulus set, we used (2) 16 face images used in Tsao et al. (2006, 2010)[5,7] provided by D. Tsao group from personal communication, (3) 50 face images from open access VGGFace2 dataset used in Cao et al.[62] (https://github.com/ox-vgg/vgg_face2); (4) 50 face images artificially generated by the FaceGen simulator (singular inversions; FaceGen Modeller Pro) in color and grayscale (https://facegen.com/). (5) To investigate the invariance of face-selective units to face images of various sizes, positions, and rotation angles, the image set was generated after modifying the size, position, and rotation angle of the faces and other objects in the similarity-controlled stimulus set[59]. (6) Viewpoint dataset: This set was used to find units that invariantly responded to face images of different viewpoints. This dataset consists of five angle-based viewpoint classes (−90°, −45°, 0°, 45°, 90°) with 10 different faces obtained from the publicly available Point' 04 dataset (http://crowley-coutaz.fr/Head%20Pose%20Image%20Database.html) used in Gourier et al.[97]. (7) To investigate the possibility of units selective to various objects, we used a publicly available ImageNet dataset[65] (https://www.image-net.org/download). For copyright reasons, the images in Figs. 1a, 4a, and 5a, d are not the actual images used in our experiments. The original images are replaced with images with similar contents for display purposes. Alternative images used in this study are purchased from https://www.shutterstock.com with a standard image license, which includes rights to publish in e-publication and printed in physical form as part of a copy of magazines, newspapers, and books. For all datasets, the image size of the input to AlexNet was fixed at 227 × 227 pixels.

**Analysis of responses of the network units.** In our model, a unit refers to a unit component at each position of the channel in an activation map of the network. We defined this unit in a convolutional network as a simplified model of a biological unit (a single neuron or a group of neurons that generates a tuned activity), considering that the dynamics of a single neuron in biological brains can be estimated from its receptive field, which behaves as a spatiotemporal filter at a local cortical position retinotopically matching the external visual space. Based on a previous study[51], face-selective units were defined as units that had significantly higher mean responses to face images than to images in any non-face class ($P < 0.001$, two-sided rank-sum test). To estimate the normalized response, the response of each unit was $z$-scored using the average and the standard deviation of responses to stimulus images. To quantify the degree of tuning, an FSI of a single unit was defined as in previous experimental research[17]

$$\text{FSI} = \frac{\left(\bar{R}_{\text{face}} - \bar{R}_{\text{nonface}}\right)}{\sqrt{\left(\sigma^2_{\text{face}} + \sigma^2_{\text{nonface}}\right)/2}} \quad (1)$$

where $\bar{R}_{\text{face}}$ is the average response to face images and $\bar{R}_{\text{nonface}}$ is the average response to all non-face images. An FSI of 0 indicates equal responses to face and non-face objects.

Among the face-selective units found, a face viewpoint-invariant unit was defined as a unit for which the response was not significantly different (one-way ANOVA, $P > 0.05$, Bonferroni adjustment $n = 5$) among all viewpoint classes. Similar to the face-selective units, the viewpoint-specific units were determined by the mean response of the preferred viewpoint class being significantly higher than that for any other viewpoint (one-way ANOVA, $P < 0.05$, Bonferroni adjustment $n = 5$). We defined mirror-symmetric tuning as the condition that arises when a viewpoint-specific face-selective unit has a symmetric shape of the tuning curve (one-way ANOVA, $P < 0.05$, Bonferroni adjustment $n = 5$; i.e., a unit shows peak responses at −45° and 45° or −90° and 90°). The invariance index of a single unit was defined as the inverse of the standard deviation of the average responses for images within each viewpoint class.

**Class-clustering index of network response.** To visualize the network responses to the similarity-controlled stimulus set, a principal component analysis[78] was used for dimension reduction. By minimizing the difference between the original and low-dimensional distributions of neighbor distances, a 2D representation of the responses of the fifth convolutional layer was obtained. To quantify the level of clustering of each class, we measured the Silhouette index (SI)[79] to estimate the consistency of data clustering. The silhouette index SI[79] for the $i$th point is defined as

$$\text{SI}_i = (b_i - a_i)/\max(a_i, b_i) \quad (2)$$

where $a_i$ and $b_i$ refer to the intra-class distance and the inter-class distance for the $i$th point, respectively. The intra-class distance was defined as the average distance between the centroid of the class and each data point in the class. The inter-class distance was defined as the average distance between the centroids of each cluster[98].

The same analysis was also performed for the stimulus set that added the face class to ILSVRC2010.

**Face vs. non-face detection task for the network.** A face vs. non-face detection task was established to investigate whether face-selective units could perform basic face perception. To determine if this was possible, an SVM was trained with network responses to images and was then made to predict whether a class of unseen images was or was not a face. To avoid the double-dipping issue[99] we undertook the training and testing of the SVM using distinct sets of images, where 260 images were prepared for each class, and 200 images were then randomly sampled from those images and used for face unit selection. Among the 60 remaining images in each class, 40 images were used for the training of the SVM, and the other 20 images were used for testing. The label of the training set was then changed to a binary class: face or non-face. For each trial, the SVM was trained with the relationship between the fifth layer's response to the training set and the new training label. After a training session, the model predicted the test label using the network response to the test set. Furthermore, to test whether the responses of face-selective units could detect faces varied in different ways, we trained an SVM with network responses for center-view face images ($N = 40$) and non-face images ($N = 40$). The model then predicted test labels using the responses of face-selective units to new test sets which consisted of a face and non-face images with different types of variation, in this case, different sizes, positions, and rotation angles.

**Preferred input feature (receptive field) analysis.** To visualize the preferred input feature of target units, the receptive field was estimated by the RC method[63] with multiple iterations. The initial stimulus set was generated as 2500 random local 2D Gaussian filters. Such stimuli were weighted by the corresponding responses and were added as an initial preferred feature image. Then, to detect the preferred feature more accurately, we calculated the PFI iteratively; the PFI of the next iteration was calculated by a new stimulus set consisting of the summation of the current PFI and 2500 random local 2D Gaussian filters (Fig. 2a). We repeated 100 iterations and obtained the final PFI.

To obtain the preferred feature images of target units, we used a generative adversarial network (X-Dream)[64]. X-Dream consists of a generative adversarial network (GAN) with a genetic algorithm as the optimization algorithm of the responses. We used a GAN[64] pre-trained with natural images (ILSVRC 2012). The response optimization algorithm[25] finds the optimal image code that maximizes the response of the target unit. In this algorithm, a single image code consists of 1000 initial values randomly sampled from a zero-centered Gaussian distribution with a standard deviation of 0.5. In each iteration, 50 image codes are randomly generated, and the five image codes with the highest optimization score are preserved for the next iteration while the other 45 codes are recombined through a pairwise-randomization step before the next iteration. Then, individual values of an image code are randomly mutated with a probability of 0.01; i.e., each value is replaced with a random number drawn from a zero-centered Gaussian with a standard deviation of 0.5. The preferred feature image of a target unit is achieved after 100 iterations.

**Trained network model.** To investigate the effect of training on a face image set, we prepared the following three different stimulus sets: (1) face-reduced ImageNet: images with those including a face excluded (ILSVRC 2010; 500 classes), (2) the original ImageNet (1,000 classes), and (3) the original ImageNet with added face images used in the current study (1,001 classes). The network was trained with each of these image sets using a stochastic gradient descent algorithm. Detailed training parameters were adapted from Krizhevsky et al., 2012: batch size = 128, momentum = 0.9, weight decay = 0.0005, training epoch = 90, learning rate = 0.01, learning rate decay = 10 times for every 30 epochs.

**Statistics.** All sample sizes, exact $P$ values, and statistical methods are indicated in the corresponding text, figure legends, and Supplementary Tables 2 and 3. A rank-sum test was used for all analyses, except for the number of face units across convolutional groups (Kolmogorov–Smirnov test; Supplementary Fig. S2h), the face detection task (Kolmogorov–Smirnov test; Fig. 3d, Supplementary Fig. S11b, Supplementary Fig. S12b), the detection of viewpoint-invariant and -specific units (one-way ANOVA with Bonferroni adjustment, Supplementary Fig. 8a) and a connectivity analysis (one-way ANOVA with Bonferroni adjustment, Supplementary Fig. 8c, d). The devisor of all Bonferroni adjustments was five viewpoint groups ($n = 5$). All statistical tests used to determine statistical significance were two-sided, except for the chance level of FSI (one-sided, Fig. 1f, Supplementary Fig. S3), response to controlled face images and novel faces (Fig. 1g, h), chance level of the face-configuration index (one-sided, Fig. 2e, f), response to controlled gazania images (Fig. 5b), intra-class image similarity (Supplementary Fig. S1g), the effective range of image correlation (Supplementary Fig. S5f), average weight between Conv4 and Conv5 (Supplementary Figs. S8c, d, and S9b) and the number of viewpoint invariant units (Supplementary Fig. S9h). All error bars and shaded areas indicate the standard deviation, except for response to PFIs (standard error; Fig. 2a, b), connectivity analysis (standard error; Supplementary Figs. S8c, d, S9b, d, f, h), viewpoint invariance index (standard error; Supplementary Fig. S8g). Box plots in Figs. 1e–h, 2e, f, 4b–d and 5b, and in Supplementary Figs. 1d–h, 2h, 3, 7b,

c, and 9d, h indicate the inter-quartile range (IQR between Q1 and Q3) of the dataset, the horizontal line depicts the median and the whiskers correspond to the rest of the distribution ($Q1 - 1.5*IQR$, $Q3 + 1.5*IQR$). In Supplementary Tables 2 and 3, we included the effect size alongside all $P$ values for each statistical test: the rank-biserial correlation value[100] for the rank-sum test, and Cohen's $f^{2,101}$ for ANOVA test, the dissimilarity value[102] for Kolmogorov–Smirnov test, respectively.

**Reporting summary**. Further information on research design is available in the Nature Research Reporting Summary linked to this article.

## Data availability
The source datasets used in this work are available at https://doi.org/10.5281/zenodo.5637812.

## Code availability
MATLAB (2018b) was used to perform the analysis. The MATLAB codes used in this work are available at https://github.com/vsnnlab/Face.

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

## Acknowledgements

We are grateful to Gabriel Kreiman, Daeyeol Lee, and Min Whan Jung for discussing and providing comments on earlier versions of this manuscript. This work was supported by the National Research Foundation of Korea (NRF) grants funded by the Korean government (MSIT) (Nos. NRF-2019R1A2C4069863, NRF-2019M3E5D2A01058328, NRF-2021M3E5D2A01019544) and the Singularity Professor Research Project of KAIST (to S.P.).

## Author contributions

S.P. conceived the project. S.B., M.S., G.K., J.J., and S.P. designed the model. S.B. and M.S. performed the simulations. S.B., M.S., G.K., J.J., and S.P. analyzed the data. S.B. and M.S. drafted the paper and designed the figures. S.P. wrote the final version of the paper with input from all authors. All authors discussed and commented on the paper.

## Competing interests

The authors declare no competing interests.
