## [Peer Review File · Nature Communications]

Reviewers' Comments:

Reviewer #1:

Remarks to the Author:

This seems to me an interesting and theoretically very relevant paper. I have some suggestions for improvements, however.

- One crucial finding in several species (and in human newborns as well, e.g. Buiatti et al. 2019) is the inversion effect, the lack of preference (or the difficulty in recognition) of upside down faces. It seems that the authors' network does not show such an effect and this needs to be discussed.
- In order to check for the lack of face selectivity in face-deprived monkeys it would be interesting trying to train the network on a set of stimuli without faces to see whether the face-selective neurons would then disappear.
- The Introduction is too primate-centric in my view, and convey to the generic reader the impression that faces are special only for primates, which is not correct (e.g. Rosa Salva et al (2011). The Evolution of Social Orienting: Evidence from Chicks (*Gallus gallus*) and Human Newborns. PLoS ONE, 6(4): e18802; Rosa Salva et al (2010). Faces are special for chicks: Evidence for inborn domain-specific mechanisms underlying spontaneous preferences for face-like stimuli. Developmental Science, 13: 565-577; Versace et al. (2020) Early preference for face-like stimuli in solitary species as revealed by tortoise hatchlings. Proceedings of the National Academy of Sciences Sep 2020, 117 (39) 24047-24049; DOI: 10.1073/pnas.2011453117. Note that these comparative studies provide accurate controls for the role of experience to a degree which would be not possible for e.g. human newborns. The issue is important because neural networks should, at the end, be able to reproduce the biological phenomena in their largest generality.

Reviewer #2:

Remarks to the Author:

This manuscript uses deep net modeling to ask whether face selective responses can arise spontaneously in a system that has never been trained on visual input. The question is of fundamental importance, the general idea of the study clever, and the writing is admirably engaging and clear. It is further commendable that the authors have placed their code in an open accessible location. I do however have some nontrivial concerns which render the evidence not yet very convincing to me, which perhaps the authors could address in a revision.

1. As far as I can tell, the entire analysis in Fig. 3 is double-dipping: the same data are used to train and test an SVM that have been used to select the units entering this analysis in the first place. This analysis is famously invalid (see e.g. Kriegeskorte, et al, Nature neuroscience, 2009) and this analysis should be removed from the paper.
2. The finding of putatively face-selective responses is intriguing but preliminary at this point, because the authors do not show that the claimed face selectivity generalizes to a novel set of faces and nonfaces, as has been shown repeatedly for face-selective responses with fMRI and single-unit neurophysiology. The size and position variation tested (Fig. 4) is a small step in this direction, but the image changes here are quite minor. The selectivities across these small variations could reflect a quite simple selectivity for curvy or round shapes, or a large dark blob above smaller dark blobs (as suggested in Figure 2), which is short of the kind of face selectivity that has been repeatedly demonstrated in brains. The authors could easily test this idea by asking whether the units in their network behave like face-selective cells in monkeys or cortical regions in humans and monkeys. Most straightforwardly, the authors could identify face-selective units as they do here and then measure the response in those units to the stimuli from Tsao et al (2006). This would enable them to directly compare the face selectivity of the units in their network to the reported neurons in monkey face patches, on the same stimuli.
3. Another test of the alternative account that the "face-selective units" found in the untrained network are really just sensitive for some mid-level visual features that were common to the face stimuli used would be to ask if you can find similar selectivities for virtually any category of stimuli

with similar shapes (which is not found in the brain). At a minimum, the authors should test selectivities for other categories in their existing data, but they could also test new categories. If they found similar selectivity for e.g. cars, or chairs, or shoes, that would be consistent with the idea that these selectivities are for mid-level shape features, not for the category – a possibility that could be tested with further stimuli.

4. When does face selectivity emerge in the network? The test for face-selective units in the network indicates that there are more face-selective units in layer 3 and 4 than in layer 5. These layers have overall more units so proportionally it might be similar but what if the authors test in layer 1 and 2? In general it would be helpful to report the proportion of face-selective units compared to all units in a layer, not just the absolute number. Would they also find face-selective units that early and in similar proportions? This would potentially undermine the ecological validity of face selectivity in randomized networks as we would only expect face selectivity to emerge in mid-level processing stages, like in the human visual cortex.

5. The manuscript has a number of mistakes and wrong uses of standard terminology:

a. The study does not as claimed show “face discrimination” or “face recognition” (see title) ability of the face-selective units. In the face literature, “discrimination” or “recognition” means distinguishing one face from another, which is not tested here. What the authors test is rather face detection. The fact that units selected for their face selectivity can discriminate (the same, or only slightly altered) faces from objects is unsurprising (nearly tautological). The authors should use the phrase “face detection” not “face discrimination” if they want to keep these results in the paper.

b. The brain in Figure 1a does not look like a brain (which way is it facing?) and the visual areas are not in the right place.

c. On lines 274-276 it is claimed that Livingstone found face selective responses in 1-month old human infants. Such a result has never been reported in the first place, and Livingstone does not study humans.

d. Units in a network should be referred to as “units” not “neurons” to avoid confusion.

e. Lines 255-256 The claim here is overstated. At most these results show that complex feature selectivity, such as face-selectivity, MIGHT arise innately from the structure of the random feedforward circuitry.

Reviewer #3:

Remarks to the Author:

In this manuscript, Baek, Song, Jang, Kim and Paik delve into an exciting debate about face recognition, specifically whether experience is necessary for it to happen in the primate brain. Although the manuscript is motivated by actual neurons, it focuses exclusively on convolutional neural networks, deep models that can be trained to perform visual classifications with great accuracy. There is growing consensus that these models capture many important features of the primate visual recognition system, and at the very least serve as springboards for insightful discussions about the goals and functions of their biological counterpart. In this regard, the manuscript is broadly interesting and very much of its time.

The principal discovery of the paper treats the best-known convolutional neural network (CNN), AlexNet as its subject. AlexNet can perform visual classification with considerable accuracy because of its training, which allows its hidden units acquire specific weight values that emphasize useful features in photographs. This is analogous to the observation that neurons in the primate brain respond to faces because of the state of their synaptic weights. Here, the authors show that obliterating all traces of training in AlexNet – by randomly shuffling its weights – does not actually remove the presence of some hidden units that still respond to faces. Not only do some of these units continue to respond more to sets of faces than to alternative sets (“bodies,” “scenes,” “objects”), but some units will respond equally well to faces transformed by viewpoint, a robustness to nuisance transformations that interest all investigators of biological and computational vision.

Overall, this key result is perfectly plausible and illuminating about how CNNs actually work. In its

current form, it just seems to have little to do with the debate that motivates the authors. It is missing sensible analyses that could raise its impact beyond that of a report on a curious aspect of *some units* in one artificial neural network. It is possible that additional work will fill these gaps, but the overall product needs a major overhaul.

Major issues.

1) Framing. Although there is a debate about whether face selectivity depends on experience, it is strongly misleading to suggest that the alternative is that this selectivity arises "spontaneously." Proponents of the alternative do not claim that face selectivity is random, they claim that it is *innate*, an adaptation, and they advocate for this view to the point of testing whether face selectivity is present in utero or in blind individuals. Beyond that, this innateness camp seeks to explain why it is that the same brain regions (face "patches") appear to concentrate these face-selective neurons. Thus this paper's use of shuffling does not weigh on the actual debate, and instead raises an alternative that is not considered strongly.

2) Framing (again). Further, the proponents of experience did not write that face selectivity could not arise by chance in neurons of the brain, only that face domains were eliminated by the lack of exposure to faces. This was a statement of expertise and degrees of selectivity, not about single neurons.

Independently of the premise, the work is interesting because it tells us one thing about the types of selectivity that exist in neural networks without training, selectivity that arises from overall architectural properties such as the number of layers, channels, and weight value distributions. This is the premise of a really interesting paper — if the focus expanded beyond this face framing. So now I focus on the actual results and how further clarifications could help the reader understand the manuscript's conceptual advances.

3) What else can shuffled-weight neural networks do? If some hidden units show selectivity for faces, it seems highly unlikely – unbelievable, even - that other hidden units would not show selectivity for well-chosen categories. What is a "well-chosen" category? At least, one comprising images that are more similar to each other than in other categories: for example, faces can be characterized by a roughly upside-down triangle – exactly as illustrated by the authors' PFIs (Fig. 5c). If random wiring led to filters shaped like an "L", then maybe these hidden units would respond selectively to the category of "frames" or "boxes." Thus some necessary analyses to contextualize the random network's "face selectivity" may be as follows:

a) characterize the pixelwise similarity of the stimulus images within each category; is the mean/median similarity of images within the face set statistically indistinguishable from the mean/median similarity in images within each of the other image sets?

b) What are the other filters that arise from this shuffling? If a face-informative filter can arise spontaneously, then other types will be present. This can be demonstrated in many ways, including the manuscript's 2D Gaussian procedure, but there are better, high-resolution ways, including the backpropagation route (as in Deep Dream) and generative adversarial network routes. As the authors use Matlab, where some of these algorithms are already automated, they could do this analysis very quickly. Another reason to complement the manuscript's PFI analysis is that it can only show "blobular patterns" - it is not very expressive.

c) When presented with thousands of images, what kinds of images do other filters seem to prefer? Can this finding be reconciled with the visualized filters?

d) How does shuffling affect the overall performance of AlexNet compared to the trained version? The current result further raises the question that, if functionally useful selective face-detectors appear by chance, and since they are functionally useful for decoding via SVMs, then why do neuronal networks need training at all? What is the performance of shuffled AlexNet at overall classifications relative to the intact version? Are face-selective units in AlexNet more selective or invariant than those in shuffled AlexNet, or is it a matter of number – more of them, but just as good?

4) What are the authors defining as a “neuron”? The work examines face selectivity in Conv5. Conv5 has 256 channels distributed in a $\sim(3 \times 3)$ matrix across the image (Supp. Table 1), yet the manuscript reports that 615 ± 122 “face-selective neurons” were found. Is that number summed across shuffled networks? It probably is not *per network*, unless a neuron is defined by both channel and position ($256 \times 3 \times 3$) would be a mistake, because a given filter is replicated at each position (“weight-sharing,” a defining property of convolutional neural networks). Please provide further clarification.

5) Support vector machines are powerful, and easy to train to classify patterns as long as images from their training set are comparable to those in their test set. Which is why I find it very, very surprising that SVMs showed chance performance when trained to discriminate pictures of faces vs. non-faces using non-face-selective units (Fig. 3c). A similar experiment was performed in vivo (Haxby et al., 2001, Science) and the opposite result was found (decoders could do well without face-selective regions, when trained well). What would be a reason for this discrepancy?

6) Invariance. “The responses of face-selective neurons are invariant to different sizes”. The receptive field of neurons in Conv5 would correspond to about 1/3 of the image, and each of the scaling/translation transformations appear to change images in a range that suitable for these large receptive fields. However, this should be true for all hidden units in the layer *if* the other units are responding to their preferred stimulus. The absence of this comparison is distracting.

b) The rotation invariance is interesting, but again, a pixelwise analysis showing how similar each reference image is to its transformed versions would provide a useful baseline for comparison, and aiding the reader in learning if this is a surprising result.

6) Viewpoint invariance. This is a particularly interesting finding raised by the manuscript. While it is plausible that hidden units could be selective for specific objects based on random selectivity, and invariant to size and position given basic RF properties, it is less clear how *viewpoint* invariance could arise randomly. The first question is, if random wiring projections can give rise to viewpoint invariance, to what else are they giving rise? My concern is that there may be other random wiring phenomena that are just as common yet decidedly non-biological, such a neuron that prefers an object at one size and another object at a different size. Again, the focused reporting of the behavior of *one type* of units in a shuffled network leaves no context to the reader.

a) It would be useful to note where in the Freiwald et al 2010 paper the Fig. 5a tuning curve appears – I couldn’t find it.

In summary, this an interesting manuscript, worth developing, but not supportive of its overall framing, and still largely underdeveloped in its analyses. What I was most intrigued about was that the authors have shown that selectivity for arbitrary image classes can occur in random networks (generalizing beyond faces), which raises the possibility that this is a general property of random computational graphs. The key reveal here might be that since selectivity for arbitrary image classes is a general property of randomly initialized CNNs, neuroscientists need to re-develop their intuitions for computation in networks, and what it means to be “selective.”

Reviewer #4:

Remarks to the Author:

The paper addresses the question of whether face-selectivity of the kind known to exist in face-related cortical regions can arise spontaneously or whether it requires training from visual experience. It reports the results of a computational study of face-selective responses in deep neural networks prior to any training, using only randomly initialized weights.

The main finding of the study is that face-selective neurons can emerge across different conditions of randomly initialized deep neural networks. The activity of these units was sufficient to enable the network to perform discrimination between face and non-face stimuli. The face-preferring neurons showed selectivity to facial features, their face selectivity index was similar to that of

cortical neurons, they were insensitive to low-level visual feature variations such as the translation, rotation, and scaling of face images, and some showed a degree of viewpoint invariance.

Overall, the findings are novel and are well supported by the experiments and results reported in the paper, but there are several comments that need to be addressed.

Regarding relevant previous work with random deep networks, the paper cites two old papers (from 2009), and a recent paper on so-called 'deep priors'. There is more directly relevant recent work, in particular work related to the so-called 'lottery ticket hypothesis'. The original paper is by Frankle & Carbin and there are by now additional papers. The most relevant one is by Ramanujan et al (2020), where they show that randomly weighted neural networks contain sub-networks, which achieve impressive performance without ever modifying the weight values. For example, they show a random sub-net that achieves state-of-the-art performance on ImageNet with no changes to the initial random weights. The results are not the same as the current work, as the previous work treats mainly sub-networks rather than individual units, but in a sense, they are stronger. The previous work also does not make comparisons with the brain, but it shows something strong about the probability of the emergence of functionally meaningful structures in random DNNs.

One of the results listed above was invariance to low-level visual feature variations such as the translation, rotation, and scaling of face images. This conclusion depends on how invariance is defined and measured, and a finer comparison may show that the network units and cortical neurons are different in the response to variations in the input image. A number of recent works have found that small translations or rescaling of the input image can drastically change the predictions of DNNs. (Some review and analysis can be found in Azulay & Weiss 2019). The invariant analysis in the current paper is based on ANOVA, which shows general statistical trends. In contrast, other work tested individual DNN units with small (few pixel) translations, rotations or scale, and found that such changes can make changes in the output, in a way that is not found in cortical units. It will be of interest to check this point with the face-preferring units, to further test their similarity to biological units in terms of local invariance.

One of the findings above reports that face-neurons are selective for facial features, including the eyes, mouth, and facial contour. The images preferred by the units show sensitivity to image features such as blobs and linear features in a face-like configuration, but I think that it will be better to describe the finding in terms of sensitivity to coarse face-like configurations rather than sensitivity to eyes, mouth etc.

The current study also examined the view-point-invariance of the face-selective units, and found both view-specific and view-invariant units, with layer-specificity, where the view-invariant units appearing at a higher level. In the monkey visual system, work on face patches by Freiwald, Tsao and others found that the view-selective units in the AL patch, feeding the view-invariant units in the AM patch, show a characteristic mirror symmetric tuning. It will be interesting to test this characteristic property in the view-variant units feeding the more invariant units at higher levels. The relevance to face-selective regions in primate cortex:

The study shows the emergence of units with preference for faces and face processing functionality (e.g. face detection), arising in DNN with random weights. This is novel and interesting, but perhaps not entirely surprising in view of the findings (following the 'lottery ticket hypothesis') that random DNNs contain sub-networks with complete functionality, e.g. performing object recognition across multiple categories. The findings in the current study are novel in their focus on face preferring units. They add to the findings about the statistics of functional components in random networks, but the possible biological relevance, even in theory is less clear. Given the background above, it becomes of interest to know whether there is something special about faces, or that the same phenomenon holds for other object classes (as appears to happen in the case of random sub-networks performing ImageNet tasks). Another question is whether the emergence of the units described in the study can conceivably seed in the brain something like face regions. This appears necessary to make the findings biologically relevant. Do the authors suggest such a conclusion? If they do, I think that the conclusion should be stated explicitly and discussed, since the gap between a tiny fraction of face-preferring units scattered throughout the network, and the anatomy of the face system is very large. To my view, a satisfactory discussion of the possible relevance of the study to brain regions is necessary to making it biologically relevant.

Response to Reviewers

Thank you for all of the effort you have devoted to reviewing our manuscript. We deeply appreciate your constructive, timely comments. Based on the questions and comments provided, we made appropriate revisions with regard to the issues raised by reviewers. We reworked the manuscript, both the text and the figures, and now provide additional data analyses and simulations. We are confident that our revision fully addresses the issues raised previously and provides further validation of our results.

Here, we provide a reply to your concerns to describe in detail how each of the issues you mention was addressed. Please find our responses to each of the questions below. Thank you very much for your kind consideration.

Reviewer #1:

This seems to me an interesting and theoretically very relevant paper. I have some suggestions for improvements, however.

Q1. One crucial finding in several species (and in human newborns as well, e.g. Buiatti et al. 2019) is the inversion effect, the lack of preference (or the difficulty in recognition) of upside-down faces. It seems that the authors' network does not show such an effect and this needs to be discussed.

We appreciate this timely suggestion. Indeed, we found that our model network also shows the "inversion effect" as observed in monkeys (Buiatti et al., 2019; Perrett et al., 1985; Tsao et al., 2006). We included this result in the revised manuscript.

To investigate whether the face-selective units observed in our untrained networks show weaker responses to inverted face images, we generated 200 inverted face images by flipping the original images upside-down (**R1Q1 a, Supplementary Fig. 9a**). We found that the responses of face-selective units to inverted face images are significantly lower than those to upright faces (**R1Q1 b, Supplementary Fig. 9b**, Mann-Whitney U test, $n = 200$, $**p = 7.8 \times 10^{-13}$), whereas these responses are still higher than those to non-face images (Mann-Whitney U test, $*p = 4.8 \times 10^{-7}$). These results indicate that our face-selective units show similar responses to upright and inverted faces, i.e. the "inversion effect", as observed in the monkey IT (Tsao et al., 2006) (**R1Q1 c, Supplementary Fig. 9c**, Mann-Whitney U test, $n = 16$, $**p = 2.6 \times 10^{-22}$, $*p = 5.2 \times 10^{-3}$). We added this result in lines 230-235 in the revised manuscript.

Revised text (Results, lines 230-235)

Notably, we found that our model units also show the "inversion effect" observed in monkeys (Buiatti et al., 2019; Perrett et al., 1985; Tsao et al., 2006). We found that the responses of face-selective units to inverted face images are significantly lower than those to upright faces (**Supplementary Fig. 9a, b**, Mann-Whitney U test, $n = 200$, $**p = 7.8 \times 10^{-13}$), whereas these responses are still higher than those to non-face images (Mann-Whitney U test, $*p = 4.8 \times 10^{-7}$). These results indicate that our face-selective units show similar responses to upright and inverted faces, i.e. the "inversion effect", as observed in the monkey IT¹³ (**Supplementary Fig. 9c**, Mann-Whitney U test, $n = 16$, $**p = 2.6 \times 10^{-22}$, $*p = 5.2 \times 10^{-3}$).

Q2. In order to check for the lack of face selectivity in face-deprived monkeys it would be interesting trying to train the network on a set of stimuli without faces to see whether the face-selective neurons would then disappear.

Thank you for this interesting comment. After additional simulations and analysis following this suggestion, we found that the face-selectivity of individual face units decreases when the network is trained with face-deprived images, whereas the selectivity increases when the network is trained with images including a face.

Figure R1Q2 (Fig. 6)

To investigate the effect of training on a face-deprived image set, we prepared the following three different stimulus sets: (1) a face-deprived ImageNet (images including faces excluded), (2) the original ImageNet, and (3) the original ImageNet with added face images used in the current study (R1Q2 a, left, Fig. 6a, top). Then, the network was trained with each of these image sets (R1Q2 a, right, Fig. 6a, bottom).

First, we found that the face-selectivity index (FSI) of the face-selective units was significantly decreased after being trained to the face-deprived image set (R1Q2 b, Fig. 6b, Mann-Whitney U test, $n_{\text{Untrained}} = 4,267$, $n_{\text{Deprived}} = 2,452$, $***p = 3.5 \times 10^{-28}$), whereas it was increased after being trained to the face-including image sets (Mann-Whitney U test, $n_{\text{Face}} = 3,585$, $**p = 5.2 \times 10^{-4}$). Notably, the FSI was significantly decreased after being trained to the original ImageNet dataset that contains images of faces but has no group labeled as "face" (R1Q2 b, Fig. 6b, Mann-Whitney U test, $n_{\text{Original}} = 3,561$, $*p = 9.3 \times 10^{-45}$). This suggests that the tuning of face-selective units could either be sharpened or weakened by training with distinct stimulus sets.

Next, we found that the number of face-selective units observed was greater in the network trained to face-including image sets compared to that trained to face-deprived images (R1Q2 c, Fig. 6c, Mann-Whitney U test, N_{Net}

= 10, $*p < 1.3 \times 10^{-3}$). Interestingly, however, we found that the number of face-selective units, when trained to face-including images, appeared to be smaller than that of untrained networks. These results imply that the training process of the network to face-including images selectively sharpens the tuning of face units so that the selectivity of strongly tuned units is sharpened while the weakly tuned units are pruned. In this condition, the face detection performance of the networks would improve in face-trained networks even if the number of face units decreased compared to the initial, untrained condition.

To validate this scenario, we trained the SVM using the response of face-selective units for a face detection task in an untrained network and in the three networks trained to each type of data set. As predicted, we found that the face detection performance was significantly increased in the networks trained to the face-including image set compared to that of the untrained network (**R1Q2 d, Fig. 6d**, Mann-Whitney U test, $n_{\text{trial}} = 1,000$, $**p = 1.3 \times 10^{-3}$), whereas the face detection performance of the network trained to the face-deprive image set was significantly decreased compared to the untrained network (**R1Q2 d, Fig. 6d**, Mann-Whitney U test, $n_{\text{trial}} = 1,000$, $***p = 2.1 \times 10^{-4}$). Taken together, these results imply that the face-selectivity and the face detection performance can be refined differently depending on stimuli given during the training process. This result is consistent with the previous observation of decreased face-selectivity in face-deprived monkeys (Arcaro et al., 2017).

Revised text (Result, lines 317-348)

We tested a scenario in which our current model can corroborate the conflicting observations regarding the role of visual experience for the development of face-selectivity. A previous report suggested that visual experience is necessary for the emergence of face-selectivity by showing that monkeys raised without exposure to faces lack face-selective domains (Arcaro et al., 2017). On the other hand, another recent study showed that the face-selective area develops robustly in congenitally blind humans, suggesting that visual experience is not necessary for face-selectivity^{28,29}. Regarding these conflicting results, we examined how face-selective units in untrained networks can be affected by training with visual inputs.

To investigate the effect of visual experience under different conditions, we prepared three distinct stimuli sets: (1) a face-deprived ImageNet (images including faces excluded), (2) the original ImageNet, and (3) the original ImageNet with added face images used in the current study (**Fig. 6a**). Then, the network was trained with each of these image sets. First, we found that the face-selectivity index (FSI) of the face-selective units was significantly decreased after being trained to the face-deprived image set (**Fig. 6b**, Mann-Whitney U test, $n_{\text{Untrained}} = 4,267$, $n_{\text{Deprived}} = 2,452$, $***p = 3.5 \times 10^{-28}$), whereas it was increased after being trained to the face-including image sets (Mann-Whitney U test, $n_{\text{Face}} = 3,585$, $**p = 5.2 \times 10^{-4}$). Notably, the FSI was significantly decreased after being trained to the original ImageNet dataset that contains images of faces but has no group labeled as “face” (**6b**, Mann-Whitney U test, $n_{\text{Original}} = 3,561$, $*p = 9.3 \times 10^{-45}$). This suggests that the tuning of face-selective units could either be sharpened or weakened by training with distinct stimulus sets.

Next, we found that the number of face-selective units observed was greater in the network trained to face-including image sets compared to that trained to face-deprived images (**Fig. 6c**, Mann-Whitney U test, $N_{\text{Net}} = 10$,

$*p < 1.3 \times 10^{-3}$). Interestingly, however, we found that the number of face-selective units, when trained to face-including images, appeared to be smaller than that of untrained networks. These results imply that the training process of the network to face-including images selectively sharpens the tuning of face units so that the selectivity of strongly tuned units is sharpened while the weakly tuned units are pruned. In this condition, the face detection performance of the networks would improve in face-trained networks even if the number of face units decreased compared to the initial, untrained condition. To validate this scenario, we trained the SVM using the response of face-selective units for a face detection task in an untrained network and in the three networks trained to each type of data set. As predicted, we found that the face detection performance was significantly increased in the networks trained to the face-including image set compared to that of the untrained network (**Fig. 6d**, Mann-Whitney U test, $n_{\text{trial}} = 1,000$, $**p = 1.3 \times 10^{-3}$), whereas the face detection performance of the network trained to the face-deprive image set was significantly decreased compared to the untrained network (**Fig. 6d**, Mann-Whitney U test, $n_{\text{trial}} = 1,000$, $*p = 2.1 \times 10^{-4}$). This result is consistent with the previous observation of decreased face-selectivity in face-deprived monkeys (Arcaro et al., 2017).

Q3. The Introduction is too primate-centric in my view, and convey to the generic reader the impression that faces are special only for primates, which is not correct (e.g. Rosa Salva et al (2011). The Evolution of Social Orienting: Evidence from Chicks (*Gallus gallus*) and Human Newborns. PLoS ONE, 6(4): e18802; Rosa Salva et al (2010). Faces are special for chicks: Evidence for inborn domain-specific mechanisms underlying spontaneous preferences for face-like stimuli. *Developmental Science*, 13: 565-577; Versace et al. (2020) Early preference for face-like stimuli in solitary species as revealed by tortoise hatchlings. *Proceedings of the National Academy of Sciences Sep 2020*, 117 (39) 24047-24049; DOI: 10.1073/pnas.2011453117. Note that these comparative studies provide accurate controls for the role of experience to a degree which would be not possible for e.g. human newborns. The issue is important because neural networks should, at the end, be able to reproduce the biological phenomena in their largest generality.

We appreciate this helpful comment. In the revised manuscript, we added citations of previous studies (Rosa-Salva et al., 2010; Salva et al., 2011; Versace et al., 2020) regarding face-selectivity in various species to avoid misleading readers with the impression that faces are special only for primates.

Revised text (Introduction, lines 19-22)

The ability to identify and recognize faces is a crucial function for social behavior, and this ability is thought to originate from neuronal tuning at the single or multi-neuronal level¹⁻²⁰. Neurons that selectively respond to faces (face-selective neurons) are observed in various species (Rosa-Salva et al., 2010; Salva et al., 2011; Versace et al., 2020), and they have been considered as the building blocks of face detection⁸.

Reviewer #2

This manuscript uses deep net modeling to ask whether face selective responses can arise spontaneously in a system that has never been trained on visual input. The question is of fundamental importance, the general idea of the study clever, and the writing is admirably engaging and clear. It is further commendable that the authors have placed their code in an open accessible location. I do however have some nontrivial concerns which render the evidence not yet very convincing to me, which perhaps the authors could address in a revision.

Q1. As far as I can tell, the entire analysis in Fig. 3 is double-dipping: the same data are used to train and test an SVM that have been used to select the units entering this analysis in the first place. This analysis is famously invalid (see e.g. Kriegeskorte, et al, Nature neuroscience, 2009) and this analysis should be removed from the paper.

We apologize for the incomplete description. Indeed, the data for training and testing shown in Fig. 3 (and all of the other analyses in the manuscript) were completely different sets such that they did not share a single image in common. Specifically, 260 images were prepared for each class. Then, 200 images were randomly sampled from each class and used for face unit selection. Among the 60 remaining images in each class, 40 images were used for the training of the SVM and the other 20 images used for testing. Thus, this is not a double-dipping condition (Kriegeskorte et al., 2009), as stated in the comment. In the revised manuscript, we added a detailed description of this method to avoid confusion.

Revised text (Methods, lines 468-481)

A face vs non-face detection task was established to investigate whether face-selective units could perform basic face perception. To determine if this was possible, an SVM was trained with network responses to images and was then made to predict whether a class of unseen images was or was not a face. To avoid the double-dipping issue (Kriegeskorte et al., 2009), we undertook the training and testing of the SVM using distinct sets of images, where 260 images were prepared for each class and 200 images were then randomly sampled from those images and used for face unit selection. Among the 60 remaining images in each class, 40 images were used for the training of the SVM and the other 20 images were used for testing. The label of the training set was then changed to a binary class: face or non-face. For each trial, the SVM was trained with the relationship between the fifth layer's response to the training set and the new training label. After a training session, the model predicted the test label using the network response to the test set. Furthermore, to test whether the responses of face-selective units could detect faces with varied in different ways, we trained a SVM with network responses for center-view face images (N = 40) and non-face images (N = 40). The model then predicted test labels using the responses of face-selective units to new test sets which consisted of face and non-face images with different types of variation, in this case different sizes, positions and rotation angles.

Q2. The finding of putatively face-selective responses is intriguing but preliminary at this point, because the authors do not show that the claimed face selectivity generalizes to a novel set of faces and nonfaces, as has been shown repeatedly for face-selective responses with fMRI and single-unit neurophysiology. The size and position variation tested (Fig. 4) is a small step in this direction, but the image changes here are quite minor. The selectivities across these small variations could reflect a quite simple selectivity for curvy or round shapes, or a large dark blob above smaller dark blobs (as suggested in Figure 2), which is short of the kind of face selectivity that has been repeatedly demonstrated in brains. The authors could easily test this idea by asking whether the units in their network behave like face-selective cells in monkeys or cortical regions in humans and monkeys. Most straightforwardly, the authors could identify face-selective units as they do here and then measure the response in those units to the stimuli from Tsao et al (2006). This would enable them to directly compare the face selectivity of the units in their network to the reported neurons in monkey face patches, on the same stimuli.

We appreciate this important comment. In the revised manuscript, we used four different sets of novel face and non-face images including the stimulus set used in Tsao *et al.* (2006) and confirmed that the face-selective units observed in untrained networks behave as face-selective cells in monkeys, showing consistent face tuning regardless of the choice of the stimulus set.

Figure R2Q2 (Supplementary Fig. 3, Fig. 1)

To test whether the face-selective units in the untrained network show selective responses to general face images, we investigated the responses of face-selective units to four different novel stimulus sets that were not used to find face-selective units. These were (1) 50 face images from our original data set, not used for finding the face-selective units; (2) 16 face images used in Tsao *et al.* (2006, 2010); (3) 50 face images used in Cao *et al.* (2018); and (4) 50 face images artificially generated by the FaceGen simulator (singular inversions) in color and grayscale. We found that face-selective units in the untrained network show significantly higher responses to novel face images compared to the responses to non-face images under all conditions (**R2Q2 a, Supplementary Fig. 3**, Mann-Whitney U test, $*p < 7.3 \times 10^{-24}$). In addition, we found that the face-selectivity index (FSI) measured using the same stimulus set used in Tsao *et al.* (2006) was comparable to the index associated with the monkey IT neurons (Tsao *et al.*, 2006), which was significantly higher compared to the values from the control measured from a shuffled response (**R2Q2 b, Fig. 1e**, Mann-Whitney U test, $n_{\text{monkey}} = 158$, $n_{\text{untrained}} = 539$, n.s., $p = 0.08$, $*p = 9.7$

$\times 10^{-28}$). These results suggest that innate face-selective units in an untrained network are selective to general face images and behave as face-selective cells in monkeys (Tsao et al., 2006). We added these results to the revised manuscript, at lines 93-100 and lines 109-117.

Revised text (Result, lines 93-100)

Surprisingly, we observed a group of “face-selective” units ($n = 288 \pm 75$ in 100 random networks, mean \pm s.d.) that show significantly higher responses to face images than to non-face images emerges in the untrained networks (**Fig. 1d**). We also found that the observed face-selective units in the untrained networks show a face-selectivity index (FSI)^{13,15} comparable to the index associated with the monkey IT neurons¹⁵ and a significantly higher value than those measured from a shuffled response (**Fig. 1e**, Mann–Whitney U test, $n_{\text{monkey}} = 158$, $n_{\text{untrained}} = 539$, n.s., $p = 0.08$, $n_{\text{untrained}} = 539$, $n_{\text{shuffled}} = 539$, $*p = 9.7 \times 10^{-28}$) for various definitions of the FSI^{7,13,50} (**Supplementary Fig. 2**, n.s., $p > 0.2$, $*p < 4.5 \times 10^{-4}$). These results suggest that face-selective units, highly tuned to the face as observed in the brain, can emerge in DNNs even in the complete absence of learning.

Revised text (Result, lines 109-117)

Next, we investigated the responses of face-selective units to four different novel stimulus sets that were not used to find face-selective units. These were (1) 50 face images from our original data set, not used for finding the face-selective units; (2) 16 face images used in Tsao *et al.* (2006, 2010)^{13,15}; (3) 50 face images used in Cao *et al.* (2018)⁵¹; and (4) 50 face images artificially generated by the FaceGen simulator (singular inversions) in color and grayscale (**Supplementary Fig. 3**). We found that face-selective units in the untrained network show significantly higher responses to novel face images compared to the responses to non-face images under all conditions (**Supplementary Fig. 3**, Mann-Whitney U test, $*p < 7.3 \times 10^{-24}$). These results suggest that innate face-selective units in an untrained network are selective to general face images and behave as face-selective cells in monkeys¹³.

Q3. Another test of the alternative account that the “face-selective units” found in the untrained network are really just sensitive for some mid-level visual features that were common to the face stimuli used would be to ask if you can find similar selectivities for virtually any category of stimuli with similar shapes (which is not found in the brain). At a minimum, the authors should test selectivities for other categories in their existing data, but they could also test new categories. If they found similar selectivity for e.g. cars, or chairs, or shoes, that would be consistent with the idea that these selectivities are for mid-level shape features, not for the category – a possibility that could be tested with further stimuli.

Thank you for the helpful comment. To address this issue, we investigated the responses of the face-selective units to a mid-level feature of the face using two types of control images in which global face features are disrupted but mid-level face features are preserved. These were (1) “scrambled” faces, in which small patches of the local face components were spatially scrambled, and (2) “texform” faces (Long et al., 2018), in which global face features are disrupted but the statistics of the local face texture is preserved. We confirmed that face-selective units show significantly higher responses to the original face images compared to the corresponding control images. In addition,

the responses of face units to these control images were not greater than those to other non-face images, implying that face-selective units are selective to the global context of faces instead of the mid-level components.

Figure R2Q3 (Fig. 1)

First, we tested whether the face-selective units are sensitive to local facial features such as the eyes, nose and mouth by comparing their responses to “faces” and scrambled “controls.” To generate the scrambled control faces, we divided face images into local patches of a size (28 pixels) that contains local facial features. Then, these local patches were randomly permuted while the convex hull of the face is retained (**R2Q3 a, Fig. 1h, left**). We found that face-selective units show significantly low responses to scrambled faces compared to those to the original face images (**R2Q3 b, Fig. 1h, right**, Mann-Whitney U test, $n = 200$, $*p = 1.1 \times 10^{-51}$). In addition, we also found that the responses of face units to a “scrambled” face were not greater than those to non-face images (Mann-Whitney U test, $n = 200$, n.s., $p = 1$).

Next, the “texform” face images were generated using a texture synthesis algorithm (Long et al., 2018) (**R2Q3 c, Fig. 1h, left**). This method captures the texture and coarse form information from the original face but scrambles its global structure so that it would not be recognized as a face. Accordingly, a “texform” face lacks certain critical high-level categorical information of the original face. We found that the responses of face-selective units to a “texform” face were significantly lower than those to face images (**R2Q3 d, Fig. 1h, right**, Mann-Whitney U test, $n_{\text{face}} = 200$, $n_{\text{texform}} = 100$, $*p = 4.8 \times 10^{-33}$) and that they were comparable with those to non-face images (Mann-Whitney U test, $n_{\text{texform}} = 100$, $n_{\text{non-face}} = 200$, n.s., $p = 0.77$). These results suggest that face-selectivity in an untrained network does not apply to the mid-level visual features of faces.

As the reviewer suggested, we also investigated whether there are units selective to other non-face classes, and this was done by measuring the responses of the untrained network to whole ImageNet classes. We observed that there are indeed units selective to other classes (**R2Q3 e, e.g., Jack-o'-lantern and Gazania**).

Interestingly, we found that these selective units are observed only for 39 classes among 1,000 classes of images (**R2Q3 f**). It is important to note that selective units must have been observed in most of the classes if they emerged from the preference to mid-level features. Thus, this result implies that category selectivity in an untrained network may not be simply described as sensitivity to a mid-level visual feature. Subsequently, we further examined the characteristics of these units selective to other non-face classes.

We found that the units selective to other classes also show selectivity to high-level categorical features rather than to mid-level visual features, similar to the face-selective units (**R2Q3 g**). For example, the responses of units selective to “Jack-o'-lantern” became significantly lower when the original images were scrambled or texformed (**R2Q3 h**, left, Mann-Whitney U test, $n = 100$, $*p < 6.0 \times 10^{-12}$) and were comparable with those of other non-selective classes (Mann-Whitney U test, $n = 100$, n.s., $p > 0.26$). Similarly, the responses of selective units to “Gazania” were significantly lower to scrambled/texform images than they were to the original images (**R2Q3 h**, right, Mann-Whitney U test, $n = 100$, $*p < 2.8 \times 10^{-22}$) as well as those to other non-selective classes (Mann-Whitney U test, $n = 100$, n.s., $p > 0.89$). Taken together, our results suggest that the observed selective units in the untrained network are not particularly selective to mid-level features but are instead selective to a high-level context of the images. In the revised manuscript, we discuss this issue in lines 118-131.

Revised text (Result, lines 118-131)

One possible scenario is that the observed face-selective units are simply sensitive for mid-level visual features common to face images. To investigate this possibility, we measured the responses of the face-selective units to a

mid-level feature of the face using two types of control images in which global face features are disrupted but mid-level face features are preserved. These were (1) “scrambled” faces, in which small patches of the local face components were spatially scrambled, and (2) “texform” faces⁵², in which global face features are disrupted but the statistics of the local face texture is preserved (**Fig. 1h**, left). We confirmed that face-selective units show significantly higher responses to the original face images compared to the corresponding control images (**Fig. 1h**, right; Mann-Whitney U test, Face vs Scrambled face, $n = 200$, $*p = 1.1 \times 10^{-51}$; Face vs Texform face, $n_{\text{face}} = 200$, $n_{\text{texform}} = 100$, $*p = 4.8 \times 10^{-33}$). In addition, the responses of face units to these control images were not greater than those to other non-face images, implying that face-selective units are selective to the global context of faces instead of the mid-level components (**Fig. 1h**, right; Mann-Whitney U test, Scrambled face vs Non-face, $n = 200$, n.s., $p = 1$; Texform face vs Non-face, $n_{\text{texform}} = 100$, $n_{\text{non-face}} = 200$, n.s., $p = 0.77$). These results suggest that the observed face units in the untrained network are not particularly selective to mid-level features but are instead selective to a high-level context of the images.

Q4. When does face selectivity emerge in the network? The test for face-selective units in the network indicates that there are more face-selective units in layer 3 and 4 than in layer 5. These layers have overall more units so proportionally it might be similar but what if the authors test in layer 1 and 2? In general, it would be helpful to report the proportion of face-selective units compared to all units in a layer, not just the absolute number. Would they also find face-selective units that early and in similar proportions? This would potentially undermine the ecological validity of face selectivity in randomized networks as we would only expect face selectivity to emerge in mid-level processing stages, like in the human visual cortex.

Thank you for this helpful comment. In the additional analysis, we investigated the layer-specific emergence of face-selective units in the hierarchical structure of an untrained network. We found that the number of observed face-selective units is highest in the mid- and high-level layers, similar to observations in the ventral visual pathway of monkeys (Livingstone et al., 2017; Tsao et al., 2006). Specifically, we found that face-selective units are hardly observed in layer 1. In layer 2, face-selective units start to arise but the number of these units was noticeably smaller compared to those observed in layers 3, 4, and 5 (**R2Q4, Supplementary Fig. 12**, Mann-Whitney U test, $N_{\text{Net}} = 100$, $*p < 7.2 \times 10^{-11}$). These results suggest that the development of face-selectivity requires a hierarchical structure of the network along with random feedforward weights, which enables multiple linear-nonlinear computations. In the revised manuscript, we added this result at lines 294-302.

Revised text (Result, lines 294-302)

We also investigated the layer-specific emergence of face-selective units in untrained networks. We found that face-selective units are also observed in earlier layers, Conv2 to 3 but are scarcely found in Conv1 (**Supplementary Fig.**

Figure R2Q4 (Supplementary Fig. 12)

12, Conv1: $n = 0.008 \pm 0.002\%$, Conv2: $n = 0.476 \pm 0.095\%$, Conv3: $n = 0.738 \pm 0.133\%$, Conv4: $n = 0.698 \pm 0.137\%$, Conv5: $n = 0.667 \pm 0.172\%$). Notably, the number of face-selective units observed in Conv2 was significantly smaller than those observed in Conv3, Conv4, and Conv5 (**Supplementary Fig. 12**, Mann-Whitney U test, $N_{\text{Net}} = 100$, $*p < 7.2 \times 10^{-11}$). Thus, the number of observed face-selective units was highest in the mid- and high-level layers, similar to observations in the ventral visual pathway of monkeys^{13,14}. These results suggest that the development of face-selectivity requires a hierarchical structure of the network along with random feedforward weights, which enables multiple linear-nonlinear computations.

Q5. The manuscript has a number of mistakes and wrong uses of standard terminology:

We appreciate your detailed comments. As suggested, we made corrections throughout the text and figures, as follows:

a. The study does not as claimed show “face discrimination” or “face recognition” (see title) ability of the face-selective units. In the face literature, “discrimination” or “recognition” means distinguishing one face from another, which is not tested here. What the authors test is rather face detection. The fact that units selected for their face selectivity can discriminate (the same, or only slightly altered) faces from objects is unsurprising (nearly tautological). The authors should use the phrase “face detection” not “face discrimination” if they want to keep these results in the paper.

Thanks for this correction. We replaced the term “face discrimination” with “face detection” throughout our manuscript.

Revised text (Title)

Face Detection in Untrained Deep Neural Networks

Revised text (Introduction, lines 20-22)

Neurons that selectively respond to faces (face-selective neurons) are observed in various species^{21–23}, and they have been considered as the building blocks of face detection⁸.

Revised text (Result, lines 175-177)

First, we compared the detection performance of the SVM using a single unit randomly sampled from face-selective units and using units without selective responses to any image classes.

Revised text (Methods, lines 468-469)

A face vs non-face detection task was established to investigate whether face-selective units could perform basic face perception.

b. The brain in Figure 1a does not look like a brain (which way is it facing?) and the visual areas are not in the right place.

We regret this confusing illustration. We modified the illustration of the brain in Figure 1a as shown below.

c. On lines 274-276 it is claimed that Livingstone found face selective responses in 1-month old human infants. Such a result has never been reported in the first place, and Livingstone does not study humans.

We apologize for this mistake in the description of the reference. In the revised manuscript, we corrected the term "human infants." It is now "infant monkeys."

Revised text (Discussion, lines 402-404)

Livingstone *et al.* show that neurons broadly tuned to faces are already observed in infant monkeys (~1 month old) and that the region where these neurons are observed is identical to where the face-neurons of adult monkeys are observed.

d. Units in a network should be referred to as "units" not "neurons" to avoid confusion.

We replaced the term "neuron" with "unit" throughout the revised manuscript.

e. Lines 255-256 The claim here is overstated. At most these results show that complex feature selectivity, such as face-selectivity, MIGHT arise innately from the structure of the random feedforward circuitry.

As suggested, we toned down our arguments to avoid over-speculation in the revised manuscript.

Revised text (Discussion, lines 374-376)

They also imply that complex types of feature selectivity, such as face-selectivity, may arise innately from the structure of the random feedforward circuitry.

Reviewer #3

In this manuscript, Baek, Song, Jang, Kim and Paik delve into an exciting debate about face recognition, specifically whether experience is necessary for it to happen in the primate brain. Although the manuscript is motivated by actual neurons, it focuses exclusively on convolutional neural networks, deep models that can be trained to perform visual classifications with great accuracy. There is growing consensus that these models capture many important features of the primate visual recognition system, and at the very least serve as springboards for insightful discussions about the goals and functions of their biological counterpart. In this regard, the manuscript is broadly interesting and very much of its time.

The principal discovery of the paper treats the best-known convolutional neural network (CNN), AlexNet as its subject. AlexNet can perform visual classification with considerable accuracy because of its training, which allows its hidden units acquire specific weight values that emphasize useful features in photographs. This is analogous to the observation that neurons in the primate brain respond to faces because of the state of their synaptic weights. Here, the authors show that obliterating all traces of training in AlexNet – by randomly shuffling its weights – does not actually remove the presence of some hidden units that still respond to faces. Not only do some of these units continue to respond more to sets of faces than to alternative sets (“bodies,” “scenes,” “objects”), but some units will respond equally well to faces transformed by viewpoint, a robustness to nuisance transformations that interest all investigators of biological and computational vision.

Overall, this key result is perfectly plausible and illuminating about how CNNs actually work. In its current form, it just seems to have little to do with the debate that motivates the authors. It is missing sensible analyses that could raise its impact beyond that of a report on a curious aspect of *some units* in one artificial neural network. It is possible that additional work will fill these gaps, but the overall product needs a major overhaul.

We sincerely appreciate your helpful comments and detailed suggestions throughout your review of our manuscript. Based on your questions and the advice provided, we revised our manuscript thoroughly as a major overhaul. In the revised manuscript, new simulations and analysis, as well as additional data analysis have been performed extensively, and we reworked both the text and the figures. Particularly, we provide a significant amount of new simulation and analysis results to address two important issues raised:

1. Expanding the discourse for investigating innate selectivity for various objects

Specifically, we performed new simulations to investigate whether there are units selective to other non-face classes, and this was done by measuring the responses of the untrained network to whole ImageNet classes. We observed that there are indeed units selective to other classes but only for 39 classes among 1,000 classes of images. The observed selective units in the untrained network were not particularly selective to mid-level features but were found instead to be selective to a high-level context of the images, similarly to face-selective units (see our responses to Question 2-a, 3-b,c for details). However, from additional analysis of latent space representations, we found that face tuning is distinctive from tunings to other object classes: the network responses to face images present stronger clustering compared to those of other tested objects, probably because face images have statistics more readily distinguishable from the statistics of other objects due to the stronger clustering of the abstracted responses in the DNN. Thus, our

results suggest that innate face-selectivity is more likely to emerge in the brain compared to other object classes due to the simple geometric features of faces. Please refer to our responses to the questions below for details.

2. We clarified our main question and the aim of the study — “How does innate face-selectivity arise before visual experience?”

We thoroughly reframed our text so that it focuses on the observation of innate visual functions in untrained networks. We demonstrate how such innate selectivity can emerge and how it is refined by visual experience. In addition, we show that our current model can also explain the observation of “face patches” in previous experimental studies. Please refer to our responses to Questions 1, 2 for details.

Q1. Framing. Although there is a debate about whether face selectivity depends on experience, it is strongly misleading to suggest that the alternative is that this selectivity arises “spontaneously.” Proponents of the alternative do not claim that face selectivity is random, they claim that it is *innate*, an adaptation, and they advocate for this view to the point of testing whether face selectivity is present in utero or in blind individuals. Beyond that, this innateness camp seeks to explain why it is that the same brain regions (face “patches”) appear to concentrate these face-selective neurons. Thus, this paper’s use of shuffling does not weigh on the actual debate, and instead raises an alternative that is not considered strongly.

We appreciate this timely comment. We found that our obscure descriptions in the introduction, results, and discussion sections in the previous manuscript were potentially misleading. Indeed, the sentence “face-selectivity is innate” summarizes exactly what we intended to claim in the current study. We regret that the term “spontaneously” was so emphasized previously that it caused unnecessary confusion. In the revised manuscript, we thoroughly reframed our text so that it focuses on the observation of innate visual functions in untrained networks and their characteristics, as well as discussions of the mechanisms underlying the emergence of object selectivity from the random feedforward connections. Details of our additional simulations and analyses are described in the responses to the reviewers and in the revised manuscript.

Figure R3Q1 (Supplementary Fig. 12, 13)

Firstly, we re-described our main question and the aim of the study in the introduction — “How does innate face-selectivity arise before visual experience?” Subsequently, we demonstrated how such innate selectivity may be refined by visual experience, providing a possible explanation of previous observations of experience-dependent selectivity in animal data (Arcaro et al., 2017). In the revised manuscript, we discuss this issue in lines 22-73.

In addition, we showed that our current model can also provide an explanation of the observation of “face patches,” in which face-selective neurons appear to concentrate both hierarchically and retinotopically (Arcaro and Livingstone, 2017; Arcaro et al., 2020; Bell et al., 2009). In brief, we found that the development of innate face-selectivity requires a hierarchical structure of the network along with random feedforward weights and thus emerges mostly in the mid- and high-level processing stages of untrained networks and in the central visual field, similar to observations in the ventral visual pathway of monkeys (Livingstone et al., 2017; Tsao et al., 2006).

Previous studies of primates reported that the face-selective units are observed mostly in the mid- and high-level hierarchy of the ventral visual stream (Livingstone et al., 2017; Tsao et al., 2006) and that they are also spatially clustered in the central vision area in retinotopy, as a “face patch” (Arcaro and Livingstone, 2017; Arcaro et al., 2020; Bell et al., 2009). Because the exact anatomical location of the brain correspondent to the units in the convolutional neural network could not be determined, we investigated instead the distribution of face-selective units in the inter-layer level and the retinotopic location of each face-selective unit in each layer. We found that the number of observed face-selective units is largest in the mid- and high-level layer (**R3Q1 a, Supplementary Fig. 12**) similar to observations in the ventral visual pathway of monkeys (Livingstone et al., 2017; Tsao et al., 2006). To investigate the retinotopic distribution of face-selective units, we examined the locations of the receptive field center position of each face-selective and object-selective unit. We backtracked the receptive field of each face-selective unit (**R3Q1 b, Supplementary Fig. 13a**) and computed the distances between the center position of the input image and the position of the receptive field. As a result, we found that the spatial encoding of face-selective units is more biased to the center of the visual field compared to units selective to other object classes (**R3Q1 c, Supplementary Fig. 13b**) throughout the entire network hierarchy (**R3Q1 d, Supplementary Fig. 13c**). This result is consistent with the previous observation in monkeys that face-selective neurons are clustered retinotopically in the center-vision area (Arcaro and Livingstone, 2017; Arcaro et al., 2020; Bell et al., 2009). Taken together, these findings show that face-selective units generated in untrained networks have biological characteristics similar to those observed in monkeys, not only in single units but also at the population and inter-layer levels.

Revised text (Introduction, lines 22-73)

Previous studies have suggested a scenario in which visual experience develops face-selective neurons^{24–26}. The experience-dependent characteristics of face-selective neurons imply that visual experience plays a critical role in developing face-selectivity in the brain. A study using functional magnetic resonance imaging reported that the category of selective neuronal activity observed depends greatly on the experience of the subject in its lifetime²⁴. It was also observed that the preferred feature images of face-selective neurons in adult monkeys are those that resemble animals or familiar people depending on individual experiences²⁵. Another study of the inferior temporal cortex (IT) in monkeys reported that robust tuning of face-selective neurons is not observed until one year after birth¹⁴ and that face-selectivity relies on experience during the early infant years. It was also reported that monkeys raised without face exposure did not develop normal face-selective domains²⁶. These results suggest that face-selective neurons develop based on training in the form of visual experience.

However, another view suggests that face-selectivity can innately arise without visual experience^{27–34}.

Although visual experience is critical for refining the development of face-selective neurons, several lines of research have demonstrated that primitive face-selectivity is observed even before visual experience^{27–31}. Primate infants behaviorally prefer to look at face-like objects as opposed to non-face objects^{32–34}, implying that face-encoding units may already exist in infants. Moreover, visual category-selective domains, including the face, are observed in the ventral visual stream in adult humans who have been blind since birth^{28,29}. Furthermore, a recent study reported that face-selective neurons are observed in infant animals and that the spatial organization of such early face-selective regions appeared similar to that observed in adults¹⁴. These results taken together imply that face-selective neurons can arise before visual experience, in contradiction to the first scenario. These contradictory results most likely stem from limitations regarding the control of the experimental conditions, as it is impossible to control the amount of visual experience for a particular category, such as face, in individual subjects. Even if the subjects are visually deprived such that they are prevented from having visual experience, the portion of category-selective neurons and their degree of tuning may vary across subjects and cannot easily be predicted. These various factors make it difficult to investigate the developmental mechanism of face-selective neurons in the brain.

A model study using biologically inspired artificial neural networks, such as deep neural networks (DNNs)^{35,36}, may offer an effective approach to the problem in this case^{37–40}. Recently, DNNs, a stack of biologically inspired feedforward projections with a linear-nonlinear neural motif, have provided insight into the underlying mechanisms of brain functions, particularly with regard to the development of various functions for visual perception^{37,38,41}. For example, a recent model study reported that the neural response of the monkey IT cortex could not only be predicted by the responses of DNNs trained to natural images^{37,38} but could also be controlled by the preferred feature image generated by the DNN model⁴². Notably, previous studies using random hierarchical networks provide important clues about the origin of innate face-selectivity in untrained neural networks. It was reported that untrained feedforward networks can initiate various cognitive functions with random weights and that a random network can perform image classification tasks in that way as well^{43–46}. It was also reported that a randomly initialized convolutional neural network could reconstruct corrupted images without any training, which implies that a random network can provide a priori information about the low-level statistics in natural images⁴⁷. Overall, such observations suggest the possibility of the emergence of innate cognitive functions, such as primitive face-selectivity in untrained, random hierarchical networks. However, the details of how this innate function emerges in untrained neural networks are not yet understood.

Herein, we show that face-selective units (model neurons) can arise in completely untrained hierarchical neural networks. Using AlexNet³⁵, a model reproducing the structure of the ventral stream of the visual cortex, we found that face-selectivity can emerge robustly across different conditions of randomly initialized deep neural networks and that these selective activities enable the network to perform face detection. The preferred feature images obtained from the generative adversarial network and the reverse-correlation method show that face-selective units are selective for a face-like configuration, distinct from units with no selectivity. We also found that their face-selectivity indices (FSI) are comparable to those observed with face-selective neurons in the brain. Intriguingly, the observed face-selective responses were found to be invariant to low-level visual feature variations such as the translation, rotation, and scaling of face images. These neuronal units also showed the layer-specific

view-point-invariance characteristics observed in monkeys, and the degree of viewpoint invariance increased along the network hierarchy in the untrained networks, as observed in the brain. Our findings suggest that early face-selectivity can innately arise without visual experience, originating from the random initial wiring of feedforward projections in hierarchical visual pathways.

Revised text (Result, lines 294-315)

We also investigated the layer-specific emergence of face-selective units in untrained networks. We found that face-selective units are also observed in earlier layers, Conv2 to 3 but are scarcely found in Conv1 (**Supplementary Fig. 12**, Conv1: $n = 0.008 \pm 0.002\%$, Conv2: $n = 0.476 \pm 0.095\%$, Conv3: $n = 0.738 \pm 0.133\%$, Conv4: $n = 0.698 \pm 0.137\%$, Conv5: $n = 0.667 \pm 0.172\%$). Notably, the number of face-selective units observed in Conv2 was significantly smaller than those observed in Conv3, Conv4, and Conv5 (**Supplementary Fig. 12**, Mann-Whitney U test, $N_{\text{Net}} = 100$, $*p < 7.2 \times 10^{-11}$). Thus, the number of observed face-selective units was highest in the mid- and high-level layers, similar to observations in the ventral visual pathway of monkeys^{13,14}. These results suggest that the development of face-selectivity requires a hierarchical structure of the network along with random feedforward weights, which enables multiple linear-nonlinear computations.

Previous studies of primates reported that the face-selective units are also spatially clustered in the central vision area in retinotopy, as a “face patch”^{62–64}. Because the exact anatomical location of the brain correspondent to the units in the convolutional neural network could not be determined, we investigated instead the retinotopic distribution of face-selective units in each layer — we examined the locations of the receptive field center position of each face-selective and object-selective unit. We backtracked the receptive field of each face-selective unit (**Supplementary Fig. 13a**) and computed the distances between the center position of the input image and the position of the receptive field. As a result, we found that the spatial encoding of face-selective units is more biased to the center of the visual field compared to units selective to other object classes (**Supplementary Fig. 13b**, Mann–Whitney U test, $N_{\text{Net}} = 100$, $*p = 3.9 \times 10^{-8}$) throughout the entire network hierarchy (**Supplementary Fig. 13c**). This result is consistent with the previous observation in monkeys that face-selective neurons are clustered retinotopically in the center-vision area^{62–64}. Taken together, these findings show that face-selective units generated in untrained networks have biological characteristics similar to those observed in monkeys, not only in single units but also at the population and inter-layer levels.

Q2. Framing (again). Further, the proponents of experience did not write that face selectivity could not arise by chance in neurons of the brain, only that face domains were eliminated by the lack of exposure to faces. This was a statement of expertise and degrees of selectivity, not about single neurons.

Thank you for pointing out this important issue. We agree that our descriptions of the previous manuscript were potentially misleading. In the revised manuscript, we reframed our text so that it clearly describes the statement of the previous experiment, that the face-selectivity in the brain could be modulated by visual experience. In addition, we confirmed that previous observations regarding visual experience (loss of face-selectivity due to the lack of

exposure to faces) are readily reproduced by our model. After additional simulations and analysis following this suggestion, we found that the face-selectivity of individual units decreases when the network is trained with face-deprived images, whereas the selectivity increases when the network is trained with images that include a face. These results are consistent with previous observations in monkeys, implying that our model of innate face detection can be complemented by additional consideration of the role of early visual experience. Eventually, this provides a more complete scenario of the development of face-selectivity in the brain.

Figure R3Q2 (Fig. 6)

To investigate the effect of visual training with or without exposure to face images, we prepared the following three different stimulus sets: (1) a face-deprived ImageNet (images including faces excluded), (2) the original ImageNet, and (3) the original ImageNet with added face images used in the current study (**R3Q2 a**, left, **Fig. 6a**, top). Then, the network was trained with each of these image sets (**R3Q2 a**, right, **Fig. 6a**, bottom). We observed that the trained networks in three conditions showed noticeable differences statistically (**R3Q2 b**, **Fig. 6b**).

We found that the face-selectivity index (FSI) of the face-selective units was significantly decreased after being trained to the face-deprived image set (**R3Q2 b**, **Fig. 6b**, Mann-Whitney U test, $n_{\text{Untrained}} = 4,267$, $n_{\text{Deprived}} = 2,452$, $***p = 3.5 \times 10^{-28}$), whereas it was increased after being trained to the face-including image sets (Mann-Whitney U test, $n_{\text{Face}} = 3,585$, $**p = 5.2 \times 10^{-4}$). Notably, the FSI was significantly decreased after being trained to the original ImageNet dataset that contains images of faces but has no group labeled as "face" (**R3Q2 b**, **Fig. 6b**, Mann-Whitney U test, $n_{\text{Original}} = 3,561$, $*p = 9.3 \times 10^{-45}$). This suggests that the tuning of face-selective units could be either sharpened or weakened by training with distinct stimulus sets. Taken together, these results imply that the face-selectivity and the face detection performance can be refined differently depending on the stimuli given during the training process. This result is consistent with a previous observation of decreased face-selectivity in face-deprived monkeys (Arcaro et al., 2017).

Revised text (Introduction, lines 22-73)

Previous studies have suggested a scenario in which visual experience develops face-selective neurons^{24–26}. The experience-dependent characteristics of face-selective neurons imply that visual experience plays a critical role in developing face-selectivity in the brain. A study using functional magnetic resonance imaging reported that the category of selective neuronal activity observed depends greatly on the experience of the subject in its lifetime²⁴. It was also observed that the preferred feature images of face-selective neurons in adult monkeys are those that

resemble animals or familiar people depending on individual experiences²⁵. Another study of the inferior temporal cortex (IT) in monkeys reported that robust tuning of face-selective neurons is not observed until one year after birth¹⁴ and that face-selectivity relies on experience during the early infant years. It was also reported that monkeys raised without face exposure did not develop normal face-selective domains²⁶. These results suggest that face-selective neurons develop based on training in the form of visual experience.

However, another view suggests that face-selectivity can innately arise without visual experience^{27–34}. Although visual experience is critical for refining the development of face-selective neurons, several lines of research have demonstrated that primitive face-selectivity is observed even before visual experience^{27–31}. Primate infants behaviorally prefer to look at face-like objects as opposed to non-face objects^{32–34}, implying that face-encoding units may already exist in infants. Moreover, visual category-selective domains, including the face, are observed in the ventral visual stream in adult humans who have been blind since birth^{28,29}. Furthermore, a recent study reported that face-selective neurons are observed in infant animals and that the spatial organization of such early face-selective regions appeared similar to that observed in adults¹⁴. These results taken together imply that face-selective neurons can arise before visual experience, in contradiction to the first scenario. These contradictory results most likely stem from limitations regarding the control of the experimental conditions, as it is impossible to control the amount of visual experience for a particular category, such as face, in individual subjects. Even if the subjects are visually deprived such that they are prevented from having visual experience, the portion of category-selective neurons and their degree of tuning may vary across subjects and cannot easily be predicted. These various factors make it difficult to investigate the developmental mechanism of face-selective neurons in the brain.

A model study using biologically inspired artificial neural networks, such as deep neural networks (DNNs)^{35,36}, may offer an effective approach to the problem in this case^{37–40}. Recently, DNNs, a stack of biologically inspired feedforward projections with a linear-nonlinear neural motif, have provided insight into the underlying mechanisms of brain functions, particularly with regard to the development of various functions for visual perception^{37,38,41}. For example, a recent model study reported that the neural response of the monkey IT cortex could not only be predicted by the responses of DNNs trained to natural images^{37,38} but could also be controlled by the preferred feature image generated by the DNN model⁴². Notably, previous studies using random hierarchical networks provide important clues about the origin of innate face-selectivity in untrained neural networks. It was reported that untrained feedforward networks can initiate various cognitive functions with random weights and that a random network can perform image classification tasks in that way as well^{43–46}. It was also reported that a randomly initialized convolutional neural network could reconstruct corrupted images without any training, which implies that a random network can provide a priori information about the low-level statistics in natural images⁴⁷. Overall, such observations suggest the possibility of the emergence of innate cognitive functions, such as primitive face-selectivity in untrained, random hierarchical networks. However, the details of how this innate function emerges in untrained neural networks are not yet understood.

Herein, we show that face-selective units (model neurons) can arise in completely untrained hierarchical neural networks. Using AlexNet³⁵, a model reproducing the structure of the ventral stream of the visual cortex, we found that face-selectivity can emerge robustly across different conditions of randomly initialized deep neural

networks and that these selective activities enable the network to perform face detection. The preferred feature images obtained from the generative adversarial network and the reverse-correlation method show that face-selective units are selective for a face-like configuration, distinct from units with no selectivity. We also found that their face-selectivity indices (FSI) are comparable to those observed with face-selective neurons in the brain. Intriguingly, the observed face-selective responses were found to be invariant to low-level visual feature variations such as the translation, rotation, and scaling of face images. These neuronal units also showed the layer-specific view-point-invariance characteristics observed in monkeys, and the degree of viewpoint invariance increased along the network hierarchy in the untrained networks, as observed in the brain. Our findings suggest that early face-selectivity can innately arise without visual experience, originating from the random initial wiring of feedforward projections in hierarchical visual pathways.

Revised text (Results, lines 317-333)

We tested a scenario in which our current model can corroborate the conflicting observations regarding the role of visual experience for the development of face-selectivity. A previous report suggested that visual experience is necessary for the emergence of face-selectivity by showing that monkeys raised without exposure to faces lack face-selective domains (Arcaro et al., 2017). On the other hand, another recent study showed that the face-selective area develops robustly in congenitally blind humans, suggesting that visual experience is not necessary for face-selectivity^{28,29}. Regarding these conflicting results, we examined how face-selective units in untrained networks can be affected by training with visual inputs.

To investigate the effect of visual experience under different conditions, we prepared three distinct stimuli sets: (1) a face-deprived ImageNet (images including faces excluded), (2) the original ImageNet, and (3) the original ImageNet with added face images used in the current study (**Fig. 6a**). Then, the network was trained with each of these image sets. First, we found that the face-selectivity index (FSI) of the face-selective units was significantly decreased after being trained to the face-deprived image set (**Fig. 6b**, Mann-Whitney U test, $n_{\text{Untrained}} = 4,267$, $n_{\text{Deprived}} = 2,452$, $***p = 3.5 \times 10^{-28}$), whereas it was increased after being trained to the face-including image sets (Mann-Whitney U test, $n_{\text{Face}} = 3,585$, $**p = 5.2 \times 10^{-4}$). Notably, the FSI was significantly decreased after being trained to the original ImageNet dataset that contains images of faces but has no group labeled as “face” (**6b**, Mann-Whitney U test, $n_{\text{Original}} = 3,561$, $*p = 9.3 \times 10^{-45}$). This suggests that the tuning of face-selective units could either be sharpened or weakened by training with distinct stimulus sets.

Q2-a. Independently of the premise, the work is interesting because it tells us one thing about the types of selectivity that exist in neural networks without training, selectivity that arises from overall architectural properties such as the number of layers, channels, and weight value distributions. This is the premise of a really interesting paper — if the focus expanded beyond this face framing. So now I focus on the actual results and how further clarifications could help the reader understand the manuscript's conceptual advances.

We deeply appreciate your constructive comments. As suggested, we devoted efforts to expand our discourse beyond simple reports of face tuning. For example, we performed new simulations to investigate the emergence of selectivity to general object classes, finding that there are indeed units selective to other classes. From an additional analysis of latent space representations, we also found that face tuning is distinctive from tunings to other object classes. Because face images have statistics more readily distinguishable from statistics in the latent space compared to that of other objects, the network responses to face images present stronger clustering compared to those of other tested objects. Thus, our results suggest that innate face-selectivity is more likely to emerge in the brain compared to other object classes due to the simple geometric features of faces. This result also suggests a theoretical explanation of how faces-selectivity is more readily observed than selectivity to other objects (Tsao et al., 2003).

Figure R3Q2-1 (Supplementary Fig. 4-6)

To investigate tunings to other object classes in untrained networks, we examined the responses of the network to 1,000 classes of the ImageNet (**R3Q2-1 a, Supplementary Fig. 6a**). We observed that there are indeed units selective to other classes (**R3Q2-1 b, Supplementary Fig. 6b**) but that these selective units are observed in only for 39 classes. Subsequently, we examined in greater detail the characteristics of these units selective to other non-face classes and found that the units selective to other classes also show selectivity to higher level categorical features rather than to mid-level visual features, similar to the face-selective units (see our responses to Reviewer 2, Question 3, and to Reviewer 4, Question 3 and 5, for more details of this analysis).

In the above result, one can argue that the untrained network may have selectivity to particular object classes such as faces, in which the intra-class image similarity is higher than those of the other classes. To investigate this possibility, we examined the units selective to each of the five object classes that have statistically comparable levels of intra-class image similarity (see our responses to Question 3-a for more detailed statistics). We initially observed that the preferred feature images (PFIs) of face-selective units present the face-like configurations in both the reverse correlation method (Bonin et al., 2011) and X-Dream with the generative adversarial network (Ponce et al., 2019), whereas the PFIs of units selective to non-face classes show only barely noticeable configurations of each object class (**R3Q2-1 c, Supplementary Fig. 4**). This result may explain why neurons selective to every object class are not readily observed in the brain, despite the fact that most objects can be decoded by the population response of IT (Tsao et al., 2003).

From this result, we hypothesized that faces can induce stronger clustering in latent representations than those of other object classes in untrained networks. To validate our hypothesis, we used the dimension-reduction method (Maaten and Hinton, 2008) (tSNE) to compare a clustered representation of each object class in terms of the raw pixel values and in the responses of Conv5. For a quantification of the representational clustering of each class, we measured the Silhouette index (Kaufman and Rousseeuw, 2009) to estimate the consistency of data clustering. The Silhouette index S_i for the i^{th} point is defined as $c_i = (b_i - a_i) / \max(a_i, b_i)$, where a_i and b_i refer to the intra-class distance and the inter-class distance for the i^{th} point, respectively.

We found that the Silhouette index is all negative when the raw pixel values of each image are represented in the tSNE space (**R3Q2-1 d, Supplementary Fig. 5a**). However, for the Conv5 layer responses, the Silhouette index of face responses showed positive values, whereas those of all of the other classes showed significantly lower negative values (**R3Q2-1 e, Supplementary Fig. 5b**). This result demonstrates that face images have statistics readily distinguishable from the statistics of other objects, leading to the strong clustering of abstracted responses in the DNN. Such a special characteristic may be understood by considering the fact that faces have a simple configuration of geometric components, a combination of only several ovals or dots. Thus, innate face-selectivity is more likely to emerge in the brain compared to other object classes due to the simple geometric features of faces. We added this result in lines 140-170 in the revised manuscript.

Revised text (Results, lines 140-170)

As a result, using both methods, we obtained the PFI of face-selective units, units selective to non-face classes, and units without selective responses to any image classes (**Fig. 2b, d, Supplementary Fig. 4**). We found that the

PFI of the face-selective units presents distinguishable features from those of other objects (**Fig. 2b, d, Supplementary Fig. 4**). We observed that the PFIs of face-selective units represent face-like configurations, whereas the PFIs of units selective to non-face classes show only barely noticeable configurations of each object class (only except for a couple of cases, i.e., flowers in a reverse correlation and hands in X-Dream).

From this result, we hypothesized that faces can induce stronger clustering in latent space representation than those of other object classes in untrained networks. To validate our hypothesis, we used the dimension-reduction method⁵⁵ (tSNE) to compare a clustered representation of each object class in terms of the raw pixel values and in the responses of Conv5. For a quantification of the representational clustering of each class, we measured the Silhouette index⁵⁶ to estimate the consistency of data clustering. We found that the Silhouette index is all negative when the raw pixel values of each image are represented in the tSNE space (**Supplementary Fig. 5a**). However, for the Conv5 layer responses, the Silhouette index of face responses showed positive values, whereas those of all of the other classes showed significantly lower negative values (**Supplementary Fig. 5b**).

Similar results were also observed in a further analysis using extended image classes containing 1,000 classes of the ImageNet dataset (**Supplementary Fig. 6a**). Among the 1,000 classes, we found that selective units are observed in only 39 classes (**Supplementary Fig. 6b**). This result is consistent with a previous observation that selective neurons are not observed for all object classes in the inferior temporal cortex (IT), although most of the objects could be decoded by the population response of IT⁵⁷. We selected the top-5 and bottom-5 classes of the number of selective units observed among those 39 classes and computed the PFI using the reverse correlation method. We found that the PFIs of the top-5 group showed a noticeable configuration of each preferred object class (**Supplementary Fig. 6c**), whereas those of the bottom-5 classes represented a barely noticeable shape of the PFI (**Supplementary Fig. 6d**). Furthermore, an additional analysis using the dimension reduction method (tSNE) revealed that there is a strong correlation between the Silhouette index in the Conv5 latent space and the number of selective units observed (**Supplementary Fig. 6e-g**). This result demonstrates that face images have statistics readily distinguishable from the statistics of other objects, leading to the strong clustering of abstracted responses in the DNN. Such a special characteristic can be understood by considering the fact that faces have a simple configuration of geometric components, a combination of only several ovals or dots. Thus, innate face-selectivity is more likely to emerge in the brain compared to other object classes due to the simple geometric features of faces. This result may explain why neurons selective to every object class are not readily observed in the brain.

Revised text (Method, lines 460-466)

To visualize the network responses to the low-level feature-controlled stimulus set, the t-distributed stochastic neighbor embedding (t-SNE) method was used for dimension reduction⁵⁵. By minimizing the difference between the original and low-dimensional distributions of neighbor distances, a 2D representation of the responses of the fifth convolutional layer was obtained. To quantify the level of clustering of each class, we measured the Silhouette index (SI)⁵⁶ to estimate the consistency of data clustering. The Silhouette index SI for the i^{th} point is defined as $c_i = (b_i - a_i) / \max(a_i, b_i)$, where a_i and b_i refer to the intra-class distance and the inter-class distance for the i^{th} point, respectively. The same analysis was also performed for the stimulus set that added the face class to ILSVRC2010.

Q3. So now I focus on the actual results and how further clarifications could help the reader understand the manuscript's conceptual advances. What else can shuffled-weight neural networks do? If some hidden units show selectivity for faces, it seems highly unlikely – unbelievable, even - that other hidden units would not show selectivity for well-chosen categories. What is a “well-chosen” category? At least, one comprising images that are more similar to each other than in other categories: for example, faces can be characterized by a roughly upside-down triangle – exactly as illustrated by the authors’ PFIs (Fig. 5c). If random wiring led to filters shaped like an “L”, then maybe these hidden units would respond selectively to the category of “frames” or “boxes.” Thus, some necessary analyses to contextualize the random network’s “face selectivity” may be as follows:

a) Characterize the pixel-wise similarity of the stimulus images within each category; is the mean/median similarity of images within the face set statistically indistinguishable from the mean/median similarity in images within each of the other image sets?

As suggested, we investigated the intra- and inter-class pixel-wise similarity among the images used in the current study (**R3Q3-1 a, b, Supplementary Fig. 1g, h**). As expected, we found that the pixel-wise similarity of face images is statistically indistinguishable from those of other images classes, as our stimulus images are already “similarity-controlled;” i.e., low-level visual features of these images, such as the luminance and contrast, were controlled to minimize the inter-class difference. Using this image set, again we confirmed that face-selective units emerge consistently in untrained networks (**R3Q3-1 c, Fig. 1b-d**). This result suggests that the observed face-selectivity is not due to the higher image similarity within the face class. Details of the analysis are as follows:

Figure R3Q3-1 (Supplementary Fig. 1, Fig. 1)

First, we measured the intra-class image similarity of each class as the pixel-wise correlation between every pair of images within the class. We confirmed that the intra-class similarity of the face images is not significantly different from those images of other classes (**R3Q3-1 a, Supplementary Fig. 1g**, two-tailed t-test, $n =$

19,900, n.s., $p > 0.33$). It should also be noted that all of these classes show a mean intra-class pixel-wise correlation close to zero, as their low-level visual features are “similarity-controlled” (R3Q3-1 a, Supplementary Fig. 1g, two-tailed t-test, $n = 19,900$, n.s., $p > 0.24$).

Next, we found that the intra-class similarity of face images is not significantly different from the inter-class similarities between the faces and the other classes (R3Q3-1 b, Supplementary Fig. 1h, two-tailed t-test, $n_{\text{face}} = 19,900$, $n_{\text{non-face}} = 20,100$, n.s., $p > 0.18$). Subsequently, we double-checked to ensure that face-selective units are observed consistently in an untrained network with such similarity-controlled stimulus sets (R3Q3-1 c, Fig. 1d). These results show that observation of the face-selective units in our untrained network is not due to the pixel-wise similarity of the stimulus images.

Furthermore, with additional control simulations, we confirmed that the emergence of face-selective units is not simply due to the effects of low-level visual features. It must be noted that the current stimulus set was initially designed to control low-level features, as in a previous experimental study (Stigliani et al., 2015) — specifically the luminance, contrast, size, and positions of objects. The distribution of the pixel luminance of each object and the background was controlled in all cases to have the same normal distribution (mean = 0.5, SD = 0.125) (R3Q3-1 d, Supplementary Fig. 1b, two-tailed t-test, $n = 200$, n.s. $p = 1$). From the controlled luminance distribution of each image, all of the images have the same RMS contrast level of 0.25 (R3Q3-1 e, Supplementary Fig. 1c, two-tailed t-test, $n = 200$, n.s. $p = 1$). In addition, the size of each object was controlled so that it has the same mean height and width with constant variation following a normal distribution (mean = 0.7, SD = 0.04 of visual field) (R3Q3-1 f, Supplementary Fig. 1d, two-tailed t-test, $n = 200$, n.s., $p > 0.57$). Similarly, the position of the object center was

controlled so that the mean location is at the center of the visual field with the positional noise following a normal distribution (mean = 0, SD = 0.04 of visual field) (**R3Q3-1 g, Supplementary Fig. 1e**, two-tailed t-test, $n = 200$, n.s., $p > 0.34$, **R3Q3-1 h, Supplementary Fig. 1f**, two-tailed t-test, $n = 200$, n.s. $p > 0.37$). Our observation of face-selective units in an untrained network with such a “low-level feature-controlled” stimuli set suggests that the face-selective units we observed are not due to the statistical differences of low-level visual features nor to any image similarity within and across the classes. In the revised manuscript, we replaced our old analyses with these new results with a new controlled stimulus set and described the results in detail.

Revised text (Result, lines 88-92)

The stimulus set consisted of grayscale images in six different categories (**Fig. 1c**), specifically the face, a scrambled face, and four non-face objects, as previously done in monkey experiments⁴⁸. The images in each class were designed to control the low-level features of the luminance, contrast, object size, and object location, and they have statistically comparable intra- and inter-class image levels of similarity (**Supplementary Fig. 1**).

b) What are the other filters that arise from this shuffling? If a face-informative filter can arise spontaneously, then other types will be present. This can be demonstrated in many ways, including the manuscript's 2D Gaussian procedure, but there are better, high-resolution ways, including the backpropagation route (as in Deep Dream) and generative adversarial network routes. As the authors use Matlab, where some of these algorithms are already automated, they could do this analysis very quickly. Another reason to complement the manuscript's PFI analysis is that it can only show "blobular patterns" - it is not very expressive.

We appreciate this comment. It must be noted that we already showed that some units are selective to other object classes (see Question 2-a for details). Here, as suggested, we performed an additional analysis using a generative adversarial network (X-Dream, Ponce et al., 2019) to examine the preferred feature image (PFI) of face-selective units and the units selective to other non-face classes. We found that the PFI of face units showed a face-shape configuration similar to those generated by the reverse correlation method, but those of units selective to other classes showed scarcely noticeable visual configurations. To understand this result, we further examined the representation of each object class in the latent space using the dimension-reduction method (tSNE). We found that the network responses to face images present much stronger clustering compared to those of other tested objects. This result provides a possible explanation of why faces can be readily distinguishable from other objects in untrained networks and of why the PFI of face-selective units shows a readily recognizable average configuration.

To compare the observed face tuning with tunings to other object classes in an untrained network, we generated the PFIs of units selective to faces and to other object classes (**R3Q3-2 a, Supplementary Fig. 4**, hand, flower, horn, chair). Using the reverse correlation method and a generative adversarial network (X-Dream, Ponce et al., 2019), we initially observed that the PFIs of face-selective units represent the face-like configurations in both the reverse correlation method and X-Dream, whereas the PFIs of units selective to non-face classes show only barely noticeable configurations of each object class (only except for a couple of cases, i.e., flowers in the reverse correlation and hands in X-Dream). This result may explain why neurons selective to every object class are not

readily observed in the brain, despite the fact that most objects can be decoded by the population response of IT (Tsao et al., 2003). From this result, we hypothesized that faces can induce stronger clustering in latent representations than those of other object classes in untrained networks. To validate our hypothesis, the t-distributed stochastic neighbor embedding (t-SNE) method was used for dimension reduction (Maaten and Hinton, 2008). By minimizing the difference between the original and low-dimensional distributions of neighbor distances, a 2D representation of the responses of the fifth convolutional layer was obtained. To quantify the level of clustering of each class, we measured the Silhouette index (SI) (Kaufman and Rousseeuw, 2009) to estimate the consistency of data clustering. The Silhouette index SI for the i^{th} point is defined as $c_i = (b_i - a_i) / \max(a_i, b_i)$, where a_i and b_i refer to the intra-class distance and the inter-class distance for the i^{th} point, respectively.

Figure R3Q3-2 (Supplementary Fig. 4, 5)

We found that the Silhouette index is all negative when the raw pixel values of each image are represented in the tSNE space (R3Q3-2 b, Supplementary Fig. 5a). However, for the Conv5 layer responses, the Silhouette index of face responses showed positive values, whereas those of all of the other classes showed significantly lower negative values (R3Q3-2 c, Supplementary Fig. 5b). This result demonstrates that face images have statistics readily distinguishable from the statistics of other objects, leading to the strong clustering of abstracted responses in the DNN. Such a special characteristic may be understood by considering the fact that faces have a simple configuration of geometric components, a combination of only several ovals or dots. Thus, innate face-selectivity is more likely to emerge in the brain compared to other object classes due to the simple geometric features of faces. We added this result in lines 140-154 in the revised manuscript.

Revised text (Results, lines 140-154)

As a result, using both methods, we obtained the PFI of face-selective units, units selective to non-face classes, and units without selective responses to any image classes (**Fig. 2b, d, Supplementary Fig. 4**). We found that the PFI of the face-selective units presents distinguishable features from those of other objects (**Fig. 2b, d, Supplementary Fig. 4**). We observed that the PFIs of face-selective units represent face-like configurations, whereas the PFIs of units selective to non-face classes show only barely noticeable configurations of each object class (only except for a couple of cases, i.e., flowers in a reverse correlation and hands in X-Dream).

From this result, we hypothesized that faces can induce stronger clustering in latent space representation than those of other object classes in untrained networks. To validate our hypothesis, we used the dimension-reduction method⁵⁵ (tSNE) to compare a clustered representation of each object class in terms of the raw pixel values and in the responses of Conv5. For a quantification of the representational clustering of each class, we measured the Silhouette index⁵⁶ to estimate the consistency of data clustering. We found that the Silhouette index is all negative when the raw pixel values of each image are represented in the tSNE space (**Supplementary Fig. 5a**). However, for the Conv5 layer responses, the Silhouette index of face responses showed positive values, whereas those of all of the other classes showed significantly lower negative values (**Supplementary Fig. 5b**).

Revised text (Method, lines 460-466)

To visualize the network responses to the low-level feature-controlled stimulus set, the t-distributed stochastic neighbor embedding (t-SNE) method was used for dimension reduction⁵⁵. By minimizing the difference between the original and low-dimensional distributions of neighbor distances, a 2D representation of the responses of the fifth convolutional layer was obtained. To quantify the level of clustering of each class, we measured the Silhouette index (SI)⁵⁶ to estimate the consistency of data clustering. The Silhouette index SI for the i^{th} point is defined as $c_i = (b_i - a_i) / \max(a_i, b_i)$, where a_i and b_i refer to the intra-class distance and the inter-class distance for the i^{th} point, respectively. The same analysis was also performed for the stimulus set that added the face class to ILSVRC2010.

c) When presented with thousands of images, what kinds of images do other filters seem to prefer? Can this finding be reconciled with the visualized filters?

We investigated whether there are units selective to other non-face classes, and this was done by measuring the responses of the untrained network to all ImageNet classes and to our face stimulus images (**R3Q3-3 a, Supplementary Fig. 6a**). We observed that there are indeed units selective to other classes (**R3Q3-3 b-d, Supplementary Fig. 6b-d**). Notably, among the 1,000 classes, we found that these selective units are observed in only for 39 classes (**R3Q3-3 b, Supplementary Fig. 6b**). This result is consistent with a previous observation that selective neurons are not observed for all object classes in the inferior temporal cortex (IT), although most of the objects could be decoded by the population response of IT (Tsao et al., 2003). We selected the top-5 and bottom-5 classes of the number of selective units observed among those 39 classes and computed the PFI using the reverse correlation method. We found that the PFIs of the top-5 group showed a noticeable configuration of each

preferred object class, whereas those of the bottom-5 classes represented a barely noticeable shape of the PFI (R3Q3-3 c, d, Supplementary Fig. 6c, d). Furthermore, an additional analysis using the dimension reduction method (tSNE) revealed that there is a strong correlation between the Silhouette index in the Conv5 latent space and the number of selective units observed (R3Q3-3 e-g, Supplementary Fig. 6e-g). We observed that the network responses to images of the top-ranked classes (in observed selective unit number) present stronger clustering compared to those of other tested objects (R3Q3-3 e, f, Supplementary Fig. 6e, f). This result may provide a possible explanation of why some object classes are readily distinguishable from other objects in untrained networks and readily generate units selective to that class.

Figure R3Q3-3 (Supplementary Fig. 6)

As a detailed method, the t-distributed stochastic neighbor embedding (t-SNE) method was used for dimension reduction (Maaten and Hinton, 2008) to visualize the network responses to the low-level feature-controlled stimulus set. By minimizing the difference between the original and low-dimensional distributions of neighbor distances, a 2D representation of the responses of Conv5 was obtained. To quantify the level of clustering of each class, we measured the silhouette index (SI, Kaufman and Rousseeuw, 2009) to estimate the consistency of data clustering. See our revised method for more details.

Revised text (Result, lines 155-170)

Similar results were also observed in a further analysis using extended image classes containing 1,000 classes of the ImageNet dataset (**Supplementary Fig. 6a**). Among the 1,000 classes, we found that selective units are observed in only 39 classes (**Supplementary Fig. 6b**). This result is consistent with a previous observation that selective neurons are not observed for all object classes in the inferior temporal cortex (IT), although most of the objects could be decoded by the population response of IT⁵⁷. We selected the top-5 and bottom-5 classes of the number of selective units observed among those 39 classes and computed the PFI using the reverse correlation method. We found that the PFIs of the top-5 group showed a noticeable configuration of each preferred object class (**Supplementary Fig. 6c**), whereas those of the bottom-5 classes represented a barely noticeable shape of the PFI (**Supplementary Fig. 6d**). Furthermore, an additional analysis using the dimension reduction method (tSNE) revealed that there is a strong correlation between the Silhouette index in the Conv5 latent space and the number of selective units observed (**Supplementary Fig. 6e-g**). This result demonstrates that face images have statistics readily distinguishable from the statistics of other objects, leading to the strong clustering of abstracted responses in the DNN. Such a special characteristic can be understood by considering the fact that faces have a simple configuration of geometric components, a combination of only several ovals or dots. Thus, innate face-selectivity is more likely to emerge in the brain compared to other object classes due to the simple geometric features of faces. This result may explain why neurons selective to every object class are not readily observed in the brain.

d) How does shuffling affect the overall performance of AlexNet compared to the trained version? The current result further raises the question that, if functionally useful selective face-detectors appear by chance, and since they are functionally useful for decoding via SVMs, then why do neuronal networks need training at all? What is the performance of shuffled AlexNet at overall classifications relative to the intact version? Are face-selective units in AlexNet more selective or invariant than those in shuffled AlexNet, or is it a matter of number – more of them, but just as good?

This is very important issue that we studied intensively in the revised manuscript. After additional simulations and an analysis, we found that both face-selectivity of the individual units and the face detection performance of the population are increased by training with face images, as well as the effective range of invariance for the spatial shift. In contrast, the face-selectivity of individual units decreases when the network is trained with face-deprived images. We also found that the number of face-selective units changes when the network was trained with images with or without a face. In general, these results imply that the face-selective units in an untrained network can be a blueprint of mature selective units and that the initial tuning of face-selective units could either be sharpened or weakened by training with distinct stimulus sets, as observed in the monkey IT (Arcaro et al., 2017).

To investigate the effect of visual training with or without exposure to face images, we prepared the following three different stimulus sets: (1) a face-deprived ImageNet (images with those including faces excluded), (2) the original ImageNet, and (3) the original ImageNet with added face images used in the current study (**R3Q3-4 a**, top, **Fig. 6a**, top). Then, the network was trained with each of these image sets (**R3Q3-4 a**, bottom, **Fig. 6a**, bottom).

First, we found that face-selectivity indices (FSI) were significantly increased after being trained to natural image set including face class (**R3Q3-4 b, Fig. 6b**, Mann-Whitney U test $n_{\text{Untrained}} = 4,267$, $n_{\text{Face}} = 3,585$, $**p = 5.2 \times 10^{-4}$), whereas, it was decreased after being trained to the face deprived image set (**R3Q3-4 b, Fig. 6b**, Mann-Whitney U test, $n_{\text{Deprived}} = 2,452$, $***p = 3.5 \times 10^{-28}$). Notably, the FSI was significantly decreased after being trained to the original ImageNet dataset that contains images of faces but has no group labeled as “face” (**R3Q3-4 b, Fig. 6b**, Mann-Whitney U test, $n_{\text{Original}} = 3,561$, $*p = 9.3 \times 10^{-45}$). This suggests that the tuning of face-selective units could be either sharpened or weakened by training with distinct stimulus sets.

Figure R3Q3-4 (Fig. 6)

Second, we found that the number of face-selective units observed was greater in the network trained to face-including image sets compared to that trained to face-deprived images (**R3Q3-4 c, Fig. 6c**, Mann-Whitney U test, $N_{\text{Net}} = 10$, $*p < 1.3 \times 10^{-3}$). Interestingly, however, we found that the number of face-selective units, when trained to face-including images, appeared to be smaller than that of untrained networks. These results imply that the training process of the network to face-including images selectively sharpens the tuning of face units so that the selectivity of strongly tuned units is sharpened while units that are weakly tuned are pruned. In this condition, the face detection performance of the networks would improve in face-trained networks even if the number of face units decreased compared to the initial, untrained condition.

To validate this scenario, we trained the SVM using the response of face-selective units for a face detection task in an untrained network and in the three networks trained to each type of data set. As predicted, we found that the face detection performance was significantly increased in the networks trained to the face-including image set compared to that of the untrained network (**R3Q3-4 d, Fig. 6d**, Mann-Whitney U test, $n_{\text{trial}} = 1,000$, $**p = 1.3 \times 10^{-3}$),

whereas the face detection performance of the network trained to the face-deprive image set was significantly decreased compared to the untrained network (**R3Q3-4 d**, **Fig. 6d**, Mann-Whitney U test, $n_{\text{trial}} = 1,000$, $*p = 2.1 \times 10^{-4}$). Taken together, these results imply that the face-selectivity and the face detection performance can be refined differently depending on stimuli given during the training process. This result is consistent with the previous observation of decreased face-selectivity in face-deprived monkeys (Arcaro et al., 2017).

Next, we investigated the effect of visual training on the observed invariance of face-selective responses to low-level feature variation. We examined the effective range of invariance for a spatial translation and the scaling of the stimulus in untrained networks (**R3Q3-4 e**, black, **Fig. 6e**, black) and networks trained to face images (**R3Q3-4 e**, blue, **Fig. 6e**, blue). We found that the effective range of invariant responses appeared to be significantly wider in the trained networks compared to the untrained ones for both translation and scaling variation (**R3Q3-4 f**, **Fig. 6f**, Mann-Whitney U test, $N_{\text{Net}} = 10$, $*p < 0.01$). Taken together, these results suggest that face-selectivity in untrained networks can be refined differently depending on the stimuli given during the visual training process. We discuss this issue in lines 317-355 in the revised manuscript.

Revised text (Result, lines 317-355)

We tested a scenario in which our current model can corroborate the conflicting observations regarding the role of visual experience for the development of face-selectivity. A previous report suggested that visual experience is necessary for the emergence of face-selectivity by showing that monkeys raised without exposure to faces lack face-selective domains (Arcaro et al., 2017). On the other hand, another recent study showed that the face-selective area develops robustly in congenitally blind humans, suggesting that visual experience is not necessary for face-selectivity^{28,29}. Regarding these conflicting results, we examined how face-selective units in untrained networks can be affected by training with visual inputs.

To investigate the effect of visual experience under different conditions, we prepared three distinct stimuli sets: (1) a face-deprived ImageNet (images including faces excluded), (2) the original ImageNet, and (3) the original ImageNet with added face images used in the current study (**Fig. 6a**). Then, the network was trained with each of these image sets. First, we found that the face-selectivity index (FSI) of the face-selective units was significantly decreased after being trained to the face-deprived image set (**Fig. 6b**, Mann-Whitney U test, $n_{\text{Untrained}} = 4,267$, $n_{\text{Deprived}} = 2,452$, $***p = 3.5 \times 10^{-28}$), whereas it was increased after being trained to the face-including image sets (Mann-Whitney U test, $n_{\text{Face}} = 3,585$, $**p = 5.2 \times 10^{-4}$). Notably, the FSI was significantly decreased after being trained to the original ImageNet dataset that contains images of faces but has no group labeled as “face” (**6b**, Mann-Whitney U test, $n_{\text{Original}} = 3,561$, $*p = 9.3 \times 10^{-45}$). This suggests that the tuning of face-selective units could either be sharpened or weakened by training with distinct stimulus sets.

Next, we found that the number of face-selective units observed was greater in the network trained to face-including image sets compared to that trained to face-deprived images (**Fig. 6c**, Mann-Whitney U test, $N_{\text{Net}} = 10$, $*p < 1.3 \times 10^{-3}$). Interestingly, however, we found that the number of face-selective units, when trained to face-including images, appeared to be smaller than that of untrained networks. These results imply that the training

process of the network to face-including images selectively sharpens the tuning of face units so that the selectivity of strongly tuned units is sharpened while the weakly tuned units are pruned. In this condition, the face detection performance of the networks would improve in face-trained networks even if the number of face units decreased compared to the initial, untrained condition. To validate this scenario, we trained the SVM using the response of face-selective units for a face detection task in an untrained network and in the three networks trained to each type of data set. As predicted, we found that the face detection performance was significantly increased in the networks trained to the face-including image set compared to that of the untrained network (**Fig. 6d**, Mann-Whitney U test, $n_{\text{trial}} = 1,000$, $**p = 1.3 \times 10^{-3}$), whereas the face detection performance of the network trained to the face-deprive image set was significantly decreased compared to the untrained network (**Fig. 6d**, Mann-Whitney U test, $n_{\text{trial}} = 1,000$, $*p = 2.1 \times 10^{-4}$). This result is consistent with the previous observation of decreased face-selectivity in face-deprived monkeys (Arcaro et al., 2017).

We further investigated the effect of visual training on the observed invariance of face-selective responses to low-level feature variation. We examined the effective range of invariance for a spatial translation, scaling and rotation of the stimulus in untrained networks (**Fig. 6e**, black) and networks trained to face images (**Fig. 6e**, blue). We found that the effective range of invariant responses appeared to be significantly wider in the trained networks compared to the untrained ones for both translation and scaling variation (**Fig. 6f**, $N_{\text{Net}} = 10$, $*p < 0.01$). Taken together, these results suggest that face-selectivity in untrained networks can be refined differently depending on the stimuli given during the visual training process.

Q4. What are the authors defining as a “neuron”? The work examines face selectivity in Conv5. Conv5 has 256 channels distributed in a $\sim(3 \times 3)$ matrix across the image (Supp. Table 1), yet the manuscript reports that 615 ± 122 “face-selective neurons” were found. Is that number summed across shuffled networks? It probably is not *per network*, unless a neuron is defined by both channel and position ($256 \times 3 \times 3$) would be a mistake, because a given filter is replicated at each position (“weight-sharing,” a defining property of convolutional neural networks). Please provide further clarification.

We agree that the term “neuron” is not the best choice for our current model. In the revised text, we replaced the term “neuron” with “unit” to avoid confusion.

In our model, “unit” is defined as a unit component at each position of the channel in an activation map of the network. For example, there are 43,264 “units” ($= 13 \times 13 \times 256$, $N_{x\text{-position}} \times N_{y\text{-position}} \times N_{\text{channel}}$) in Conv5. We defined this “unit” in a convolutional network as a simplified model of a biological unit (a single neuron or a group of neurons that generates a tuned activity), considering that the dynamics of a single neuron in biological brains can be estimated from its receptive field, which behaves as a spatiotemporal filter at a local cortical position retinotopically matching the external visual space. In the revised manuscript, we describe the definition of our “unit” in the convolutional network explicitly.

Revised text (Result, lines 82-84)

Here, a “unit” is defined as a unit component at each position of the channel in an activation map of the network. For example, there are 43,264 “units” ($=13 \times 13 \times 256, N_{x\text{-position}} \times N_{y\text{-position}} \times N_{\text{channel}}$) in Conv5.

Q5. Support vector machines are powerful, and easy to train to classify patterns as long as images from their training set are comparable to those in their test set. Which is why I find it very, very surprising that SVMs showed chance performance when trained to discriminate pictures of faces vs. non-faces using non-face-selective units (Fig. 3c). A similar experiment was performed in vivo (Haxby et al., 2001, Science) and the opposite result was found (decoders could do well without face-selective regions, when trained well). What would be a reason for this discrepancy?

Please note that our analysis in Fig. 3c and that in Haxby et al. are quite different: in our current analysis, the control group “non-selective” refers to units that are not selective to any of the five object classes (face, character, body, scene and object) used here. On the other hand, Haxby et al. used activities of neurons just outside the face-selective regions, very likely composed of activities selective to other objects and possibly some visual features relevant to the information of faces. Thus, it is not very surprising that these two results are different. Importantly, furthermore, we confirmed that the results of Haxby et al. can be successfully reproduced by our current model using the SVM trained with units selective to non-face classes in an untrained network. These results demonstrate that our finding of face detection in an untrained network is not contradictory to previous observations, including that by Haxby et al. 2001, but may lead to meaningful discussions relevant to biological characteristics observed in human and animal studies.

Figure R3Q5 (Supplementary Fig. 7)

In detail, to regenerate the experiment condition of the previous human experiment (Haxby et al., 2001), we first trained the SVM using the responses of four distinct populations: (1) all of the units selective to each class (“All-selective”), (2) units selective to non-face classes (“Non-face-selective”), (3) face-selective units only (“Face-selective”), and (4) units not selective to any of the classes (“Non-selective”) (**R3Q5 a, Supplementary Fig. 7a**). As a result, we found that the SVM trained with “Non-face-selective” units showed a performance comparable with the results of Haxby et al. (2001) (**R3Q5 b, c, Supplementary Fig. 7b, c**). Interestingly, the performance was also

comparable with those with “All-selective” units (**R3Q5 b, Supplementary Fig. 7b**, Mann-Whitney U test, $n = 100$, n.s., $p = 0.87$) and those with face-selective units only (Mann-Whitney U test, $n = 100$, n.s., $p = 0.44$). This result is understandable considering that there are only five image classes; thus, even “Non-face-selective” units can provide information for discriminating face and non-face images by generating different levels of activities for each class. Overall, these results imply that units selective to non-face classes can enable the SVM to detect a face, as the reviewer noted in the comment.

Again, we confirmed that the SVM performance outcomes using “Non-selective” units are significantly lower than those with face-selective units (**R3Q5 b, Supplementary Fig. 7b**, Mann-Whitney U test, $n = 100$, $**p = 1.4 \times 10^{-34}$) and not significantly different from those with shuffled responses (Mann-Whitney U test, $n = 100$, n.s., $p = 0.59$). We discuss this issue in lines 188-202 in the revised manuscript.

Revised text (Result, lines 188-202)

Notably, we also found that the SVM can successfully detect faces when it is trained with the responses of units selective to non-face classes, similar to the results in a previous experiment in human⁵⁸, whereas it failed to detect faces with units not selective to any of the classes (**Supplementary Fig. 7**). To compare our results with the experiment condition of the previous human experiment⁵⁸, we first trained the SVM using the responses of four distinct populations: (1) all of the units selective to each class (“All-selective”), (2) units selective to non-face classes (“Non-face-selective”), (3) face-selective units only (“Face-selective”), and (4) units not selective to any of the classes (“Non-selective”) (**Supplementary Fig. 7a**). As a result, we found that the SVM trained with “Non-face-selective” units showed a performance comparable with the results of Haxby *et al.* (2001)⁵⁸ (**Supplementary Fig. 7b, c**). Interestingly, the performance was also comparable with those with “All-selective” units (**Supplementary Fig. 7b**, Mann-Whitney U test, $n_{\text{trial}} = 100$, n.s., $p = 0.87$) and those with face-selective units only (Mann-Whitney U test, $n_{\text{trial}} = 100$, n.s., $p = 0.44$), similar to the results in a previous experiment in human⁵⁸. This result is understandable considering that there are only five image classes; thus, even “Non-face-selective” units can provide information for discriminating face and non-face images by generating different levels of activities for each class. Taken together, these results imply that the information provided by selective units that emerge in the untrained networks is sufficient to detect between faces and non-face objects.

Q6. Invariance. “The responses of face-selective neurons are invariant to different sizes”. The receptive field of neurons in Conv5 would correspond to about 1/3 of the image, and each of the scaling/translation transformations appear to change images in a range that suitable for these large receptive fields. However, this should be true for all hidden units in the layer *if* the other units are responding to their preferred stimulus. The absence of this comparison is distracting. The rotation invariance is interesting, but again, a pixelwise analysis showing how similar each reference image is to its transformed versions would provide a useful baseline for comparison, and aiding the reader in learning if this is a surprising result.

We appreciate this constructive comment. In the revision, we further investigated the invariance of face-selective units using stimulus sets in which the variation of size/position/rotation exceeds the size of the receptive field. We found that face-selective units retain their invariance within a certain range (within their receptive fields) of variation

but lose invariance outside this range, confirming that the observed invariance is not obvious or guaranteed under any condition. Notably, we also observed that the effective range of invariance is wider in the population response compared to the single unit response, as is the response of a single face unit compared to that estimated from raw image correlations.

In detail, we initially generated stimulus sets in which the variation of the size/position/rotation exceeds the size of the receptive field. First, we observed that single face units show constant face tuning under a fairly wide range of size/position/rotation variations (**R3Q6 a, Fig. 4c**, Mann-Whitney U test, $n = 200$, $*p < 0.05$). In addition, we confirmed that the invariance was also observed in the population response for which we averaged the responses of all face-selective units in an untrained network (**R3Q6 b, Fig. 4d**). Next, to examine the effective range of the observed invariance in the face unit responses quantitatively, we estimated the boundary of the size/position/rotation variations around which face tuning is lost (**R3Q6 b, Fig. 4d**). We found that translation invariance was retained when the positional shift was within approximately 75% of the receptive field radius (i.e., from $-0.75 \times r_{RF}$ to $0.75 \times r_{RF}$) (**R3Q6 b, Fig. 4d, left**). This indicates that these units show consistent face tuning within their receptive fields. Similarly, scaling invariance was retained when the original images (100%) were scaled from 40% to 190%. Rotation invariance was retained when the original images were rotated by any angle (from -180 to 180 degrees).

Next, we compared the effective range of invariance using the single unit responses of face-selective units (**R3Q6 a, Fig. 4c**), that of population responses (**R3Q6 b, Fig. 4d**), and that estimated from the pixelwise raw image correlation between original faces and faces with low-level features varied (**R3Q6 c, Fig. 4e**): In this case, the effective range of the image correlation was defined as the range in which the image correlation between an original

face and a transformed face image is significantly larger than that estimated between the original face and transformed non-face images (**R3Q6 c**, **Fig. 4e**, Mann-Whitney U-test, $n = 200$, $*p < 0.05$). As a result, we found that single unit responses show a narrower effective range of invariance to object translation/scaling/rotation compared to that of population average responses (**R3Q6 d-f**, **Fig. 4f**, Mann-Whitney U-test, $n_{\text{pop}} = 100$, $n_{\text{single}} = 28836$, $**p < 9.9 \times 10^{-7}$), whereas both the population responses and the single unit responses still showed a significantly wider effective range of translation/scaling/rotation invariance compared to that of the raw image pixel correlation (Mann-Whitney U-test, $n_{\text{single}} = 28836$, $n_{\text{corr}} = 1$, $*p < 0.03$). These results imply that the invariant response of a face-selective unit is not due to raw image correlations. We added these results and a related discussion to the revised manuscript in lines 204-229.

Revised text (Result, lines 204-229)

To investigate whether the observed face-selective units show invariant representations of face images regardless of the corresponding image condition, we measured the responses of face-selective units to face and non-face object images with various positions, sizes, and rotation angles (**Fig. 4a, b**). First, we observed that single face units show constant face tuning under a fairly wide range of size/position/rotation variations (**Fig. 4c**, Mann-Whitney U test, $n = 200$, $*p < 0.05$). In addition, we confirmed that the invariance was also observed in the population response for which we averaged the responses of all face-selective units in an untrained network (**Fig. 4d**). Next, to examine the effective range of the observed invariance in the face unit responses quantitatively, we estimated the boundary of the size/position/rotation variation around which face tuning is lost. We found that translation invariance was retained when the positional shift was within approximately 75% of the receptive field radius (**Fig. 4d**, left, Mann-Whitney U test, $n = 200$, $*p < 0.05$). This indicates that these units show consistent face tuning within their receptive fields. Similarly, scaling invariance was retained when the original images (100%) were scaled from 40% to 190% (**Fig. 4d**, middle). Rotation invariance was retained when the original images were rotated by any angle (from -180 to 180 degrees) (**Fig. 4d**, right).

Next, we compared the effective range of invariance using the single unit responses of face-selective units (**Fig. 4c**), that of population responses (**Fig. 4d**), and that estimated from the pixelwise raw image correlation between original faces and faces with low-level features varied (**Fig. 4e**): In this case, the effective range of the image correlation was defined as the range in which the image correlation between an original face and a transformed face image is significantly larger than that estimated between the original face and transformed non-face images (**Fig. 4e**, Mann-Whitney U-test, $n = 200$, $*p < 0.05$). As a result, we found that single unit responses show a narrower effective range of invariance to object translation/scaling/rotation compared to that of population average responses (**Fig. 4f**, Mann-Whitney U-test, $n_{\text{pop}} = 100$, $n_{\text{single}} = 28,836$, $**p < 9.9 \times 10^{-7}$), whereas both the population responses and the single unit responses still showed a significantly wider effective range of translation/scaling/rotation invariance compared to that of the raw image pixel correlation (**Fig. 4f**, Mann-Whitney U-test, $n_{\text{single}} = 28,836$, $n_{\text{corr}} = 1$, $*p < 0.03$). In addition, we found that the effective range of the invariant response shows values comparable to those measured in face-selective neurons in the monkey IT⁵⁹ (**Supplementary Fig. 8a, b**, Mann-Whitney U test, $n = 200$, n.s., $p = 0.11$).

Q7. Viewpoint invariance. This is a particularly interesting finding raised by the manuscript. While it is plausible that hidden units could be selective for specific objects based on random selectivity, and invariant to size and position given basic RF properties, it is less clear how *viewpoint* invariance could arise randomly. The first question is, if random wiring projections can give rise to viewpoint invariance, to what else are they giving rise? My concern is that there may be other random wiring phenomena that are just as common yet decidedly non-biological, such a neuron that prefers an object at one size and another object at a different size. Again, the focused reporting of the behavior of *one type* of units in a shuffled network leaves no context to the reader.

We appreciate this very relevant comment. First, we are confident that the analyses in Figs. 5d-f provide a plausible explanation of how viewpoint invariance can arise in hierarchical random feedforward networks. This result demonstrates that viewpoint-invariant units can be generated by receiving inputs from viewpoint-specific units of different viewpoint angles in the previous layer, which is realized readily in random feedforward networks, as in our simulations. This mechanism is comparable to the emergence of orientation-tuned complex cells from the wirings of simple cells of different phases in the visual cortex, as suggested by the classical Hubel-Wiesel model.

Figure R3Q7 (Fig. 5)

For a detailed validation of the scenario in which the viewpoint-invariant units can arise by receiving feedforward connections from viewpoint-specific units of various angles, as shown in the previous analysis (R3Q7 a), we backtracked the projections of the units from the source layer (Conv4) to the target layer (Conv5) and examined the profile of connected viewpoint-specific units (R3Q7 a, left, Fig. 5d, left). We found that the viewpoint-invariant units in Conv5 receive inputs from Conv4 units with a fairly homogeneous distribution of viewpoint angles (R3Q7 a, middle and right, Fig. 5d, middle and right). These results suggest that the viewpoint-invariant face tunings can develop due to random feedforward projections by chance from the viewpoint-specific units existing in the pre-synaptic layer.

With regard to the question of what else can arise from random wiring projections, this is indeed an important question, and we are working on several different projects regarding this issue. One interesting example is the emergence of visual number sense in untrained DNNs, the details of which are discussed in our recent publication (R3Q7 b, Kim and Jang et al. Science Advances 2021). There are also other higher cognitive functions that we are investigating now, and they will be introduced in our subsequent studies. Eventually, we hope to address this broad question: “How can various functions arise from untrained random networks?” and the current study can lay the important groundwork for this type of subsequent inquiry.

As suggested, one may be concerned that some observed results may not be biologically plausible — and this is always possible in any type of model study. We are aware of this and consider it important. Therefore, as shown in the current manuscript, we are attempting to find relevant experimental observations to compare with our simulation findings, and thus far most of our findings appear to be consistent with previously observed data in the brain. On the other hand, in the case of a novel finding that has never been reported previously in experimental observations, this can contribute as a new prediction, and we may suggest a biological experiment that can validate our prediction. Even in cases where our simulation results could not be readily reproduced or observed in biological brains, such results may provide insight into feasible designs of new types of artificial neural networks as an engineering application.

Q7-a. It would be useful to note where in the Freiwald et al 2010 paper the Fig. 5a tuning curve appears – I couldn't find it.

We apologize for our incomplete description. The tuning curve in Fig. 5a was plotted from the digitized data of Fig. 2 in Freiwald et al. (2010). In the revised manuscript, we replaced this tuning curve with our additional analysis result. For details, please refer to our Fig. 5a.

In summary, this an interesting manuscript, worth developing, but not supportive of its overall framing, and still largely underdeveloped in its analyses. What I was most intrigued about was that the authors have shown that selectivity for arbitrary image classes can occur in random networks (generalizing beyond faces), which raises the possibility that this is a general property of random computational graphs. The key reveal here might be that since selectivity for arbitrary image classes is a general property of randomly initialized CNNs, neuroscientists need to re-develop their intuitions for computation in networks, and what it means to be “selective.”

We sincerely appreciate your considerate comments and suggestions. As described in our responses to your previous questions, here we redeveloped our overall framing and demonstrated that selectivity for arbitrary images classes can arise in untrained random networks. Our additional simulations and analyses suggest that our findings can be considered as a general property of randomly initialized hierarchical neural networks, with this providing new insight into neural tuning, which is often defined as “selective” responses.

It should be noted, however, that units selective for only a small number of object classes could arise in an untrained network, such as faces that show distinctive clustering in the latent space. These results imply that an untrained network is not a master key for generating selectivity to any arbitrary image or any arbitrary function. Instead, this provides a possible explanation of why some types of selectivity are readily observed in the brain, whereas others are not. This may be a crucial factor to consider for understanding the mechanism underlying the emergence and evolution of innate cognitive functions in biological systems.

Reviewer #4

The paper addresses the question of whether face-selectivity of the kind known to exist in face-related cortical regions can arise spontaneously or whether it requires training from visual experience. It reports the results of a computational study of face-selective responses in deep neural networks prior to any training, using only randomly initialized weights. The main finding of the study is that face-selective neurons can emerge across different conditions of randomly initialized deep neural networks. The activity of these units was sufficient to enable the network to perform discrimination between face and non-face stimuli. The face-preferring neurons showed selectivity to facial features, their face selectivity index was similar to that of cortical neurons, they were insensitive to low-level visual feature variations such as the translation, rotation, and scaling of face images, and some showed a degree of viewpoint invariance. Overall, the findings are novel and are well supported by the experiments and results reported in the paper, but there are several comments that need to be addressed.

Q1. Regarding relevant previous work with random deep networks, the paper cites two old papers (from 2009), and a recent paper on so-called ‘deep priors’. There is more directly relevant recent work, in particular work related to the so-called ‘lottery ticket hypothesis’. The original paper is by Frankle & Carbin and there are by now additional papers. The most relevant one is by Ramanujan et al (2020), where they show that randomly weighted neural networks contain sub-networks, which achieve impressive performance without ever modifying the weight values. For example, they show a random sub-net that achieves state-of-the-art performance on ImageNet with no changes to the initial random weights. The results are not the same as the current work, as the previous work treats mainly sub-networks rather than individual units, but in a sense, they are stronger. The previous work also does not make comparisons with the brain, but it shows something strong about the probability of the emergence of functionally meaningful structures in random DNNs.

Thank you for this important comment. In the revised manuscript, we added citations of relevant studies regarding the “lottery ticket hypothesis” and further discussed the similarities and differences between this model and our current work.

It is true that the lottery ticket model and our current model have a common aspect in that functional structures can emerge without modifications of the initial random weights. However, the main difference between the two models is that the lottery ticket hypothesis showed that a random “subnetwork” can perform tasks without learning, while our model demonstrates that the functional tuning of “single units” (neuronal tuning) can arise spontaneously. Therefore, the previous model could not explicitly predict the functional tuning in individual units, which is a key result when comparing the model results with observations in the brain. Indeed, our model not only suggests a possible mechanism of the origin of the functional tuning of single neurons in the early developmental process in the brain but also provides insight into the mechanism underlying the emergence of functional subnetworks, possibly from tuned individual units in random artificial neural networks.

Another notable difference is that the lottery ticket hypothesis requires a training process to find subnetworks. To find a subnetwork in the lottery ticket model (Frankle et al., 2019; Ramanujan et al., 2020) — even when the weights of the subnetwork are unchanged — it is necessary to prune the unnecessary weights in the remaining network. This pruning process can be a type of pseudo-training (or a modulation of tuning) to individual

units, as the tuning profile of a single unit could be significantly altered by the process. For example, our model shows that the selectivity of a face unit can be noticeably sharpened if all of the weights unnecessary for face feature detection are pruned, as doing so will drop the response to non-faces. Overall, extensive studies that test for a relationship between the functional tunings of individual units and that of a subnetwork must be conducted in successive studies. We added a discussion of this issue in the revised manuscript.

Revised text (Introduction, lines 52-60)

Notably, previous studies using random hierarchical networks provide important clues about the origin of innate face-selectivity in untrained neural networks. It was reported that untrained feedforward networks can initiate various cognitive functions with random weights and that a random network can perform image classification tasks in that way as well (Frankle et al., 2019; Ramanujan et al., 2020). It was also reported that a randomly initialized convolutional neural network could reconstruct corrupted images without any training, which implies that a random network can provide a priori information about the low-level statistics in natural images⁴⁷. Overall, such observations suggest the possibility of the emergence of innate cognitive functions, such as primitive face-selectivity in untrained, random hierarchical networks. However, the details of how this innate function emerges in untrained neural networks are not yet understood.

Revised text (Discussion, lines 377-385)

Furthermore, recent model studies using the “lottery ticket hypothesis” (Frankle et al., 2019; Ramanujan et al., 2020) showed that randomly weighted neural networks contain subnetworks that can perform tasks without modifying the initial random weight values, implying that functional architectures can emerge from random initial wirings. This model and our current model have a common aspect in that functional structures can emerge without any modification to the initial random weights. However, the lottery ticket hypothesis showed that a random “subnetwork” can perform tasks without learning, while our model demonstrates that the functional tuning of “single units” can arise spontaneously. Indeed, our model not only suggests a possible mechanism of the origin of the functional tuning of single neurons in the early developmental process in the brain but also provides insight into the mechanism underlying the emergence of functional subnetworks, possibly from tuned individual units that emerge in random artificial neural networks.

Q2. One of the results listed above was invariance to low-level visual feature variations such as the translation, rotation, and scaling of face images. This conclusion depends on how invariance is defined and measured, and a finer comparison may show that the network units and cortical neurons are different in the response to variations in the input image. A number of recent works have found that small translations or rescaling of the input image can drastically change the predictions of DNNs. (Some review and analysis can be found in Azuly & Weiss 2019). The invariant analysis in the current paper is based on ANOVA, which shows general statistical trends. In contrast, other work tested individual DNN units with small (few pixel) translations, rotations or scale, and found that such changes can make changes in the output, in a way that is not found in cortical units. It

will be of interest to check this point with the face-preferring units, to further test their similarity to biological units in terms of local invariance.

We appreciate this constructive comment. In the revised manuscript, we used a modified definition of invariance for a finer comparison, as described below in detail. In addition, we investigated the tuning curves of face units quantitatively while varying the low-level feature variations of input images to estimate the effective range of invariance in the tuning curves. We confirmed that the observed characteristics of model units are similar to those observed in the monkey IT neurons.

Figure R4Q2 (Fig. 4, Supplementary Fig. 8, 9)

First, we redefined the “invariance” of face tuning (**R4Q2 a, b, Fig. 4a, b**): Instead of using ANOVA, we compared each pair of responses to face and non-face images using Mann-Whitney U-tests. Thus, the invariance of face-selectivity was checked more strictly; i.e., it was considered “invariant” only when the responses of units to face images are significantly higher than the responses to any other non-face image (in every face vs. non-face pair). Under this new condition, we observed that single face units show constant face tuning under a fairly wide range of size/position/rotation variations (**R4Q2 c, Fig. 4c**). In addition, we confirmed that the invariance was also observed in the population response for which we averaged the responses of all face-selective units in an untrained network (**R4Q2 d, Fig. 4d**).

To examine the effective range of invariance in the face unit responses quantitatively, we estimated the boundary of the size/position/rotation variations around which face tuning is lost (**R4Q2 c, Fig. 4c**). We found that translation invariance was retained when the positional shift was within approximately 75% of the receptive field radius (i.e., from $-0.75 \times r_{RF}$ to $0.75 \times r_{RF}$) (**R4Q2 d, left, Fig. 4d, left**). This indicates that these units show consistent face tuning within their receptive fields. Similarly, scaling invariance was retained when the original

images (100%) were scaled from 40% to 190% (**R4Q2 d**, middle, **Fig. 4d**, middle). Rotation invariance was retained when the original images were rotated by any angle (from -180 to 180 degrees) (**R4Q2 d**, right, **Fig. 4d**, right).

Next, we compared the effective ranges of invariance observed in our untrained network and in the previous data in the monkey IT. We found that the effective ranges for the size, position, and rotation invariance observed in our model networks were fairly comparable to those observed in the monkey IT (Zoccolan et al., 2007, Tsao et al., 2006) (**R4Q2 e-g**, **Supplementary Fig. 8, 9**). These results imply that the face-selective units observed in our model networks reveal a similar characteristic of invariance with those in biological brains.

Revised text (Result, lines 204-235)

To investigate whether the observed face-selective units show invariant representations of face images regardless of the corresponding image condition, we measured the responses of face-selective units to face and non-face object images with various positions, sizes, and rotation angles (**Fig. 4a, b**). First, we observed that single face units show constant face tuning under a fairly wide range of size/position/rotation variations (**Fig. 4c**, Mann-Whitney U test, $n = 200$, $*p < 0.05$). In addition, we confirmed that the invariance was also observed in the population response for which we averaged the responses of all face-selective units in an untrained network (**Fig. 4d**). Next, to examine the effective range of the observed invariance in the face unit responses quantitatively, we estimated the boundary of the size/position/rotation variation around which face tuning is lost. We found that translation invariance was retained when the positional shift was within approximately 75% of the receptive field radius (**Fig. 4d**, left, Mann-Whitney U test, $n = 200$, $*p < 0.05$). This indicates that these units show consistent face tuning within their receptive fields. Similarly, scaling invariance was retained when the original images (100%) were scaled from 40% to 190% (**Fig. 4d**, middle). Rotation invariance was retained when the original images were rotated by any angle (from -180 to 180 degrees) (**Fig. 4d**, right).

Next, we compared the effective range of invariance using the single unit responses of face-selective units (**Fig. 4c**), that of population responses (**Fig. 4d**), and that estimated from the pixelwise raw image correlation between original faces and faces with low-level features varied (**Fig. 4e**): In this case, the effective range of the image correlation was defined as the range in which the image correlation between an original face and a transformed face image is significantly larger than that estimated between the original face and transformed non-face images (**Fig. 4e**, Mann-Whitney U-test, $n = 200$, $*p < 0.05$). As a result, we found that single unit responses

show a narrower effective range of invariance to object translation/scaling/rotation compared to that of population average responses (**Fig. 4f**, Mann-Whitney U-test, $n_{\text{pop}} = 100$, $n_{\text{single}} = 28,836$, $**p < 9.9 \times 10^{-7}$), whereas both the population responses and the single unit responses still showed a significantly wider effective range of translation/scaling/rotation invariance compared to that of the raw image pixel correlation (**Fig. 4f**, Mann-Whitney U-test, $n_{\text{single}} = 28,836$, $n_{\text{corr}} = 1$, $*p < 0.03$). In addition, we found that the effective range of the invariant response shows values comparable to those measured in neurons in the monkey IT⁵⁹ (**Supplementary Fig. 8a, b**, Mann-Whitney U test, $n = 200$, n.s., $p = 0.11$).

Notably, we found that our model units also show the “inversion effect” observed in monkeys^{13,27,60}. We found that the responses of face-selective units to inverted face images are significantly lower than those to upright faces (**Supplementary Fig. 9a, b**, Mann-Whitney U test, $n = 200$, $**p = 7.8 \times 10^{-13}$), whereas these responses are still higher than those to non-face images (Mann-Whitney U test, $*p = 4.8 \times 10^{-7}$). These results indicate that our face-selective units show similar responses to upright and inverted faces, i.e. the “inversion effect”, as observed in the monkey IT¹³ (**Supplementary Fig. 9c**, Mann-Whitney U test, $n = 16$, $**p = 2.6 \times 10^{-22}$, $*p = 5.2 \times 10^{-3}$).

Q3. One of the findings above reports that face-neurons are selective for facial features, including the eyes, mouth, and facial contour. The images preferred by the units show sensitivity to image features such as blobs and linear features in a face-like configuration, but I think that it will be better to describe the finding in terms of sensitivity to coarse face-like configurations rather than sensitivity to eyes, mouth etc.

Thank you for pointing this out. As suggested, we corrected the terminology in the revised manuscript: the terms “sensitivity to eyes, mouth” were replaced with “coarse face-like configurations”. In addition, we performed an additional analysis using the generative adversarial network (GAN) that was used to examine the preferred feature images (PFI) of monkey IT neurons in a previous study (X-Dream, Ponce et al., 2019) to demonstrate the PFI of face-selective units containing facial features more precisely. As a result, we confirmed that the coarse face configurations were observed in the PFI generated by X-Dream as well as those generated by the reverse correlation method. These results show that the coarse face configuration could generally appear in PFI regardless of the PFI computation method used (**R4Q3, Supplementary Fig. 4**), demonstrating that face-selective units in untrained networks encode visual facial features.

Figure R4Q3 (Supplementary Fig. 4)

Revised text (Results, lines 140-146)

As a result, using both methods, we obtained the PFI of face-selective units, units selective to non-face classes,

and units without selective responses to any image classes (**Fig. 2b, d, Supplementary Fig. 4**). We found that the PFI of the face-selective units presents distinguishable features from those of other objects (**Fig. 2b, d, Supplementary Fig. 4**). We observed that the PFIs of face-selective units represent face-like configurations, whereas the PFIs of units selective to non-face classes show only barely noticeable configurations of each object class (only except for a couple of cases, i.e., flowers in a reverse correlation and hands in X-Dream).

Q4. The current study also examined the view-point-invariance of the face-selective units, and found both view-specific and view-invariant units, with layer-specificity, where the view-invariant units appearing at a higher level. In the monkey visual system, work on face patches by Freiwald, Tsao and others found that the view-selective units in the AL patch, feeding the view-invariant units in the AM patch, show a characteristic mirror symmetric tuning. It will be interesting to test this characteristic property in the view-variant units feeding the more invariant units at higher levels.

In the revision, we found that the characteristic mirror-symmetric tuning mentioned above is observed in our model neural network.

Figure R4Q4 (Supplementary Fig. 11)

As suggested, we investigated mirror-symmetric tuning in an untrained network (**R4Q4 a, Supplementary Fig. 11a**). We defined mirror-symmetric tuning as the condition that arises when a viewpoint-specific face-selective unit has a symmetric shape of the tuning curve (i.e., a unit shows peak responses at -45° and 45° or -90° and 90°). From the simulations of 100 randomly initialized untrained networks, we observed 74 ± 24 mirror-symmetric units out of 43,264 units in Conv5. The number of mirror-symmetric units is largest in the mid-level layer, i.e., Conv4,

similar to observations in monkey data (Freiwald and Tsao, 2010) (**R4Q4 b, Fig. 5a, Supplementary Fig. 11b**).

Next, we tested whether mirror-symmetric characteristics appear in viewpoint-selective units Conv4, which feed the viewpoint-invariant units in Conv5. We examined the ratio of mirror-symmetric units among the viewpoint-specific units in Conv4, connected to viewpoint-specific and viewpoint-invariant units in Conv5, respectively (**c, left**). As a result, we found that the ratio of mirror-symmetric units is significantly higher in units connected to viewpoint-invariant units compared to those connected to viewpoint-specific units (**c, right**; Mann–Whitney U test, $N_{\text{Net}} = 100$, $*p = 2.4 \times 10^{-3}$), as observed in the monkey (AL, AM; Freiwald and Tsao, 2010). These results reveal that face-selectivity in an untrained network regenerates various characteristics of face units observed in the brain. We added this result in lines 286-293 in the revised manuscript.

Revised text (Result, lines 286-293)

Next, we investigated whether mirror-symmetric tuning, another interesting characteristic of face neurons observed in the monkey IT¹⁵, can arise in our untrained networks. To do this, we defined mirror-symmetric tuning as a condition in which a viewpoint-specific face-selective unit has a symmetric shape of the tuning curve (i.e., a unit shows peak responses at -45° and 45° or -90° and 90°). From the repeated simulations of 100 randomly initialized untrained networks, we observed 74 ± 24 mirror-symmetric units out of 43,264 units in Conv5 (**Supplementary Fig. 11a**). The number of mirror-symmetric units is largest in the mid-level layer, i.e., Conv4, similar to observations in monkey data (**Supplementary Fig. 11b**, Conv1: $n = 0.002 \pm 0.001\%$, Conv2: $n = 0.098 \pm 0.023\%$, Conv3: $n = 0.167 \pm 0.041\%$, Conv4: $n = 0.173 \pm 0.039\%$, Conv5: $n = 0.171 \pm 0.056\%$).

The relevance to face-selective regions in primate cortex:

Q5. The study shows the emergence of units with preference for faces and face processing functionality (e.g. face detection), arising in DNN with random weights. This is novel and interesting, but perhaps not entirely surprising in view of the findings (following the ‘lottery ticket hypothesis’) that random DNNs contain sub-networks with complete functionality, e.g. performing object recognition across multiple categories. The findings in the current study are novel in their focus on face preferring units. They add to the findings about the statistics of functional components in random networks, but the possible biological relevance, even in theory is less clear. Given the background above, it becomes of interest to know whether there is something special about faces, or that the same phenomenon holds for other object classes (as appears to happen in the case of random sub-networks performing ImageNet tasks).

We appreciate this timely comment. From an additional analysis of latent space representations, we suggest that face tuning is distinctive from tunings to other object classes. Using the dimension-reduction method, we found that the network responses to face images present stronger clustering compared to those of other tested objects. This result may provide a possible explanation of why faces are readily distinguishable from other objects in untrained networks. Specifically, a t-distributed stochastic neighbor embedding (tSNE) analysis of the network response shows that face images are statistically more clustered in the embedding space than other objects.

To compare the observed face tuning with tunings to other object classes in an untrained network, we generated the PFIs of units selective to faces and to other object classes (**R4Q5 a, Supplementary Fig. 4**, hand, flower, horn, chair). Using the reverse correlation method and a generative adversarial network (X-Dream, Ponce et al., 2019), we initially observed that the PFIs of face-selective units represent face-like configurations in both the reverse correlation method and X-Dream, whereas the PFIs of units selective to non-face classes show only barely noticeable configurations of each object class (only except for a couple of cases, i.e., flowers in the reverse correlation and hands in X-Dream). From this result, we hypothesized that faces can induce stronger clustering in latent representations than those of other object classes in untrained networks. To validate our hypothesis, the t-distributed stochastic neighbor embedding (t-SNE) method was used for dimension reduction (Maaten and Hinton, 2008). By minimizing the difference between the original and low-dimensional distributions of neighbor distances, a 2D representation of the responses of the fifth convolutional layer was obtained. To quantify the level of clustering of each class, we measured the Silhouette index (SI) (Kaufman and Rousseeuw, 2009) to estimate the consistency of data clustering. The Silhouette index SI for the i^{th} point is defined as $c_i = (b_i - a_i) / \max(a_i, b_i)$, where a_i and b_i refer to the intra-class distance and the inter-class distance for the i^{th} point, respectively.

Figure R4Q5 (Supplementary Fig. 4, 5)

We found that the Silhouette index is all negative when the raw pixel values of each image are represented in the tSNE space (**R4Q5 b, Supplementary Fig. 5a**). However, for the Conv5 layer responses, the Silhouette index of face responses showed positive values, whereas those of all of the other classes showed significantly lower negative values (**R4Q5 c, Supplementary Fig. 5b**). This result demonstrates that face images have statistics readily distinguishable from the statistics of other objects, leading to the strong clustering of abstracted responses

in the DNN. Such a special characteristic may be understood by considering the fact that faces have a simple configuration of geometric components, a combination of only several ovals or dots. Thus, innate face-selectivity is more likely to emerge in the brain compared to other object classes due to the simple geometric features of faces. This result may explain why neurons selective to every object class are not readily observed in the brain (Tsao et al., 2003).

Revised text (Results, lines 140-154)

As a result, using both methods, we obtained the PFI of face-selective units, units selective to non-face classes, and units without selective responses to any image classes (**Fig. 2b, d, Supplementary Fig. 4**). We found that the PFI of the face-selective units presents distinguishable features from those of other objects (**Fig. 2b, d, Supplementary Fig. 4**). We observed that the PFIs of face-selective units represent face-like configurations, whereas the PFIs of units selective to non-face classes show only barely noticeable configurations of each object class (only except for a couple of cases, i.e., flowers in a reverse correlation and hands in X-Dream).

From this result, we hypothesized that faces can induce stronger clustering in latent space representation than those of other object classes in untrained networks. To validate our hypothesis, we used the dimension-reduction method⁵⁵ (tSNE) to compare a clustered representation of each object class in terms of the raw pixel values and in the responses of Conv5. For a quantification of the representational clustering of each class, we measured the Silhouette index⁵⁶ to estimate the consistency of data clustering. We found that the Silhouette index is all negative when the raw pixel values of each image are represented in the tSNE space (**Supplementary Fig. 5a**). However, for the Conv5 layer responses, the Silhouette index of face responses showed positive values, whereas those of all of the other classes showed significantly lower negative values (**Supplementary Fig. 5b**).

Revised text (Method, lines 460-466)

To visualize the network responses to the low-level feature-controlled stimulus set, the t-distributed stochastic neighbor embedding (t-SNE) method was used for dimension reduction⁵⁵. By minimizing the difference between the original and low-dimensional distributions of neighbor distances, a 2D representation of the responses of the fifth convolutional layer was obtained. To quantify the level of clustering of each class, we measured the Silhouette index (SI)⁵⁶ to estimate the consistency of data clustering. The Silhouette index SI for the i^{th} point is defined as $c_i = (b_i - a_i) / \max(a_i, b_i)$, where a_i and b_i refer to the intra-class distance and the inter-class distance for the i^{th} point, respectively. The same analysis was also performed for the stimulus set that added the face class to ILSVRC2010.

Q6. Another question is whether the emergence of the units described in the study can conceivably seed in the brain something like face regions. This appears necessary to make the findings biologically relevant. Do the authors suggest such a conclusion? If they do, think that the conclusion should be stated explicitly and discussed, since the gap between a tiny fraction of face-preferring units scattered throughout the network, and the anatomy of the face system is very large. To my view, a satisfactory discussion of the possible relevance of the study to brain regions is necessary to making it biologically relevant.

We sincerely appreciate this important, constructive comment. In the revised manuscript, we discuss a possible scenario of clustered face units in an untrained network, which can link the current result with that observed in primates. In brief, our observation that face-selective units in an untrained network appear mostly in mid- and high-level layers in the hierarchy of the network implies an aspect similar to the anatomy of the face system in biological brains.

Previous studies of primates reported that the face-selective units are observed mostly in the mid- and high-level hierarchy of the ventral visual stream (Livingstone et al., 2017; Tsao et al., 2006) and that they are also spatially clustered in the central vision area in retinotopy, as a “face patch” (Arcaro and Livingstone, 2017; Arcaro et al., 2020; Bell et al., 2009). Because the exact anatomical location of the brain correspondent to the units in the convolutional neural network could not be determined, we investigated instead the distribution of face-selective units in the inter-layer level and in the retinotopic location of each face-selective unit in each layer. We found that the number of observed face-selective units is largest in the mid- and high-level layer (**R4Q6 a, Supplementary Fig. 12**) similar to observations in the ventral visual pathway of monkeys (Livingstone et al., 2017; Tsao et al., 2006).

To investigate the retinotopic distribution of face-selective units, we examined the locations of the receptive field center position of each face-selective and object-selective unit. We backtracked the receptive field of each face-selective unit (**R4Q6 b, Supplementary Fig. 13a**) and computed the distances between the center position of the input image and the position of the receptive field. As a result, we found that the spatial encoding of face-selective units is more biased to the center of the visual field compared to units selective to other object classes (**R4Q6 c, Supplementary Fig. 13b**, Mann–Whitney U test, $N_{\text{Net}} = 100$, $*p = 3.9 \times 10^{-8}$) throughout the entire network hierarchy (**R4Q6 d, Supplementary Fig. 13c**). This result is consistent with the previous observation in monkeys that face-selective units are clustered retinotopically in the center-vision area (Arcaro and Livingstone, 2017; Arcaro et al., 2020; Bell et al., 2009). We added this result at lines 294-315 in the revised manuscript.

Revised text (Result, lines 294-315)

We also investigated the layer-specific emergence of face-selective units in untrained networks. We found that face-selective units are also observed in earlier layers, Conv2 to 3 but are scarcely found in Conv1 (**Supplementary Fig. 12**, Conv1: $n = 0.008 \pm 0.002\%$, Conv2: $n = 0.476 \pm 0.095\%$, Conv3: $n = 0.738 \pm 0.133\%$, Conv4: $n = 0.698 \pm 0.137\%$, Conv5: $n = 0.667 \pm 0.172\%$). Notably, the number of face-selective units observed in Conv2 was

significantly smaller than those observed in Conv3, Conv4, and Conv5 (**Supplementary Fig. 12**, Mann-Whitney U test, $N_{\text{Net}} = 100$, $*p < 7.2 \times 10^{-11}$). Thus, the number of observed face-selective units was highest in the mid- and high-level layers, similar to observations in the ventral visual pathway of monkeys^{13,14}. These results suggest that the development of face-selectivity requires a hierarchical structure of the network along with random feedforward weights, which enables multiple linear-nonlinear computations.

Previous studies of primates reported that the face-selective units are also spatially clustered in the central vision area in retinotopy, as a “face patch”^{62–64}. Because the exact anatomical location of the brain correspondent to the units in the convolutional neural network could not be determined, we investigated instead the retinotopic distribution of face-selective units in each layer — we examined the locations of the receptive field center position of each face-selective and object-selective unit. We backtracked the receptive field of each face-selective unit (**Supplementary Fig. 13a**) and computed the distances between the center position of the input image and the position of the receptive field. As a result, we found that the spatial encoding of face-selective units is more biased to the center of the visual field compared to units selective to other object classes (**Supplementary Fig. 13b**, Mann–Whitney U test, $N_{\text{Net}} = 100$, $*p = 3.9 \times 10^{-8}$) throughout the entire network hierarchy (**Supplementary Fig. 13c**). This result is consistent with the previous observation in monkeys that face-selective neurons are clustered retinotopically in the center-vision area^{62–64}. Taken together, these findings show that face-selective units generated in untrained networks have biological characteristics similar to those observed in monkeys, not only in single units but also at the population and inter-layer levels.

References

- Arcaro, M.J., and Livingstone, M.S. (2017). Retinotopic organization of scene areas in macaque inferior temporal cortex. *J. Neurosci.* *37*, 7373–7389.
- Arcaro, M.J., Schade, P.F., Vincent, J.L., Ponce, C.R., and Livingstone, M.S. (2017). Seeing faces is necessary for face-domain formation. *Nat. Neurosci.* *20*, 1404–1412.
- Arcaro, M.J., Ponce, C., and Livingstone, M. (2020). The neurons that mistook a hat for a face. *Elife* *9*, e53798.
- Bell, A.H., Hadj-Bouziane, F., Frihauf, J.B., Tootell, R.B.H., and Ungerleider, L.G. (2009). Object representations in the temporal cortex of monkeys and humans as revealed by functional magnetic resonance imaging. *J. Neurophysiol.* *101*, 688–700.
- Bonin, V., Histed, M.H., Yurgenson, S., and Reid, R.C. (2011). Local diversity and fine-scale organization of receptive fields in mouse visual cortex. *J. Neurosci.* *31*, 18506–18521.
- Buiatti, M., Di Giorgio, E., Piazza, M., Polloni, C., Menna, G., Taddei, F., Baldo, E., and Vallortigara, G. (2019). Cortical route for facelike pattern processing in human newborns. *Proc. Natl. Acad. Sci. U. S. A.* *116*, 4625–4630.
- Frankle, J., Dziugaite, G.K., Roy, D.M., and Carbin, M. (2019). Stabilizing the lottery ticket hypothesis. *ArXiv Prepr. ArXiv1903.01611*.
- Freiwald, W.A., and Tsao, D.Y. (2010). Functional compartmentalization and viewpoint generalization within the macaque face-processing system. *Science* *330*, 845–851.
- Haxby, J. V, Gobbini, M.I., Furey, M.L., Ishai, A., Schouten, J.L., and Pietrini, P. (2001). Distributed and overlapping representations of faces and objects in ventral temporal cortex. *Science* *293*, 2425–2430.
- Kaufman, L., and Rousseeuw, P.J. (2009). Finding groups in data: an introduction to cluster analysis (John Wiley & Sons).
- Kriegeskorte, N., Simmons, W.K., Bellgowan, P.S.F., and Baker, C.I. (2009). Circular analysis in systems neuroscience: the dangers of double dipping. *Nat. Neurosci.* *12*, 535.
- Livingstone, M.S., Vincent, J.L., Arcaro, M.J., Srihasam, K., Schade, P.F., and Savage, T. (2017). Development of the macaque face-patch system. *Nat. Commun.* *8*, 1-12.
- Long, B., Yu, C.-P., and Konkle, T. (2018). Mid-level visual features underlie the high-level categorical organization of the ventral stream. *Proc. Natl. Acad. Sci.* *115*, E9015–E9024.
- Maaten, L. van der, and Hinton, G. (2008). Visualizing data using t-SNE. *J. Mach. Learn. Res.* *9*, 2579–2605.
- Perrett, D.I., Smith, P.A.J., Potter, D.D., Mistlin, A.J., Head, A.S., Milner, A.D., and Jeeves, M.A. (1985). Visual cells in the temporal cortex sensitive to face view and gaze direction. *Proc. R. Soc. London. Ser. B. Biol. Sci.* *223*, 293–317.
- Ponce, C.R., Xiao, W., Schade, P.F., Hartmann, T.S., Kreiman, G., and Livingstone, M.S. (2019). Evolving images for visual neurons using a deep generative network reveals coding principles and neuronal preferences.

Cell 177, 999–1009.

Ramanujan, V., Wortsman, M., Kembhavi, A., Farhadi, A., and Rastegari, M. (2020). What's Hidden in a Randomly Weighted Neural Network? In Proceedings of the IEEE/CVF Conference on Computer Vision and Pattern Recognition, pp. 11893–11902.

Rosa-Salva, O., Regolin, L., and Vallortigara, G. (2010). Faces are special for newly hatched chicks: evidence for inborn domain-specific mechanisms underlying spontaneous preferences for face-like stimuli. *Dev. Sci.* 13, 565–577.

Salva, O.R., Farroni, T., Regolin, L., Vallortigara, G., and Johnson, M.H. (2011). The evolution of social orienting: evidence from chicks (*Gallus gallus*) and human newborns. *PLoS One* 6, e18802.

Stigliani, A., Weiner, K.S., and Grill-Spector, K. (2015). Temporal processing capacity in high-level visual cortex is domain specific. *J. Neurosci.* 35, 12412–12424.

Tovee, M.J., Rolls, E.T., and Azzopardi, P. (1994). Translation invariance in the responses to faces of single neurons in the temporal visual cortical areas of the alert macaque. *J. Neurophysiol.* 72, 1049–1060.

Tsao, D.Y., Freiwald, W.A., Knutsen, T.A., Mandeville, J.B., and Tootell, R.B.H. (2003). Faces and objects in macaque cerebral cortex. *Nat. Neurosci.* 6, 989–995.

Tsao, D.Y., Freiwald, W.A., Tootell, R.B.H., and Livingstone, M.S. (2006). A cortical region consisting entirely of face-selective cells. *Science* 311, 670–674.

Versace, E., Damini, S., and Stancher, G. (2020). Early preference for face-like stimuli in solitary species as revealed by tortoise hatchlings. *Proc. Natl. Acad. Sci.* 117, 24047–24049.

Zoccolan, D., Kouh, M., Poggio, T., and DiCarlo, J.J. (2007). Trade-off between object selectivity and tolerance in monkey inferotemporal cortex. *J. Neurosci.* 27, 12292–12307.

Reviewers' Comments:

Reviewer #1:

Remarks to the Author:

It seems to me that the authors have adequately addressed all my concerns.

Reviewer #2:

Remarks to the Author:

I thank and congratulate the authors for their detailed and comprehensive response to the reviews. The authors put an impressive amount of work in the revision and thoroughly addressed all concerns raised. I have only three small remaining comments regarding the revision:

1) The authors added a new condition to the manuscript which was "face-deprived". Based on my own experience in the past, it is tremendously difficult to remove all the faces from ImageNet. The authors don't give any more details as to how the faces were removed. It is only noted in the methods that 500 classes of the ILSVRC2021 dataset were selected. Did the authors additionally remove faces from these categories using face detection algorithms, or did they manually curate the dataset? In any case, if they cannot be 99.9% certain that there are no more faces in the dataset, this condition should be termed "face-reduced" to make clear that it is possible that there are still faces in the dataset. Also please add more details to the methods.

2) The authors' addressed my concern regarding the generalization of face-selective units (Q2) by testing several other datasets. The results show that face-selective responses indeed generalize to novel sets of faces. However, in the revised text the authors conclude "...face-selective units in an untrained network are selective to general face images...". This is misleading and not the point. Instead, the results suggest that the face-selectivity of units defined by one specific dataset generalize to other datasets. Please correct.

3) The term "innate face-selective units" is confusing and incorrect. In fact, it assumes that an untrained CNN represents a newborns' brain at birth. We don't know that and in fact there is a strong debate whether training CNNs models phylogeny or ontogeny (e.g., Hasson et al., Neuron, 2020). Please remove the term "innate" when describing face selectivity or face-selective units in untrained CNNs.

Nancy Kanwisher

Reviewer #3:

Remarks to the Author:

We applaud the authors for their responsiveness and comprehensive new set of analyses. They clearly showed diligence and hard work in considering our and other reviewers' concerns. We thank you for your efforts.

While the manuscript is improved, it is also apparent where the authors stopped short of pushing their analyses in interesting directions, as it would radically affect the framing of the overall story. This remains the key problem — with the authors' latest batch of results, the current framing becomes massively misleading, and there are facts about convolutional networks that require description as part of the Introduction and which may affect the novelty of the results.

Major concerns.

Concern 1: unmerited elevation of faces as a category with special properties

Per the introduction, discussion and references, this manuscript centers around animals and face selectivity. The results show that in untrained networks, units can show selectivity to some visual categories — as long as such categories have simple geometric features. The authors find dozens

of such categories, including pumpkins, flowers, household tools and yes – faces, too. Given this compelling analysis, I reaffirm the common concern of Reviewer #2 (Q3), Reviewer #4 (Q5) and my own: with the discovery of dozens of new visual categories that elicit selectivity properties as those of face-selective units, the premise of the paper is misleadingly narrow. It would be as if a neurologist described a post-stroke patient's inability to move her arm as an inability to play the piano. True enough, but also a puzzlingly arbitrary underreporting of the facts.

So, are faces special or not? Are they worth of the headline or not? This is not clear. After reporting that units in untrained networks can develop selectivity to dozens other pre-defined semantic categories, the authors try to bring emphasis back to faces by performing a clustering analysis using t-SNE and a silhouette index, concluding that at the level of conv-5, face representations are more clustered compared to other categories. This makes "face tuning ... distinctive from tunings to other object classes" (rebuttal, p. 47). I have two concerns about that, one conceptual, one methodological.

First, the conceptual concern. Why are faces clustered at the conv5 level? The authors suggest that "face images have statistics more readily distinguishable from the statistics of other objects *due to the stronger clustering of the abstracted responses in the DNN* [emphasis mine]" (rebuttal, p. 14). This is circular reasoning. Fundamentally, all a priori image classes are either a) characterized by combinations of specific mid-level features or b) by extra-visual information such as cognitive, top-down signals or anatomical constraints. Since DNNs are feedforward models with no recurrence or anatomical clustering, the answer to why faces are more separable must rely on specific "simple geometric features" (rebuttal, p.15). However, this is an interpretation that the authors work mightily to disprove through their scrambling analyses. They oppose the interpretation that mid-level features make faces special, suggesting that unit selectivity is due to the "global context" of faces (rebuttal, p.9). My first question is — could the authors define what this global context means? Is it the holistic, global configuration of mid-level features? Is it that an eye must appear above a mouth? Or below a head shape? Is it that two eyes and a nose and a mouth become surrounded by a skull?

The team may suggest that this is why they did the PFI analyses – to reveal the global configuration. Indeed, when they perform their visualization procedures using the Gaussian and xdream approaches, they get images of faces back – compellingly *bona fide* faces. Nevertheless, their image generator was trained on face-enriched data. In fact, they used a face generator capable of generating an unlimited number of faces relative to ImageNet categories (FaceGen, per line 494 in main text). This is classic confirmation bias. The point of PFI is that the image generation method is unconstrained so that any consistency in their output is compelling. Faces naturally exist in many ImageNet categories, no FaceGen enrichment was needed. If the authors desired more interpretability, then there are a number of higher quality image generators available pre-trained (e.g. BigGAN trained on 1000 categories).

However, more evidence would be needed. CNNs are broadly susceptible to adversarial attacks, thus one expects that there exists PFIs that activate face units far in excess of that unit's response to faces. The question is whether the excessive activation is due to irrelevant features (high-frequency artifacts) or relevant features (shape/color/texture of image segments). Here, their analysis might be convincing if they present how images changed during the genetic algorithm process. It is necessary to show several images from the first group ("generation"), the generation whose activation matches responses to their preferred class, and to the generation where activation plateaus as well as generations in between. These must be shown for face-selective units (chosen at random) and non-face-selective units (also chosen at random). Further they should show the change in activity for each unit during the genetic process, these may appear in the supplemental.

This is important, especially based on statements like "...the PFIs of units selective to non-face classes show only barely noticeable configurations of each object class." Noticeable to whom or what? Further, the PFI (X-dream version) for faces has a gray background, whereas that of other PFIs do not; one wonders if the "barely noticeable" evaluation is biased by the image generator's re-training with FaceGen. Since we are all humans and subject to pareidolia, without a quantification "noticeable configuration" is not convincing.

Second, the technical concern — about the tSNE analysis. T-distributed Stochastic Neighbor Embedding is best used as a data-visualization technique, not a dimensionality-reduction technique. The problem is that it has two hyperparameters that are easily tuned to get desired results (“exaggeration” and “perplexity”). tSNE has a range of hyperparameter values (i.e., how SI changes as a function of exaggeration and perplexity). These are not reported here. Generally, it is best to draw conclusions from another technique and then use tSNE to visualize those conclusions (see 1). When tSNE is used for drawing conclusions, the best practice is to demonstrate falsifiability: set up an optimization loop to find the exaggeration and perplexity which most contradicts one’s conclusion. Hopefully, any contradictions found are minimal, but if one does find a large contradiction then a scatter plot will evince a trivial explanation (e.g., all points are on a line). A second problem with latent space clustering methods are confounds introduced from the un-naturalness of the stimuli. From looking at the examples shared in their repository several features are apparent, faces are always more-or-less round, they always have a dark spot on top, faces are predominantly light-skinned (what darker-toned faces exist are washed), and have shadowed or darkened, eyes nose and mouth (we would consider these “mid-level features”). Such regularity is not found in the other examples and is not a general property of photos of faces. An attempt was made to control for some basic properties (low-level features: contrast, luminance, etc.), but those properties were chosen based on guesses about what is important and ends up exaggerating the confounds listed above. Since we cannot say exactly what the DNN is computing (due to the large numbers of filters and nonlinearities) there is no circumstance in which we can say a-priori which features must be controlled for. It is best to avoid making claims of this nature, or to use the most diverse stimulus sets feasible.

This brings us to another problem with this analysis. The purpose of the authors’ tSNE analysis is to demonstrate that the preference of single units for other categories (e.g., websites) is of an entirely different nature than the preference of other units for faces. However, tSNE was not performed on ImageNet categories alone. Instead, it was performed on a set of images which does not show uniform characteristics (the faces are cropped and well-segmented, unlike the other categories). ImageNet does have the “ballplayer” category which has faces in most images, so that is a better choice. Furthermore, there are data sets with unaltered photographs of faces which are better choices to use alongside the ImageNet categories which evoked selective responses. To summarize my concerns with the tSNE analysis: Since hyperparameters were not reported, the authors do not show that the latent-space embeddings are robust. Since they did not use a diverse set of naturalistic images, it is unclear that the latent space separation was due to properties of the category itself rather than properties of the hand-picked stimulus set. Because tSNE was not performed on a comparable set of images, there remains doubt that it answers the intended question.

In summary for this first major concern, the key question that should be addressed at the level of the title, introduction, results and discussion is this: why talk about faces and not jack-o’-lanterns? If the answer is that faces are more popular in the literature and more headline-grabbing, then so be it, but it should be clear that the results do not support this selective reporting.

To help – and I honestly hope to help, the manuscript has many interesting analyses - I reduce the above major concern into actionable bullets below.

- a. There must be a strong rationale for framing the manuscript around faces; as it stands, the framing presents itself as a provocative example that might mislead casual readers (see reservoir computing and lottery ticket hypothesis below). Please consider structuring your arguments around lines 159 through 175, which are some of the most comprehensible and credible results.
- b. The terms “low-level”, “mid-level”, and “high-level” require precise definitions as the authors and reviewers seem to be misunderstanding each other (R2Q3). These definitions should be placed in the introduction if those terms are to be used, or better use the precise terms instead.
- c. Please revise the PFI analysis to use unconstrained generators with no targeted face enrichment. Show that PFIs match or exceed the level of activation as preferred images, in a main-text figure. In a supplemental figure show the images and activations from at least four

stages of evolutions: first generation, shortly thereafter, the generation which matches the level of activation, the generation just before activations plateau. Please do this for face-selective units (chosen at random) and non-face-selective units (also chosen at random).

d. Please quantify the structures (dark/light splotch relative locations and extents) in the reverse correlation PFIs and report it in the main text to support claims of differences in configuration patterns. Try to quantify the same for the image generator PFIs.

e. Whatever latent-space embedding method is used, please run the latent-space analysis on all 39 classes for which selectivity exists, feel free to add a category of naturalistic (in the wild) face images. Consider using something other than tSNE that has an objective minimum dimension number (e.g., PCA) and perform the silhouette tests in a moderate dimension space. It will help to use tSNE or multidimensional scaling as an illustration of the conclusions. If the authors still wish to use tSNE, they should run a robustness analysis and report the findings (one sentence in the main text would suffice, if details are in the supplemental). At minimum, they should vary exaggeration and perplexity over a wide range of values (recommending a span from ¼ to 4x the values chosen in the paper). It is appropriate to make a heatmap of the SI values over this range for several image categories and put in the supplemental. For a more convincing analysis, do set up an optimization loop as a challenge test.

f. Rebuttal figure "e" (ImageNet categories selected for) should be in the main text as it is a significant finding. The authors will have room because figure 4 could be removed or combined with figure 5, as discussed in concern 3.

Concern 2: Definition of "unit"

There appears to be a misunderstanding around the way units are defined, given that convolutional neural networks work through weight-sharing. According to line 86 and supplemental table 1, "units" are at different spatial locations of the same filter. This sounds like a misunderstanding of the CNN architecture. Weight sharing is the fundamental feature that enables efficient training of deep architectures. The first layer, conv1, takes an image 227x277 pixels in size and with 3 color channels. The layer then contains 96 "filters", each "filter" is 11x11x3 matrix (X, Y, and Color). However, there are more than 96 "units", in fact there are 290,400 "units". The explanation is that each filter gets copied to 55x55 spatial positions ($55 \times 55 \times 96 = 290400$). **Thus each "unit" is one of 3,025 copies**. Because weight sharing is active across all layers, each of the $2 \times 128 = 256$ filters is copied $13 \times 13 = 169$ times. Thus line 86 says conv5 has "43,264 'units' (= $13 \times 13 \times 256$)". To put it briefly: if a given filter represents a face feature in one spatial location, it should also represent it at every other location. Thus it is problematic to think of face-selective units beyond the channel dimension (in conv5, 256 channels). The manuscript never addresses this basic fact, while the code does not remove weight sharing and the number of weights and biases reported is consistent with a method that still has weight-sharing active. What would it mean if face-selective "filters" are not face selective in all their positions? We can't say whether they are or are not, it is not reported. We can't say whether the analysis is just multiple repetitions of the same unit.

Recommendations:

a. Please clarify the relationship between face selectivity of the same filter at different spatial locations. Let the reader know if/why this makes sense or is surprising. Please quantify how many of the face selective units are repeats of the same filter at different spatial locations (and thus possibly over-counted).

b. Please break down the reporting of face selective units down by filter (AKA "channel") and convolution group (there is grouped convolution) as well as by layer.

Concern 3: Invariance

A related concern to all the issues above (selective emphasis of selectivity, PFI analyses, and

weight sharing) is that of invariance. First, some forms of invariance are a basic property of CNNs, particularly motivating the development of AlexNet (see 2 3 4 5 6). Translation invariance is induced through the simple act of convolution, especially if a filter is smooth (as opposed to a grating). This is because filters have width (e.g., 11x11, 5x5, or 3x3). Not all filters produce translational invariance and the translational invariance due to filtering is only approximate. By coarse-graining (using stride >1) larger invariance width is induced. Max pooling is an exact form of translational invariance that is most impactful for subsequent layers. Translational invariance begets rotational and pose invariance when using a feature detector instead of a more basic filter (e.g. an oriented gradient filter will not become invariant to the rotation of a triangle by simply pooling translations of it, but a corner detector will.). These are the intentions behind the design of CNNs. A fact which has received much study is that CNNs do not work as intended (see 7 8 9). These invariance mechanisms work best when a filter is smooth (e.g. a spatial averaging filter). When a CNN is trained the filters are no longer anything like spatial averages. However, when it is initialized the variation within each filter is unstructured and limited. Thus, random initializations are expected to be more like averaging filters and thus CNNs are expected to be closer to the theoretical limits of invariance.

Another, more interesting form of invariance (which the manuscript addresses) is that an earlier layer can have different filters corresponding to different versions (viewpoints) of the same feature (semi-redundant features). Rather than co-opting translation invariance it is possible that the subsequent layer can combine these inputs. However, unlike pooling and convolution, there was no way to enforce this condition. It is simply hoped that CNNs will "discover" that they can do this during the training process. Unfortunately it is difficult to separate this form of invariance from translation. In a given channel, say one from conv5. It has a 3x3x256 convolution operator. It is possible that one of the 3x3 filters has all 9 values which are much larger than the other "slices". This would be a "pass-through" filter and would implement viewpoint invariance by virtue of its spatial averaging function. These two must be separated.

Given that we expect viewpoint invariance from CNN theory, the viewpoint invariance analysis is largely over-emphasized. It is nice to see that expectations are confirmed but it is framed as an unexpected finding. It is expected for anyone with experience with the CNN architecture and its motivations. However, most of it should be removed to the supplemental. We acknowledge that many in the neuroscientific community may lack this background and would be interested in seeing it. Computer-vision specialists may also find it interesting to see the performance confirmed in this matter (especially with regard to 7). However, it is overstated in its current form and no acknowledgement of its origins in the basic CNN algorithm is made. Thus, rather than educating the neuroscientific community, it may promulgate misconceptions, and members of the computer vision community might regard the lack of context as undermining the paper's credibility. Thus, the section "Invariant face representation of face-selective units in the untrained network" should be reduced to a brief summary to be expounded upon in the supplemental for the curious.

Commendably, one part of the viewpoint analysis stands above the rest. Above, I described how the researchers who developed CNNs hoped that they would combine semi-redundant feature detectors. This is a possibility that researchers left open to the CNN training process, but they had no way of ensuring it actually happened (as opposed to translation-based invariance, which is forced upon all CNNs by coarse-graining). The imposition of translation invariance through coarse graining does not depend on training but combining semi-redundant filters may or may not happen. In Figure 5c, the authors report they found direct evidence that viewpoint-invariant conv5 units receive projections from viewpoint-specific conv4 units (again, using the face-trained image generator). Except they show 4-5 inputs per conv5 unit, yet a given conv5 unit receives inputs from *all* channels in the previous layer (that is part of the weight matrix, which relates all 192 channels in conv4 to each of the 256 channels in conv5). So how were the featured inputs selected for the figure? The authors say they "back-projected," but this is undefined. Was it based on weight magnitude? What about the next five strongest weights? Notably, even a non-selective conv5 unit receives input from these view-specific channels in conv4. How is the influence of those inputs minimized?

Again, we reduce this major concern to the following action points:

- a. The authors should summarize what is known and about invariance in CNNs. They should reduce figure 4 and lines 209 through 252 to a single paragraph summarizing the intrinsic

invariance of convolution (weak invariance) and max-pooling (strong invariance) with a brief summary of how they confirmed this was not destroyed. It should specifically address how the lack of spatial structure in your filters relates to the kinds of spatial structure in a trained AlexNet (one might consult OpenAI Microscope 10 for comparison) and how types of spatial structure in filters can enhance or detract from invariance. Details can be moved to the supplementary.

b. Lines 254 through 268 should be re-written to include a concise description of why the CNN architecture is likely to produce the scenario described on lines 267 and 268, and why trained and untrained networks are different.

c. The authors should elaborate on how they chose filters for display on their figures and how they performed "back-projection." They should make sure to explain how they know that these specific inputs matter more than the others. This should include the context of the Gaussian distribution from which weights are pulled, and the context of weaker inputs, which may all become active with the same image and thus matter more than the few strong connections.

d. The authors should characterize the contributions of weights within the same kernel to weights in different kernels and thus show whether their data evinces convolution within one channel or combinations of semi-redundant channels.

e. Lines 309 through 321 are confusing. The images are padded with fixed integers prior to convolution. This means that variability among pixels along the periphery is less able to contribute to detections (i.e., they are washed out). This gets exacerbated with each subsequent convolution and explains why CNNs show a centrality preference — this is not related to face patches. One could challenge this objection if one of the 39 highly selected for categories showed an edge preference. However, given the convolutional nature of CNNs selectivity in convolutional layers is always expected to show some kind of spatial regularity. Thus, CNNs may not be useful for discussing the spontaneous emergence of patches with respect to the visual field. These lines should probably be removed.

f. Line 285 –286 "We found that the level of invariance increased..." Again, we know a-priori that invariance will increase through hierarchy due to repeated convolution operators. It is supposed to do that because engineers were inspired by the brain to make CNNs that way.

Concern 4: Face selectivity across layers

It is reported that conv2 contains around 60-70% as many face-selective units as deeper layers. The receptive fields of conv2 are about 51 x 51 pixels. All the shown stimuli contain faces that appear to occupy > 65 % of the 227 x 227-pixel image. How can a such a small RF be selective for a face as well as deeper layers given the same RF? Would it give rise to a different face configuration per PFI, or to a face part? These early-layer responses cannot be for whole faces, only for face fragments – which again, the authors actively disregard as an explanation for the face selectivity. Please clarify the PFIs attained by layer, as well as the activations attained during the PFI generation process.

a. Explain whether units (especially in early layers) require whole faces or just face parts. If they require whole face explain how that is possible.

b. Clarify how viewpoint invariant units (agglomerated via "back-projection") relate to face parts vs whole faces.

Concern 5: Statistical tests and reporting

There are no reports of adjustments for multiple comparisons. When looking for selectivity among a small set of classes Bonferroni correction is one option. When looking for single units with a high FSI among hundreds of thousands of units, another method such as step-down minP is appropriate. Correction is necessary because of the lottery-ticket hypothesis the idea is that there is some chance in your extended joint-distribution of gaussians to get a filter with the properties

needed to detect a series of blobs in a roughly face-like spatial arrangement. Since all weights come from the same distribution these serendipitous filters are still in the tail of the distribution, not true outliers, thus a properly executed significance test may not give them very low p-values. Despite their possibly high p-values they are still interesting and consistent with the lottery ticket hypothesis. So, one might seek another criterion for deciding which units are selective or not (such as the receiver-operator characteristic).

Another issue with statistical reporting is that the Mann-Whitney U test seems to be misapplied. Lines 212 and 213 show a Mann-Whitney U test that is used to show "constant face tuning". This is confusing for two reasons: 1) it is not said what has gone into the analysis, this test detects differences between two populations according to a single metric. These populations are not named, the metric is not named. Because the name Mann-Whitney U is often conflated with the Wilcoxon signed-rank test we do not know if they are matched (signed-rank) or un-matched (rank-sum). 2) The Mann-Whitney U test cannot detect similarity. It is inappropriate to say $p > 0.05$ therefore these two populations are the same. All populations are different, one must state the resolution at which their differences matter.

Other miscellaneous statistical reporting problems: What is *P? We found no explanation in the main text. There are no effect sizes. There are many to choose from but the one which is most interpretable and consistent across datasets is the simple difference formula 11, another is the rank-biserial correlation. We also recommend using the term "signed-rank" or "rank-sum" rather than "Mann-Whitney U". Signed-rank and rank-sum are clear terms that describe how the test is conducted in their own names and make it obvious to the reader whether the test is matched or unmatched.

The use of the Kolmogorov-Smirnov test needs to be justified for the face detection task. This test is only designed to tell if two data sets are pulled from the same distribution. It cannot detect if a metric is generally higher for one group than the other (except in cases where the rank-sum test is also diagnostic).

Actionable points:

- a. Please include multiple comparison adjustments or justify for why they are not needed.
- b. Do not use the rank-sum or signed-rank test alone for detecting whether two groups are not distinct from each other. You should use follow up tests (such as KS), a higher P-value threshold (e.g., $P > 0.1$ is more believable) and state the resolution of the test (the statistical power given the population size, or estimate).
- c. Justify the use of the KS test in the face-detection task. (methods or main text)
- d. Report effect sizes alongside all P-values
- e. State exactly what groups and what metrics are used for each reporting of P-values. (supplemental is OK so long as you give some explanation in methods or main text)

Concern 6: Missing important related works

The manuscript is missing an obvious relation to neural encoding paradigms. We regret not mentioning this in our first review and commend the other reviewers for the astute recommendation to invoke the lottery ticket hypothesis as a significant and relevant body of work. However, reservoir computing 12 is more closely related to theories of brain encoding (here is a starting point for your research 13 14 15 16). This is an area of research called "reservoir computing". Two theoretical models of neural computation are "echo state networks" and "liquid state machines". These approaches seek to explain how to quickly train a large, complicated network while resolve some of the ways that populations of spiking neurons behave inconsistently with older concepts of neural code. Arguably, this paper accidentally implements and tests a simple version of a reservoir computer.

a. Accurately place the work in the context of reservoir computing, one or two sentences in the discussion alongside "lottery ticket hypothesis" is fine.

Concern 7: Continued conflation of the terms "innate" and "spontaneous"

There continues to be some misunderstanding between the concepts of "innate" and "spontaneous." Innate is a broader term (one way to be innate is to be spontaneous), another way to be innate is to be programmed by evolution. Proponents of innateness for faces seek to explain why face preferring neurons consistently show up in the same locations across individuals of the same species. Spontaneousness cannot account for that. However, compelling evidence for spontaneous emergence is helpful for the innateness argument as it reduces the burden from explaining how face preferences can exist at all and why in the same locations, to just explaining why it shows such regularity. On this point there is a schism between the introduction, results, and discussion. The introduction is still beleaguering a strong innateness argument whereas the discussion seems to acknowledge that the evidence is incomplete.

a. The introduction and abstract should be rewritten to clarify the two aspects of the innateness argument and to orient the reader as to which aspect the paper has evidence for.

Minor issues

- a. Fig. 3 Analysis: the double-dipping argument from Reviewer #2 was only partially addressed: this information should be included in the Figure legend.
- b. Figure R2Q3 should be in the main text
- c. Line 352, "face-deprive" is missing a d
- d. Supp. Figure 12. y-label - ratio is not a percent (%)
- e. Line 301: face units are "scarcely found in Conv1"; conv1 can only implement linear filters. This comment may suggest to readers that compound feature detectors may exist in conv1 when Conv1 is actually severely limited in what it can do for reasons that are completely known.
- f. Line 13 is ungrammatical. It should read: "across different random initializations of the network" (delete the "conditions of" and pluralize "initialization"). There are other definite article and subject-verb agreement mistakes throughout.

Links:

1. <https://towardsdatascience.com/why-you-are-using-t-sne-wrong-502412aab0c0>
2. <https://towardsdatascience.com/a-short-history-of-convolutional-neural-networks-7032e241c483>
3. <https://en.wikipedia.org/wiki/Neocognitron>
4. <https://leon.bottou.org/publications/pdf/cvpr-2004.pdf>
5. <https://papers.nips.cc/paper/2012/file/c399862d3b9d6b76c8436e924a68c45b-Paper.pdf>
6. https://link.springer.com/chapter/10.1007/978-3-642-33863-2_51
7. <https://arxiv.org/abs/1805.12177>
8. <https://arxiv.org/pdf/1508.01983.pdf>
9. http://cs231n.stanford.edu/reports/2016/pdfs/107_Report.pdf
10. <https://openai.com/blog/microscope/>
11. <https://journals.sagepub.com/doi/pdf/10.2466/11.IT.3.1>
12. <https://www.sciencedirect.com/science/article/pii/S089360800700038X>
13. <https://igi-web.tugraz.at/PDF/248.pdf>
14. <https://www.youtube.com/watch?v=zdDzSonhymE&list=PLgKuh-IKre10qVKXL6EqR08qxyHf8R7-A&index=8>
15. <https://www.youtube.com/watch?v=rWvhd9O6R5Y&list=PLgKuh-IKre10qVKXL6EqR08qxyHf8R7-A&index=9>
16. <https://www.springer.com/gp/book/9789811316869>

Reviewer #4:
None

Response to Reviewers

We sincerely appreciate your helpful and constructive comments. Based on the questions and suggestions provided, we made revisions to address the remaining issues. We are confident that the reviewer's concerns about our work have been fully addressed now.

Here we provide a reply to your concerns to describe in detail how each question and suggestion was addressed in our revised manuscript. Thank you very much for your kind consideration.

Reviewer #1:

It seems to me that the authors have adequately addressed all my concerns.

We sincerely appreciate your consideration and support during the revision process.

Reviewer #2:

I thank and congratulate the authors for their detailed and comprehensive response to the reviews. The authors put an impressive amount of work in the revision and thoroughly addressed all concerns raised. I have only three small remaining comments regarding the revision:

a. The authors added a new condition to the manuscript which was “face-deprived”. Based on my own experience in the past, it is tremendously difficult to remove all the faces from ImageNet. The authors don't give any more details as to how the faces were removed. It is only noted in the methods that 500 classes of the ILSVRC2021 dataset were selected. Did the authors additionally remove faces from these categories using face detection algorithms, or did they manually curate the dataset? In any case, if they cannot be 99.9% certain that there are no more faces in the dataset, this condition should be termed “face-reduced” to make clear that it is possible that there are still faces in the dataset. Also please add more details to the methods.

We appreciate your comment on this issue. The current dataset was curated according to our visual inspection, not with an automated face detection algorithm — we manually selected 500 classes in which no noticeable faces in any images could be found. Considering that there may still be faces remaining in the dataset, although a very small number, we agree that our control dataset should have been termed “face-reduced” instead of “face-deprived.” In the revised manuscript, we corrected our terms and provided details to the methods as to how our “face-reduced” dataset was selected.

Revised text (Results, Line 250-252)

To investigate the effect of training on a face image set, we prepared the following three different stimulus sets: (1) face-reduced ImageNet: 500 classes including no recognizable face images were manually curated from the ILSVRC 2010 dataset according to a visual inspection by the authors.

b. The authors' addressed my concern regarding the generalization of face-selective units (Q2) by testing several other datasets. The results show that face-selective responses indeed generalize to novel sets of faces. However, in the revised text the authors conclude "...face-selective units in an untrained network are selective to general face images...". This is misleading and not the point. Instead, the results suggest that the face-selectivity of units defined by one specific dataset generalize to other datasets. Please correct.

We thank you for this careful correction. In the revised text, we corrected our conclusion to indicate that our results suggest that face-selective units defined by one specific dataset demonstrate consistent face-selectivity to other novel datasets.

Revised text (Results, Line 147-148)

These results suggest that the observed face-selectivity in an untrained network defined by one specific dataset can be generalized to other novel sets of faces.

c. The term "innate face-selective units" is confusing and incorrect. In fact, it assumes that an untrained CNN represents a newborns' brain at birth. We don't know that and in fact there is a strong debate whether training CNNs models phylogeny or ontogeny (e.g., Hasson et al., Neuron, 2020). Please remove the term "innate" when describing face selectivity or face-selective units in untrained CNNs.

We regret that we previously used misleading terms such as "innate" or "spontaneous." We removed these confusing terms throughout our revised manuscript.

Reviewer #3:

We applaud the authors for their responsiveness and comprehensive new set of analyses. They clearly showed diligence and hard work in considering our and other reviewers' concerns. We thank you for your efforts.

While the manuscript is improved, it is also apparent where the authors stopped short of pushing their analyses in interesting directions, as it would radically affect the framing of the overall story. This remains the key problem — with the authors' latest batch of results, the current framing becomes massively misleading, and there are facts about convolutional networks that require description as part of the Introduction and which may affect the novelty of the results.

Major concerns.

Concern 1: unmerited elevation of faces as a category with special properties

Per the introduction, discussion and references, this manuscript centers around animals and face selectivity. The results show that in untrained networks, units can show selectivity to some visual categories — as long as such categories have simple geometric features. The authors find dozens of such categories, including pumpkins, flowers, household tools and yes – faces, too. Given this compelling analysis, I reaffirm the common concern of Reviewer #2 (Q3), Reviewer #4 (Q5) and my own: with the discovery of dozens of new visual categories that elicit selectivity properties as those of face-selective units, the premise of the paper is misleadingly narrow. It would be as if a neurologist described a post-stroke patient's inability to move her arm as an inability to play the piano. True enough, but also a puzzlingly arbitrary underreporting of the facts.

So, are faces special or not? Are they worth of the headline or not? This is not clear. After reporting that units in untrained networks can develop selectivity to dozens other pre-defined semantic categories, the authors try to bring emphasis back to faces by performing a clustering analysis using t-SNE and a silhouette index, concluding that at the level of conv-5, face representations are more clustered compared to other categories. This makes “face tuning ... distinctive from tunings to other object classes” (rebuttal, p. 47). I have two concerns about that, one conceptual, one methodological.

First, the conceptual concern. Why are faces clustered at the conv5 level? The authors suggest that “face images have statistics more readily distinguishable from the statistics of other objects *due to the stronger clustering of the abstracted responses in the DNN* [emphasis mine]” (rebuttal, p. 14). This is circular reasoning. Fundamentally, all a priori image classes are either a) characterized by combinations of specific mid-level features or b) by extra-visual information such as cognitive, top-down signals or anatomical constraints. Since DNNs are feedforward models with no recurrence or anatomical clustering, the answer to why faces are more separable must rely on specific “simple geometric features” (rebuttal, p.15). However, this is an interpretation that the authors work mightily to disprove through their scrambling analyses. They oppose the interpretation that mid-level features make faces special, suggesting that unit selectivity is due to the “global context” of faces (rebuttal, p.9). My first question is — could the authors define what this global context means? Is it the holistic, global configuration of mid-level features? Is it that an eye must appear above a mouth? Or below a head shape? Is it that two eyes and a nose and a mouth become surrounded by a skull?

The team may suggest that this is why they did the PFI analyses – to reveal the global configuration. Indeed, when they perform their visualization procedures using the Gaussian and xdream approaches, they get images of faces back –

compellingly *bona fide* faces. Nevertheless, their image generator was trained on face-enriched data. In fact, they used a face generator capable of generating an unlimited number of faces relative to ImageNet categories (FaceGen, per line 494 in main text). This is classic confirmation bias. The point of PFI is that the image generation method is unconstrained so that any consistency in their output is compelling. Faces naturally exist in many ImageNet categories, no FaceGen enrichment was needed. If the authors desired more interpretability, then there are a number of higher quality image generators available pre-trained (e.g. BigGAN trained on 1000 categories).

However, more evidence would be needed. CNNs are broadly susceptible to adversarial attacks, thus one expects that there exists PFIs that activate face units far in excess of that unit's response to faces. The question is whether the excessive activation is due to irrelevant features (high-frequency artifacts) or relevant features (shape/color/texture of image segments). Here, their analysis might be convincing if they present how images changed during the genetic algorithm process. It is necessary to show several images from the first group ("generation"), the generation whose activation matches responses to their preferred class, and to the generation where activation plateaus as well as generations in between. These must be shown for face-selective units (chosen at random) and non-face-selective units (also chosen at random). Further they should show the change in activity for each unit during the genetic process, these may appear in the supplemental.

This is important, especially based on statements like "...the PFIs of units selective to non-face classes show only barely noticeable configurations of each object class." Noticeable to whom or what? Further, the PFI (X-dream version) for faces has a gray background, whereas that of other PFIs do not; one wonders if the "barely noticeable" evaluation is biased by the image generator's re-training with FaceGen. Since we are all humans and subject to pareidolia, without a quantification "noticeable configuration" is not convincing.

Second, the technical concern — about the tSNE analysis. T-distributed Stochastic Neighbor Embedding is best used as a data-visualization technique, not a dimensionality-reduction technique. The problem is that it has two hyperparameters that are easily tuned to get desired results ("exaggeration" and "perplexity"). tSNE has a range of hyperparameter values (i.e., how SI changes as a function of exaggeration and perplexity). These are not reported here. Generally, it is best to draw conclusions from another technique and then use tSNE to visualize those conclusions (see 1). When tSNE is used for drawing conclusions, the best practice is to demonstrate falsifiability: set up an optimization loop to find the exaggeration and perplexity which most contradicts one's conclusion. Hopefully, any contradictions found are minimal, but if one does find a large contradiction then a scatter plot will evince a trivial explanation (e.g., all points are on a line). A second problem with latent space clustering methods are confounds introduced from the un-naturalness of the stimuli. From looking at the examples shared in their repository several features are apparent, faces are always more-or-less round, they always have a dark spot on top, faces are predominantly light-skinned (what darker-toned faces exist are washed), and have shadowed or darkened, eyes nose and mouth (we would consider these "mid-level features"). Such regularity is not found in the other examples and is not a general property of photos of faces. An attempt was made to control for some basic properties (low-level features: contrast, luminance, etc.), but those properties were chosen based on guesses about what is important and ends up exaggerating the confounds listed above. Since we cannot say exactly what the DNN is computing (due to the large numbers of filters and nonlinearities) there is no circumstance in which we can say a-priori which features must be controlled for. It is best to avoid making claims of this nature, or to use the most diverse stimulus sets feasible.

This brings us to another problem with this analysis. The purpose of the authors' tSNE analysis is to demonstrate that the preference of single units for other categories (e.g., websites) is of an entirely different nature than the preference of other units for faces. However, tSNE was not performed on ImageNet categories alone. Instead, it was performed on a set of images which does not show uniform characteristics (the faces are cropped and well-segmented, unlike the other categories). ImageNet does have the "ballplayer" category which has faces in most images, so that is a better choice. Furthermore, there are data sets with unaltered photographs of faces which are better choices to use alongside the ImageNet categories which evoked selective responses. To summarize my concerns with the tSNE analysis: Since hyperparameters were not reported, the authors do not show that the latent-space embeddings are robust. Since they did not use a diverse set of naturalistic images, it is unclear that the latent space separation was due to properties of the category itself rather than properties of the hand-picked stimulus set. Because tSNE was not performed on a comparable set of images, there remains doubt that it answers the intended question.

In summary for this first major concern, the key question that should be addressed at the level of the title, introduction, results and discussion is this: why talk about faces and not jack-o'-lanterns? If the answer is that faces are more popular in the literature and more headline-grabbing, then so be it, but it should be clear that the results do not support this selective reporting.

To help – and I honestly hope to help, the manuscript has many interesting analyses - I reduce the above major concern into actionable bullets below.

a. There must be a strong rationale for framing the manuscript around faces; as it stands, the framing presents itself as a provocative example that might mislead casual readers (see reservoir computing and lottery ticket hypothesis below). Please consider structuring your arguments around lines 159 through 175, which are some of the most comprehensible and credible results.

We sincerely appreciate your detailed comments on this important issue. Indeed, we have considered this issue very carefully and even considered rewriting the whole manuscript to expand the scope of the study from face-selectivity to general object selectivity in untrained networks. However, we finally decided to focus on face-selectivity in the current manuscript, though we will prepare a separate, follow-up manuscript in which we will explicitly discuss the origin of the selective responses to general objects in untrained networks. Details of our rationale on this issue are presented below.

Face-selectivity in animals has been considered as a special function of the brain, which is distinct from other object selectivity, as a face is a special class of object for visual recognition related to survival. Thus, it has been a topic of strong interest in neuroscience studies (Gauthier et al., 2000; Kanwisher, 2000; Kanwisher et al., 1997, 1998; Rhodes et al., 2004; Tarr and Gauthier, 2000; Tsao and Livingstone, 2008). Our current result can shed new light on this issue and may ignite relevant discussions in fields related to neuroscience. Although face-selectivity has been considered as very important in the neuroscience field in relation to the social behaviors of higher mammals, there remain important questions to answer, particularly with regard to its origin. On this issue, our revised manuscript will focus on face-selectivity, but it is thoroughly reframed to discuss the above question by demonstrating that selectivity to various objects can arise in untrained networks. In summary, our rationale for focusing on face-selectivity in the current manuscript is summarized below.

1. “Face-selectivity” of the brain has been considered a special topic in neuroscience.

In the history of neuroscience, face-selectivity has been considered as a special function, and it has been studied extensively. For example, a recent finding of a distinct face patch area in the IT implied that face detection is a specialized function of the brain in social animals. With this notion, in the current manuscript, we will focus on our findings relevant to face-selectivity, but we will also introduce our own view that face-selectivity may be a type of neural selectivity to one of various visual objects that arises innately. Then, we will discuss this further issue by expanding the current story to a broader and more general scope of innate object selectivity.

2. We plan to study the “innate visual selectivity to general objects” in a separate follow-up paper.

Because our current finding is important, we are, in fact, preparing a separate paper in which “general object detection” in an untrained network would be thoroughly studied. Currently, we have a number of meaningful findings that cannot be summarized altogether in a single paper. Thus, we would like to investigate this issue intensively in a subsequent study.

3. Currently available experimental data are mostly about face-selectivity.

Because face-selectivity has been of great interest in neuroscience, there are experimental observations pertaining to this function available for comparison with our simulation results. The comparison between biological data and our model results enables us to examine the similarities and differences between them such that our model may provide a possible scenario of biological principles.

In the revised manuscript, we demonstrate our new finding that selectivity to a number of objects can arise in a random network, simply due to their structural simplicity, and discuss how face-selectivity can emerge in the same way and thus may not be a special function of the brain, as previously considered.

Previously, we have indeed been trying to show that there are "special characteristics" of faces distinguished from those of other objects. However, from the additional analyses suggested by the reviewer, we noted that it is not reasonable to claim that faces are special. Therefore, in the revised manuscript, we removed all claims such as “faces are special compared to other objects,” and we decided to report objectively that selectivity occurs for various objects with a simple structure in random networks and this is also true with regard to selectivity for faces. We then discuss that this might be how face-selectivity arises in infant animals without visual experience and that selectivity for other objects can also arise in infant animals, both of which need to be studied further in animal experiments.

Revised text (Abstract, Line 8-17)

Face-selective neurons are observed in the primate visual pathway and are considered as the basis of face detection in the brain. However, it has been debated as to whether this neuronal selectivity can arise innately or whether it requires training from visual experience. Here, using a hierarchical deep neural network model of the ventral visual stream, we suggest a novel mechanism in which face-selectivity arises in the complete absence of training. We found that units selective to faces emerge robustly in random initialized networks and these units

reproduce the characteristics observed in monkeys. This innate selectivity also enables the untrained network to perform face-detection tasks. Intriguingly, we observed that units selective to various non-face objects can also arise innately in untrained networks. Our results imply that the random feedforward connections in early, untrained deep neural networks may be sufficient for initializing primitive visual selectivity.

Revised text (Introduction, Line 19-86)

The ability to identify and recognize faces is a crucial function for social behavior, and this ability is thought to originate from neuronal tuning at the single or multi-neuronal level¹⁻²⁰. Neurons that selectively respond to faces (face-selective neurons) are observed in various species²¹⁻²³, and they have been considered as the building blocks of face detection¹⁸. The observation of this type of intriguing neuronal tuning in the brain has inspired neuroscientists, raising important questions about its developmental mechanism — whether face-selective neurons can arise innately in the brain or require visual experience, and whether neuronal tuning to faces is a “special” type of function distinctive from tunings to other visual objects.

Regarding the emergence of neuronal face-selectivity, previous studies have suggested a scenario in which visual experience develops face-selective neurons²⁴⁻²⁶. The experience-dependent characteristics of face-selective neurons imply that visual experience plays a critical role in developing face-selectivity in the brain. It was observed that the preferred feature images of face-selective neurons in adult monkeys are those that resemble animals or familiar people depending on individual experiences²⁵. Another study of the inferior temporal cortex (IT) in monkeys reported that robust tuning of face-selective neurons is not observed until one year after birth⁶ and that face-selectivity relies on experience during the early infant years. It was also reported that monkeys raised without face exposure did not develop normal face-selective domains²⁶. However, another view suggests that face-selectivity can innately arise without visual experience²⁷⁻³⁴. Although visual experience is critical for refining the development of face-selective neurons, several lines of research have demonstrated that primitive face-selectivity is observed even before visual experience²⁷⁻³¹. Primate infants behaviorally prefer to look at face-like objects as opposed to non-face objects³²⁻³⁴, implying that face-encoding units may already exist in infants. Moreover, visual category-selective domains, including the face, are observed in the ventral visual stream in adult humans who have been blind since birth^{28,29}. Furthermore, a recent study reported that face-selective neurons are observed in infant animals and that the spatial organization of such early face-selective regions appeared similar to that observed in adults⁶. These results taken together imply that face-selective neurons can arise before visual experience, in contradiction to the first scenario.

There has been another important debate as to whether face-selectivity is a special type of visual function distinguished from other processes of object recognition, the developmental mechanism of which needs to be considered and examined distinctively from other visual neural tunings. After early observations of the face-selective responses of single neurons in the IT, face detection has been considered one of the most important visual functions necessary for the survival of social animals³⁵⁻³⁸. Observation of the fusiform face area (FFA), which is specialized for face-recognition, also reinforced the idea that face-selectivity is a specialized neuronal tuning, which may develop differentially from cognition of other general visual objects^{12,39-42}. However, more recent studies have

reported that selectivity to objects such as a car or a bird can also develop in the FFA from visual experience^{43,44}, implying that faces may not be a special, distinct type of object class for visual function and that neuronal tuning to various visual objects can also arise similarly to selectivity to faces.

The argument concerning these issues, which reveals our incomplete understanding of face-selectivity, likely stems from limitations regarding the control of the experimental conditions, as it is impossible to control the amount of visual experience for a particular category, such as face, in individual subjects. Even if the subjects are visually deprived such that they are prevented from having visual experience, the portion of category-selective neurons and their degree of tuning may vary across subjects and cannot easily be predicted. These various factors make it difficult to investigate the developmental mechanism of face- and other object-selective neurons in the brain.

A model study using biologically inspired artificial neural networks, such as deep neural networks (DNNs)^{45,46}, may offer an effective approach to the problem in this case⁴⁷⁻⁵⁰. Recently, DNNs, a stack of biologically inspired feedforward projections with a linear-nonlinear neural motif, have provided insight into the underlying mechanisms of brain functions, particularly with regard to the development of various functions for visual perception^{47,48,51}. For example, a recent model study reported that the neural response of the monkey IT cortex could not only be predicted by the responses of DNNs trained to natural images^{47,48} but could also be controlled by the preferred feature image generated by the DNN model⁵². Notably, previous studies using random hierarchical networks provide important clues about the origin of innate face-selectivity in untrained neural networks. It was reported that untrained feedforward networks can initiate various cognitive functions with random weights and that a random network can perform image classification tasks in that way as well⁵³⁻⁵⁶. It was also reported that a randomly initialized convolutional neural network could reconstruct corrupted images without any training, which implies that a random network can provide a priori information about the low-level statistics in natural images⁵⁷. Overall, such observations suggest the possibility of the emergence of innate cognitive functions, such as primitive face-selectivity in untrained, random hierarchical networks. However, the details of how this innate function emerges in untrained neural networks are not yet understood.

Herein, we show that face-selective units (model neurons) can arise in completely untrained hierarchical neural networks. Using AlexNet⁴⁵, a model reproducing the structure of the ventral stream of the visual cortex, we found that face-selectivity can emerge robustly across different conditions of randomly initialized deep neural networks. We found that their face-selectivity indices (FSI) are comparable to those observed with face-selective neurons in the brain. The preferred feature images obtained from the reverse-correlation method and the generative adversarial network show that face-selective units are selective for a face-like configuration, distinct from units with no selectivity. Furthermore, we found that face-selective units enable the network to perform face detection. Intriguingly, we found that units selective to various non-face objects can also arise innately in untrained neural networks, implying that face-selectivity may not be a special type of visual tuning and that selectivity to various object classes can arise innately in untrained deep neural networks, spontaneously from random feedforward wirings. Overall, our results imply a possible scenario in which the random feedforward connections that develop in early, untrained networks may be sufficient for initializing primitive face-selectivity as well as selectivity to other visual objects in general.

b. The terms “low-level”, “mid-level”, and “high-level” require precise definitions as the authors and reviewers seem to be misunderstanding each other (R2Q3). These definitions should be placed in the introduction if those terms are to be used, or better use the precise terms instead.

We appreciate this timely comment about using these ambiguous descriptions. Throughout the revised manuscript, we replaced the terms with more direct and precise descriptions, as follows: “low-level” as “luminance,” “contrast,” and other low-level parameters; “mid-level” as “local face components;” and “high-level” as categorical label information (e.g. whole face).

Revised text (Results, Line 127-140)

One possible scenario for the emergence of such face-selectivity in random networks is that the observed face-selective units are simply sensitive for local face parts common to facial images. To investigate this possibility, we measured the responses of the face-selective units to a local feature of the face using two types of control images in which global face features are disrupted but local face features are preserved. These were (1) “scrambled” faces, in which small patches of the local face components were spatially scrambled, and (2) “texform” faces⁶¹, in which global face features are disrupted but the statistics of the local face texture is preserved (**Fig. 1g**, left). We confirmed that face-selective units show significantly higher responses to the original face images compared to the corresponding control images (**Fig. 1g**, right; Rank-sum test, Face vs. Scrambled face, $n = 200$, $*p = 1.7 \times 10^{-52}$; Face vs. Texform face, $n = 100$, $*p = 4.1 \times 10^{-30}$). In addition, the responses of face units to these control images were not greater than those to other non-face images, implying that face-selective units are selective to the global context of faces instead of the local components (**Fig. 1g**, right; Rank-sum test, Scrambled face vs. Non-face, $n = 200$, n.s., $p = 1$; Texform face vs. Non-face, $n = 100$, n.s., $p = 0.94$). These results suggest that the observed face units in the untrained network are not particularly selective to local face parts, but are instead selective to a whole face.

c. Please revise the PFI analysis to use unconstrained generators with no targeted face enrichment. Show that PFIs match or exceed the level of activation as preferred images, in a main-text figure. In a supplemental figure show the images and activations from at least four stages of evolutions: first generation, shortly thereafter, the generation which matches the level of activation, the generation just before activations plateau. Please do this for face-selective units (chosen at random) and non-face-selective units (also chosen at random).

We appreciate your detailed comment. As suggested, we revised the PFI analysis using unconstrained generators. In short, from the revised PFI analysis using an unconstrained GAN retrained with a face-reduced dataset, we confirmed the following:

1. The revised PFIs of face-selective units show a face-like configuration with a face contour and local face components such as the eyes and mouth.
2. The activity of face units induced by the revised PFIs exceeds the level of activation induced by the original face stimulus images.

First, to generate the PFI with the unconstrained GAN, we removed all FaceGen images from the dataset. In addition, we manually removed all face images existing in the original ImageNet dataset: **(a)** (“face-reduced” dataset — see our answers to Reviewer 2 question a). Then we generated PFIs of face-selective and non-face-selective units using a GAN trained with this face-reduced dataset. These PFIs were generated using a genetic algorithm, as previously noted. As a result, we observed that the PFI of face-selective units still shows a face-like configuration with a face contour and local face components such as the eyes and mouth **(b, left)**, although the original ImageNet has no face class or images containing noticeable faces (For details of the quantification of a face configuration, see our response to Concern 1 Question d). We also found that the PFIs of units selective to non-face objects show a preferred configuration, similar to the images of each object class **(b, right)**.

Next, to test whether the activation of the target units by PFIs can match or even exceed the level of activation by the original stimulus images, we tracked the changes of the PFI during the evolution process — the response of the target units induced by the PFI in each iteration step was measured. First, we confirmed that the initial random images (iteration = 0) gradually change to images with a configuration of each preferred object class as evolution progresses **(c-d)**. We observed that the responses of target units induced by the PFI increase, being higher than those induced by face stimulus images, as the iteration number of the genetic algorithm exceeds a certain value (iteration > 15). We confirmed that the responses of face-selective units induced by PFI in the final iteration were significantly higher than those induced by any face stimulus used for face unit selection **(e, Rank-sum test, $n_{\text{unit}} = 465$, $*p = 8.59 \times 10^{-50}$)** and were also higher than those induced by any non-face stimulus **(e, Rank-sum test, $n_{\text{unit}} = 465$, $**p = 4.06 \times 10^{-74}$)**. These results indicate that our method successfully finds the preferred input feature of face-selective units, even when the GAN was not trained with face images.

In the analysis of non-face-selective units, similar results were also obtained — e.g., flower-selective units **(f, Rank-sum test, $n_{\text{unit}} = 107$, $*p = 5.64 \times 10^{-11}$, $**p = 2.08 \times 10^{-13}$)**. These results show that our genetic algorithm successfully extracts the preferred feature image for both units selective to face and non-face objects.

Revised text (Methods, Line 467-468)

The GAN (**Supplementary Table 2**) was trained with natural images (ILSVRC 2010) in which images with faces were removed.

Revised text (Results, Line 158-174)

Next, to characterize the feature-selective responses of these face-selective units qualitatively, we reconstructed preferred feature images (PFI) of individual units using a reverse-correlation method⁶³ and a generative adversarial network algorithm (X-Dream)²⁵ (see Methods). In the reverse-correlation analysis, we presented 2,500 images of bright and dark 2D Gaussian filters at random positions as input stimuli to an untrained network (**Fig. 2a**). By adding stimuli weighted according to the corresponding neural response with 100 repeated iterations, we obtained the preferred feature images of the target units. In the X-Dream analysis, a deep generative adversarial neural network⁶⁴ was trained to the face-reduced ImageNet datasets to synthesize preferred feature images from the image codes scored by a genetic algorithm using the responses of units from repeated iterations (**Fig. 2b**). We found that the response of target units induced by the PFI was increased, being higher than those induced by face stimulus images, as the iteration number of the genetic algorithm exceeds a certain value (**Figs. 2a-b**, right). These results indicate that our PFI generation method successfully finds the most preferred input feature of face-selective units. As a result, using both methods, we obtained the PFI of face-selective units, units selective to non-face classes, and units without selective responses to any image classes (**Fig. 2c**, **Supplementary Fig. 4**). We found that the PFI of the face-selective units presents distinguishable features from those of other objects. We observed that the PFIs of face-selective units represent face-like configurations, whereas the PFIs of units selective to non-face classes show only barely noticeable configurations of each object class (only except for a couple of cases, i.e., flowers in a reverse correlation and hands in X-Dream).

d. Please quantify the structures (dark/light splotch relative locations and extents) in the reverse correlation PFIs and report it in the main text to support claims of differences in configuration patterns. Try to quantify the same for the image generator PFIs.

We appreciate this suggestion. To quantify the structures of the PFIs, we defined a “face-configuration index” as the pixel-wise correlation between the original face stimuli and the generated PFIs. From an additional analysis using this index, we confirmed that both of the PFIs of face-selective units generated by the reverse correlation and the X-Dream algorithm show a significantly higher face-configuration index (i.e., a higher correlation to face images) compared to all of the other non-face object classes. Details of the analysis are described below.

To quantify the structural similarity of the PFIs to face images, we defined the face-configuration index as the averaged pixel-wise correlation between a PFI and the 200 face images used for face unit selection (**a**). We estimated the face-configuration index of each PFI of face-selective and non-face-selective units generated by the reverse-correlation method. As a result, we found that the estimated PFIs of face-selective units (with a visually observable face-like configuration) have significantly higher average values of the index compared to the control images, as measured from pixel-wise randomized PFIs (**b**, Rank-sum test, $n_{\text{Face}} = 465$, $*p = 9.42 \times 10^{-86}$). On the other hand, the indices of PFIs measured from non-face-selective units did not differ from the control images (**b**, Rank-sum test, $n_{\text{Hand}} = 7$, $n_{\text{Horn}} = 772$, $n_{\text{Flower}} = 107$, $n_{\text{Chair}} = 63$, n.s., $p > 0.19$). Accordingly, the estimated PFIs of face-selective units have a significantly higher average value of the index than that from the units selective to non-face objects (**b**, Rank-sum test, $n_{\text{Face}} = 465$, $n_{\text{Hand}} = 7$, $n_{\text{Horn}} = 772$, $n_{\text{Flower}} = 107$, $n_{\text{Chair}} = 63$, $**p < 5.63 \times 10^{-4}$). Notably, the average pairwise correlation estimated between each face stimulus image (**b**, Face stimulus) was nearly zero, significantly lower than the average face-configuration index of the PFIs of face-selective units. This implies that the observed index of face-selective units reflects the structural similarity to the abstracted (or a prototype) feature of face stimulus images rather than similarly to particular face images accidentally observable (**b**).

We repeated the same analysis for PFIs generated by the X-Dream method and obtained similar results (**c**, Rank-sum test, $n_{\text{Face}} = 465$, $n_{\text{Hand}} = 7$, $n_{\text{Horn}} = 772$, $n_{\text{Flower}} = 107$, $n_{\text{Chair}} = 63$, $**p < 2.70 \times 10^{-2}$). We added these results to Figure 2 in the revised manuscript.

Revised text (Results, Line 175-184).

To quantify the structural similarity of the PFIs to face images, we defined the face-configuration index as the averaged pixel-wise correlation between a PFI and the 200 face images used for face unit selection (**Fig. 2d**). We

estimated the face-configuration index of each PFI of face-selective and non-face-selective units generated by the reverse-correlation method. We found that the estimated PFIs of face-selective units (with a visually observable face-like configuration) have a significantly higher average value of the index than that from the units selective to non-face objects (**Figs. 2e, f**, left, Rank-sum test, Face PFI vs. Non-face PFI, $n_{\text{Face}} = 465$, $n_{\text{Hand}} = 7$, $n_{\text{Horn}} = 772$, $n_{\text{Flower}} = 107$, $n_{\text{Chair}} = 63$, $**p < 2.7 \times 10^{-2}$). Notably, the average pairwise correlation estimated between each face stimulus image shows a significantly lower value of nearly zero, implying that the observed index of face-selective units reflects structural similarity to the averaged (or a prototype, abstract) face image rather than similarly to particular face images accidentally observable.

Revised text (Fig.2 caption, Line 762-765)

d. Illustration of the face-configuration index (FCI) of a face unit's PFI. The FCI was defined as the pixel-wise correlation between the original face stimuli and the generated PFIs. **e.** FCI of PFI, using the reverse correlation method, of units selective to each class. **f.** FCI of PFI, using X-Dream, of units selective to each class.

e. Whatever latent-space embedding method is used, please run the latent-space analysis on all 39 classes for which selectivity exists, feel free to add a category of naturalistic (in the wild) face images. Consider using something other than tSNE that has an objective minimum dimension number (e.g., PCA) and perform the silhouette tests in a moderate dimension space. It will help to use tSNE or multidimensional scaling as an illustration of the conclusions. If the authors still wish to use tSNE, they should run a robustness analysis and report the findings (one sentence in the main text would suffice, if details are in the supplemental). At minimum, they should vary exaggeration and perplexity over a wide range of values (recommending a span from $\frac{1}{4}$ to 4x the values chosen in the paper). It is appropriate to make a heatmap of the SI values over this range for several image categories and put in the supplemental. For a more convincing analysis, do set up an optimization loop as a challenge test.

Thank you for this timely suggestion. In the revised manuscript, we conducted a latent-space analysis of all 39 classes for which selective units are observed. We also conducted the same analysis for a new additional category

of naturalistic face images (a total of 40 classes). As suggested for the latent-space embedding method, we utilized two different approaches: (i) a principal component analysis (PCA) and (ii) t-distributed stochastic neighbor embedding (t-SNE) with variation of the exaggeration and perplexity parameters.

Under these conditions, we confirmed our previous findings:

1. Selective units were still observed for 40 classes, including the category of naturalistic face images.
2. The clustering level of the Conv5 response in the PCA space appeared to be correlated with the number of observed selective units for each class, as previously shown by the t-SNE method.
3. The observed results in the t-SNE analysis were consistent under variation of the exaggeration and perplexity parameters.

Details of the analysis are described below.

1. Selective units were still observed for 40 classes, including the category of naturalistic face images.

First, to avoid unexpected bias from controlled face images, we replaced the similarity-controlled face stimulus with naturalistic face images adopted from VGG face2 and measured the responses of untrained networks to all ImageNet classes and naturalistic face images (a). We found units selective to 39 classes among the 1,000 classes, identical to the results in the previous manuscript, and found units selective to naturalistic face images (b).

2. The clustering level of the Conv5 response in the PCA space appeared to be correlated with the number of observed units selective for each class, as previously shown by the t-SNE method.

We then performed the principal component analysis (PCA) on the responses of the Conv5 units. We demonstrated the result using the first and second principal components in the latent space (c, explained variance, $PC_{1st} = 18.1\%$, $PC_{2nd} = 14.3\%$). The Silhouette index (SI) was estimated to quantify the clustering level in the latent space, and we observed that the value of SI measured in the PCA latent space was correlated with the number of observed selective units, similar to the result achieved in the previous t-SNE analysis (d-e).

Subsequently, we compared results of the PCA analysis with those from the t-SNE analysis. We repeated the latent space analysis using the t-SNE method and estimated the SI. The hyperparameters were set to 30 for perplexity and 4 for exaggeration, which are the default values in the MATLAB built-in function. We found that the values of SI from 40 classes in the t-SNE space was fairly consistent with those estimated in the PCA space (f-g, inset, $n_{\text{Net}} = 50$, $r = 0.52$, $p = 6.72 \times 10^{-4}$) and were also correlated with the number of observed selective units (h, $n_{\text{Net}} = 50$, $r = 0.34$, $p = 0.03$), as observed in the previous manuscript.

Please note that we decided to replace the t-SNE analysis in the previous figure with our new PCA analysis, as the two results are consistent and the PCA results are free from any dependency on hyperparameters.

3. The observed results in the t-SNE analysis were consistent under variation of the exaggeration and perplexity parameters.

Additionally, to test whether our previous results gained when using the t-SNE analysis are robust under variation of the hyperparameters, we measured the SI in a wide range of the parameter space from $\frac{1}{4}$ to 4 fold of the two hyperparameter values (perplexity and exaggeration) originally chosen (i). We varied the two parameters from their initial values — 30 for perplexity and 4 for exaggeration, the default values in the MATLAB built-in function, within the range suggested previously regarding robustness of the method (Maaten and Hinton, 2008).

We then estimated the correlation between the SI values in the PCA method and those measured from the t-SNE method under variation of the two hyperparameters (j, k). As a result, we confirmed that the observed SI values measured from each object class using the PCA and t-SNE methods showed a robust relationship such that the relative order of the SI value measured from the t-SNE method remained fairly consistent with those measured from the PCA method under variation of the hyperparameters in all conditions tested here (l, $n_{\text{Net}} = 50$, $0.48 < r < 0.63$, $* p < 10^{-3}$). In addition, the correlation between SI and the number of observed selective units was observed to be consistent in all hyperparameter spaces tested here (m-o, $n_{\text{Net}} = 50$, $0.29 < r < 0.54$, $* p < 0.01$). Taken together, these results suggest that our observations are not biased by a particular dimensional reduction method or dependent on the choice of hyperparameter sets.

Revised text (Results, Line 291-303)

From this result, we hypothesized that objects with a simple configuration, such as faces, can induce stronger clustering in the latent space representation than those of other object classes and therefore may have a greater likelihood of generating units selective to them. To validate our hypothesis, we used the dimension-reduction method (PCA⁷⁷) to compare a clustered representation of each object class in terms of the raw pixel values and in the responses of Conv5. For quantification of the representational clustering of each class, we measured the Silhouette index⁷⁸ to estimate the consistency of data clustering. We found that there is a strong correlation between the Silhouette index in the Conv5 latent space and the number of selective units observed (**Figs. 5e-g**, Pearson correlation coefficient, $n_{\text{Net}} = 50$, $r = 0.58$, $p = 7.5 \times 10^{-5}$). This result demonstrates that objects with a simple profile, readily distinguishable from those of other objects statistically, lead to a strong clustering of abstracted responses in the DNN and are more likely to generate units selective to it. Thus, this implies that the observed face-selectivity may not be a special case of tuning, whereas selectivity to other visual objects can also arise in random networks simply due to the relatively simple configuration of the corresponding geometric components.

Revised text (Method, Line 436-437)

To visualize the network responses to the low-level feature-controlled stimulus set, a principal component analysis⁷⁷ was used for dimension reduction.

f. Rebuttal figure “e” (ImageNet categories selected for) should be in the main text as it is a significant finding. The authors will have room because figure 4 could be removed or combined with figure 5, as discussed in concern 3.

Thank you for this important comment. As suggested, we included this result in the revised Figure 5 and now indicate that units selective to non-face object categories also arise in untrained networks. Accordingly, in the revised text, we described our analysis of object selectivity for the 1,000 ImageNet class, including the results of the latent space analysis using PCA.

In the revised manuscript, we demonstrate our new finding that selectivity to a number of objects can arise in a random network, simply due to their structural simplicity, and discuss how face-selectivity can emerge in the same way and thus may not be a special function of the brain, as previously considered. Thus, we removed all claims such as a “face is special compared to other objects,” and we decided to report objectively that selectivity occurs for various objects with a simple structure in random networks, as does selectivity for faces. We then discuss that this might be how face-selectivity arises in infant animals without visual experience and that selectivity for other objects can also arise in infant animals, though this should be studied in further animal experiments.

Revised text (Results, Line 279-303)**Emergence of selectivity to various objects in untrained deep neural networks**

Lastly, we investigated the possibility that units selective to various objects other than faces also emerge similarly in untrained neural networks. For this, we measured the responses of units in random networks to a stimulus dataset

of ImageNet containing 1,000 classes of objects (**Fig. 5a**). As a result, we found that selective units are observed in 39 classes among the 1,000 classes (**Figs. 5b, c**). From the analysis using “scrambled” and “textform” control images, we confirmed that these object-selective units are not particularly selective to local image parts but are selective to a whole object, similar with units selective to faces (**Fig. 5b**, inset). This result suggests that units selective innately to various objects such as faces can emerge in untrained neural networks.

Next, to investigate the emergent condition of units selective to each object further, we sorted those 39 classes according to the number of selective units observed and computed the PFI using the reverse correlation method (**Figs. 5c, d**). In general, we observed a tendency in which the PFIs of the “large number” group showed a relatively simple configuration of each preferred object class that was visually observable (**Fig. 5d**, top), whereas those of the “small number” classes represented a more complicated structure of the PFI (**Fig. 5d**, bottom). From this result, we hypothesized that objects with a simple configuration, such as faces, can induce stronger clustering in the latent space representation than those of other object classes and therefore may have a greater likelihood of generating units selective to them. To validate our hypothesis, we used the dimension-reduction method (PCA⁷⁷) to compare a clustered representation of each object class in terms of the raw pixel values and in the responses of Conv5. For quantification of the representational clustering of each class, we measured the Silhouette index⁷⁸ to estimate the consistency of data clustering. We found that there is a strong correlation between the Silhouette index in the Conv5 latent space and the number of selective units observed (**Figs. 5e-g**, Pearson correlation coefficient, $n_{\text{Net}} = 50$, $r = 0.58$, $p = 7.5 \times 10^{-5}$). This result demonstrates that objects with a simple profile, readily distinguishable from those of other objects statistically, lead to a strong clustering of abstracted responses in the DNN and are more likely to generate units selective to it. Thus, this implies that the observed face-selectivity may not be a special case of tuning, whereas selectivity to other visual objects can also arise in random networks simply due to the relatively simple configuration of the corresponding geometric components.

Concern 2: Definition of “unit”

There appears to be a misunderstanding around the way units are defined, given that convolutional neural networks work through weight-sharing. According to line 86 and supplemental table 1, “units” are at different spatial locations of the same filter. This sounds like a misunderstanding of the CNN architecture. Weight sharing is the fundamental feature that enables efficient training of deep architectures. The first layer, conv1, takes an image 227x277 pixels in size and with 3 color channels. The layer then contains 96 “filters”, each “filter” is 11x11x3 matrix (X, Y, and Color). However, there are more than 96 “units”, in fact there are 290,400 “units”. The explanation is that each filter gets copied to 55x55 spatial positions ($55 \times 55 \times 96 = 290400$). **Thus each “unit” is one of 3,025 copies**. Because weight sharing is active across all layers, each of the $2 \times 128 = 256$ filters is copied $13 \times 13 = 169$ times. Thus line 86 says conv5 has “43,264 ‘units’ (= $13 \times 13 \times 256$)”. To put it briefly: if a given filter represents a face feature in one spatial location, it should also represent it at every other location. Thus it is problematic to think of face-selective units beyond the channel dimension (in conv5, 256 channels). The manuscript never addresses this basic fact, while the code does not remove weight sharing and the number of weights and biases reported is consistent with a method that still has weight-sharing active. What would it mean if face-selective “filters” are not face selective in all their positions? We

can't say whether they are or are not, it is not reported. We can't say whether the analysis is just multiple repetitions of the same unit.

We regret that we did not explain this issue explicitly, particularly why we defined our "unit" in this way. Indeed, we did not misunderstand or ignore weight-sharing in convolutional neural networks. In brief, we considered each unit (of the same filter) at different spatial locations as a different one, as the selectivity of units at different locations appears to be distinct despite the fact that they share the same filter. Accordingly, we confirmed that the preferred feature images of units (of the same filter) vary across different locations. Therefore, we defined each unit at different locations as a separate unit. Details are as follows:

1. Face-selectivity appears different across spatial positions; thus, units at different locations are to be considered separately.

- The response of an identical filter appeared either face-selective or non-face-selective depending on its spatial position of the network. Thus, a "face-selective unit" is not equal to a "face-selective filter." Instead, a face-selective response is induced by a filter at a particular location. This is how we defined our unit. (See also the response to Q-(a) for more details.)
- The number of face-selective units observed for any single filter was not distinctively larger than those for the other filters. This suggests that face-selective units are not dominantly generated by a particular filter and that our estimation of the number of units is not overly high.

Notably, our definition of units may provide a better scenario by which to compare the mathematical architectures of CNN models with those of biological neural networks so as to understand similarities and differences between them: Intriguingly, the current scenario implies that weight-sharing among units at different locations may be approximated by the topographic functional maps in sensory cortices, where similar neural tunings are repeated periodically. Details are presented below.

2. Periodic tuning maps in biological brains may implement functions similar to the weight-sharing of units in CNN models.

In the visual cortex, it has been observed that neurons with a similar tuning appear periodically and are organized into topographic functional maps. For example, in the primary visual cortex (V1), neurons with similar preferred orientation and receptive fields appear periodically across the cortical surface, organizing a hexagonally periodic map (a). Importantly, the responses of these neurons vary

across cortical positions even though their receptive fields (i.e., filters) and tuning profiles are nearly identical, as they always receive distinct inputs from different locations in the visuotopic space. Thus, in CNN models, units with the same filter but at different spatial locations may be considered as a simplified model of each neuron in this type of cortical architecture.

Revised text (Results, Line 106-110)

Here, a “unit” is defined as a unit component at each position of the channel in an activation map of the network. For example, there are 43,264 “units” ($=13 \times 13 \times 256$, $N_{x\text{-position}} \times N_{y\text{-position}} \times N_{\text{channel}}$) in Conv5. We considered each unit (of the same filter) at different spatial locations as different ones, as the selectivity of units at different locations appears to be distinct despite the fact that they share the same filter (**Supplementary Figs. 2a-b**).

Revised text (Results, Line 203-207)

From the similarities between the CNN and the biological brain models, i.e., that the fundamentals of the both CNNs and sensory cortices are based on the hierarchical feedforward structure and that the process of convolution via weight sharing in CNNs can be approximated by a biological model⁷²⁻⁷⁵ of periodic functional maps with hypercolumns in the visual cortex, our result may inspire new insight into how “innate” invariance can arise in infant animals.

Recommendations:

a. Please clarify the relationship between face selectivity of the same filter at different spatial locations. Let the reader know if/why this makes sense or is surprising. Please quantify how many of the face selective units are repeats of the same filter at different spatial locations (and thus possibly over-counted).

From the additional analysis, we found that the selectivity of units at different locations is not identical despite the fact that they share the same filter. Accordingly, the preferred feature image of units varies across different locations. In addition, we observed that the number of face-selective units from any single filter was not distinctively larger than those from the others. This result suggests that face-selective units are not dominantly generated by a particular filter and that our current estimation of the number of units is not overly high. The results of our new analysis are summarized below.

1. Face-selective units, not face-selective filters — there are no filters that are face-selective in all their positions.

First, to investigate whether the observed face-selective units emerge dominantly from a particular filter, we examined the face-selective and non-face-selective responses from the entire response matrix of the Conv5 units (**a**). As a result, we found that no filters show face-selectivity in all spatial positions. In all cases

of response matrices, only a small number of face-selective units can be observed for each filter (**b**). Furthermore, we found that the positions of observed face-selective units in the response matrix varied greatly across each filter. For example, face-selective positions mostly appeared in the central position of the response matrix of filter #67, whereas they appeared in the periphery position in the response matrices of filters #99 and #247 (**b**).

2. Face-selective units are not dominantly generated by particular filters — the number of face-selective units for any single filter was not distinctively larger than those for the other filters.

To investigate whether particular filters dominantly generate face-selective units, we counted the number of face-selective units observed for each filter (**c**). As the number of face units per matrix increased, the number of observed cases quickly decreased exponentially. This distribution is not statistically distinct from the distribution obtained by shuffling the filter index for each face unit (**d**, Kolmogorov–Smirnov test, $p = 0.16$). These results confirm that face-selective units are not dominantly generated by any particular filter.

Furthermore, the results suggest that we need to consider the spatial positions of units or the structure of the corresponding filters separately in our definition of the units in the current study, as face-selectivity significantly varies according to both parameters. We added this result and show it in Supplementary Figure 2 in the revised manuscript.

Revised text (Results, Line 106-110)

Here, a “unit” is defined as a unit component at each position of the channel in an activation map of the network. For example, there are 43,264 “units” ($=13 \times 13 \times 256$, $N_{x\text{-position}} \times N_{y\text{-position}} \times N_{\text{channel}}$) in Conv5. We considered each unit (of the same filter) at different spatial locations as different ones, as the selectivity of units at different locations appears to be distinct despite the fact that they share the same filter (**Supplementary Figs. 2a-b**).

Revised text (Supplementary Fig. 2 caption)

a. Schematic of definition of unit and matrix in network. **b.** Face-selective and non-face-selective responses from the entire response matrix of Conv5 units. No filters show face-selectivity in all spatial positions. In all cases of response matrices, only a small number of face-selective units are observed for each filter. **c.** Histogram of the counted number of face-selective units observed for each filter.

b. Please break down the reporting of face selective units down by filter (AKA “channel”) and convolution group (there is grouped convolution) as well as by layer.

As suggested, we examined the statistics of the face units for each filter and convolution group. From this additional analysis, we found that the number of face-selective units increases in deeper hierarchical layers, but there is no significant difference across the convolutional group or filters within each layer. These results imply that face-selective units may not be generated by any particular filter or convolutional group.

As shown in our previous manuscript, we confirmed that the number of face-selective units increases in deeper hierarchical layers (**a**). Then, with the notion that AlexNet contains the grouped convolutional structure, we investigated whether these numbers change across the convolutional groups within Conv4 and Conv5 (**b**), in which face units were observed. As a result, we found that the number of face-selective units do not differ significantly across the convolutional groups in both layers (**c**, $n_{\text{Net}} = 100$, Conv4, Rank-sum test, n.s., $p = 0.46$, Kolmogorov–Smirnov test, n.s., $p = 0.26$; Conv5, Rank-sum test, n.s., $p = 0.54$, Kolmogorov–Smirnov test, n.s., $p = 0.56$).

Next, we investigated whether the number of observed face-selective units shows any statistical bias for particular filters within each convolutional group and layer, as we did in question (**a**). We found that the distribution

of the face-selective unit numbers is not significantly different when we shuffled the matrix index of each face-selective unit in all convolutional groups and layers (**d**, $n_{\text{Net}} = 100$, Kolmogorov–Smirnov test, Conv3, $p = 0.16$, Conv4: Convolutional group 1, $p = 0.17$, Conv4: Convolutional group 2, $p = 0.26$, Conv5: Convolutional group 1, $p = 0.16$, Conv5: Convolutional group 2, $p = 0.18$). Taken together, these results imply that face-selective units are not dominantly generated by any particular convolutional group or filter structure.

Revised text (Results, Line 106-119)

Here, a “unit” is defined as a unit component at each position of the channel in an activation map of the network. For example, there are 43,264 “units” ($=13 \times 13 \times 256$, $N_{x\text{-position}} \times N_{y\text{-position}} \times N_{\text{channel}}$) in Conv5. We considered each unit (of the same filter) at different spatial locations as different ones, as the selectivity of units at different locations appears to be distinct despite the fact that they share the same filter (**Supplementary Figs. 2a-b**). We also investigated the layer-specific emergence of face-selective units in untrained networks. We found that face-selective units are also observed in earlier layers, Conv3 to 5 but are scarcely found in Conv1 and 2 (**Fig. 1e**, Conv1: $n = 0.008 \pm 0.002\%$, Conv2: $n = 0.047 \pm 0.009\%$, Conv3: $n = 0.491 \pm 0.089\%$, Conv4: $n = 0.534 \pm 0.103\%$, Conv5: $n = 0.579 \pm 0.146\%$). We found that the number of face-selective units and the face-selectivity index of each unit increased through the layer hierarchy (**Fig. 1e-f**). Notably, the number of face-selective units did not show significant differences across the convolutional group or filters within each layer (**Supplementary Figs. 2c, d**). This suggests that face-selective units are not dominantly generated by a particular filter. The number of observed face-selective units was highest in the mid- and high-level layers, similar to observations in the ventral visual pathway of monkeys^{5,6}. These results suggest that the development of face-selectivity requires a hierarchical structure of the network along with random feedforward weights, which enables multiple linear-nonlinear computations.

Revised text (Supplementary Fig. 2 caption)

d. Schematics of the grouped convolution structure in AlexNet. **e**. The number of face-selective units in each convolutional group of Conv4 and Conv5 (1st Conv4 vs. 2nd Conv4, Rank-sum test, $n_{\text{Net}} = 100$, n.s., $p = 0.46$, Kolmogorov-Smirnov test, n.s., $p = 0.26$; 1st Conv5 vs. 2nd Conv5, Rank-sum test, $n_{\text{Net}} = 100$, n.s., $p = 0.54$, Kolmogorov-Smirnov test, n.s., $p = 0.56$).

Concern 3: Invariance

A related concern to all the issues above (selective emphasis of selectivity, PFI analyses, and weight sharing) is that of invariance. First, some forms of invariance are a basic property of CNNs, particularly motivating the development of AlexNet (see 2 3 4 5 6). Translation invariance is induced through the simple act of convolution, especially if a filter is smooth (as opposed to a grating). This is because filters have width (e.g., 11x11, 5x5, or 3x3). Not all filters produce translational invariance and the translational invariance due to filtering is only approximate. By coarse-graining (using stride >1) larger invariance width is induced. Max pooling is an exact form of translational invariance that is most impactful for subsequent layers. Translational invariance begets rotational and pose invariance when using a feature detector instead of a more basic filter (e.g. an oriented gradient filter will not become invariant to the rotation of a triangle by simply pooling translations of it, but a corner detector will.). These are the intentions behind the design of CNNs. A fact which has received much study is that CNNs do not work as intended (see 7 8 9). These invariance mechanisms work best when a filter is smooth (e.g. a spatial averaging filter). When a CNN is trained the filters are no longer anything like spatial averages. However, when it is initialized the variation within each filter is unstructured and limited. Thus, random initializations are expected to be more like averaging filters and thus CNNs are expected to be closer to the theoretical limits of invariance.

Another, more interesting form of invariance (which the manuscript addresses) is that an earlier layer can have different filters corresponding to different versions (viewpoints) of the same feature (semi-redundant features). Rather than co-opting translation invariance it is possible that the subsequent layer can combine these inputs. However, unlike pooling and convolution, there was no way to enforce this condition. It is simply hoped that CNNs will “discover” that they can do this during the training process. Unfortunately it is difficult to separate this form of invariance from translation. In a given channel, say one from conv5. It has a 3x3x256 convolution operator. It is possible that one of the 3x3 filters has all 9 values which are much larger than the other “slices”. This would be a “pass-through” filter and would implement viewpoint invariance by virtue of its spatial averaging function. These two must be separated.

Given that we expect viewpoint invariance from CNN theory, the viewpoint invariance analysis is largely over-emphasized. It is nice to see that expectations are confirmed but it is framed as an unexpected finding. It is expected for anyone with experience with the CNN architecture and its motivations. However, most of it should be removed to the supplemental. We acknowledge that many in the neuroscientific community may lack this background and would be interested in seeing it. Computer-vision specialists may also find it interesting to see the performance confirmed in this matter (especially with regard to 7). However, it is overstated in its current form and no acknowledgement of its origins in the basic CNN algorithm is made. Thus, rather than educating the neuroscientific community, it may promulgate misconceptions, and members of the computer vision community might regard the lack of context as undermining the paper’s credibility. Thus, the section “Invariant face representation of face-selective units in the untrained network” should be reduced to a brief summary to be expounded upon in the supplemental for the curious.

Commendably, one part of the viewpoint analysis stands above the rest. Above, I described how the researchers who developed CNNs hoped that they would combine semi-redundant feature detectors. This is a possibility that researchers left open to the CNN training process, but they had no way of ensuring it actually happened (as opposed to translation-based invariance, which is forced upon all CNNs by coarse-graining). The imposition of translation invariance through coarse

graining does not depend on training but combining semi-redundant filters may or may not happen. In Figure 5c, the authors report they found direct evidence that viewpoint-invariant conv5 units receive projections from viewpoint-specific conv4 units (again, using the face-trained image generator). Except they show 4-5 inputs per conv5 unit, yet a given conv5 unit receives inputs from *all* channels in the previous layer (that is part of the weight matrix, which relates all 192 channels in conv4 to each of the 256 channels in conv5). So how were the featured inputs selected for the figure? The authors say they “back-projected,” but this is undefined. Was it based on weight magnitude? What about the next five strongest weights? Notably, even a non-selective conv5 unit receives input from these view-specific channels in conv4. How is the influence of those inputs minimized?

Again, we reduce this major concern to the following action points:

a. The authors should summarize what is known and about invariance in CNNs. They should reduce figure 4 and lines 209 through 252 to a single paragraph summarizing the intrinsic invariance of convolution (weak invariance) and max-pooling (strong invariance) with a brief summary of how they confirmed this was not destroyed. It should specifically address how the lack of spatial structure in your filters relates to the kinds of spatial structure in a trained AlexNet (one might consult OpenAI Microscope 10 for comparison) and how types of spatial structure in filters can enhance or detract from invariance Details can be moved to the supplementary.

We appreciate your detailed comments and suggestions on this issue, considering both views from neuroscience and computer science. In particular, bearing in mind potential readers of the paper from the computer vision field, our description of the invariance in the previous results may appear to be overemphasized. Thus, as suggested, we summarized these results regarding the invariance of units in the revised manuscript, focusing on the two main issues below:

- Some forms of “intrinsic” invariance are a basic property of CNNs (LeCun, 2012; LeCun et al., 2004), and we confirmed that an important example, i.e., view-point invariance, is observed in our model condition with the random initialization of the weights.
- Our additional analysis suggests that the observed invariance can arise from the hierarchical structure with convolutional filtering without the contribution of any particular features in the spatial structure of the filters.

1. Viewpoint invariance is confirmed in our model CNN with randomly initialized weights

As the reviewer suggested, initially we briefly summarized previous research pertaining to the “intrinsic” characteristics of invariance in CNNs in that the CNN architectures were designed to implement invariant cognition (Fukushima and Miyake, 1982; LeCun, 2012; LeCun and others, 1989; Zeiler and Fergus, 2014). Specifically, we described three key components — the convolutional layer, the pooling layer, and the hierarchical structure — that play crucial roles to induce different types of invariance over a wide range of image transformations. We also clarified that a number of previous studies suggested that various types of invariance (e.g., translation, scaling, rotation) can be implemented in a CNN (Chidester et al., 2018; Kavukcuoglu et al., 2010; LeCun, 2012; LeCun et al., 2004; Srivastava and Grill-Spector, 2018).

On the other hand, we regret that our expected scenario was explained insufficiently in the previous manuscript — In fact, the reviewer’s suggestions are not different from what we originally intended in our analyses. Thus, we agree that an appropriate interpretation of our previous results would be that (1) some types of invariance are expected in a CNN from its intrinsic design factors, such as its hierarchical and convolutional structures, and (2) we confirmed that a certain type of invariance, in this case viewpoint invariance, is observed in CNNs with randomized weights. Notably, it is important that the current result suggest a possible scenario in which to understand how viewpoint-invariant selectivity can arise in infant animals. From the similarities between the CNN and animal models that (1) the fundamentals of the both CNNs and sensory cortices are based on the hierarchical feedforward structure and that (2) the process of convolution via weight sharing in CNNs can be approximated by a biological model of periodic functional maps with hypercolumns in the visual cortex, we suggest this idea in the revised text, as we described in Concern 2 earlier. All things considered together, our results may inspire new insight related to how “innate” invariance can arise in infant animals.

2. Invariance from hierarchical structures without structured spatial filters

As the reviewer suggested, the structure of spatial filters could affect certain types of invariance, such as translation. Considering that trained networks have smooth filters, whereas filters in the random network are unstructured, it is important to compare the invariance generated by randomly structured filters with that from smooth filters. To do this, we conducted an analysis using a new control in which the phase of trained filters is randomized (randomized structure) while the corresponding spatial frequency (smoothness) is maintained (a).

As a result, we found that the invariance with the control (**b**) was only slightly higher than that from the unstructured random filters, without a noticeable difference (**c-e**). To estimate the degree of invariance quantitatively, we measured the effective range of invariance within which responses to face images were significantly higher than those to non-face images. We found that the effective ranges in the two cases were not significantly different (**c**, $n_{\text{Net}} = 100$, Rank-sum test, n.s., $p = 0.13$). Moreover, the effective ranges of the invariant responses in the two cases were significantly smaller than that of the pre-trained network (**e**, $n_{\text{Pretrained}} = 1$, $n_{\text{Phase}} = 100$, Signed-rank test, * $p = 4.04 \times 10^{-19}$). These results imply that the smoothness of the filter may not be crucially related to the emergence of the observed invariance, which originates from the hierarchical structure of the network. In the revised manuscript, we summarized our results with these discussions.

Revised text (Results, Line 185-207)

Next, we hypothesized that the observed face-selective units may encode invariant representations of the prototype face images (**Fig. 2**), as some types of “intrinsic” invariance are a basic property of CNNs. A number of previous studies suggested that various types of invariance (e.g., translation, scaling, rotation) over a wide range of image transformations can be implemented in a CNN^{65–69}, mostly due to three key components — the convolutional layer, the pooling layer, and the hierarchical structure — in CNN models. Thus, to investigate whether the observed face-selective units show invariant representations of face images regardless of the corresponding image condition, we measured the responses of face-selective units to face and non-face object images with various positions, sizes, and rotation angles. First, we observed that single face units show constant face tuning under a fairly wide range of size/position/rotation variations and that range was comparable with those of face-selective neurons in IT⁷⁰ (**Supplementary Figs. 5, 6**). Notably, we found that our model units also show the “inversion effect” observed in monkeys^{5,27,71}. We found that the responses of face-selective units to inverted face images are significantly lower than those to upright faces (**Supplementary Fig. 7**).

Similarly, we found viewpoint-invariant face-selective units (**Supplementary Figs. 8, 9**) and mirror-symmetric viewpoint-specific units (**Supplementary Fig. 10**) in a random network. Interestingly, the number of viewpoint-invariant face-selective units increased along the network hierarchy, similar to previous observations in the brain⁷. These results show that the observed invariances can arise from the hierarchical structure with convolutional filtering without the contribution of structured spatial filters. Notably, the current result also suggests a possible scenario through which to understand how viewpoint invariant selectivity can arise in infant animals. From the similarities between the CNN and the biological brain models, i.e., that the fundamentals of the both CNNs and sensory cortices are based on the hierarchical feedforward structure and that the process of convolution via weight sharing in CNNs can be approximated by a biological model^{72–75} of periodic functional maps with hypercolumns in the visual cortex, our result may inspire new insight into how “innate” invariance can arise in infant animals.

b. Lines 254 through 268 should be re-written to include a concise description of why the CNN architecture is likely to produce the scenario described on lines 267 and 268, and why trained and untrained networks are different.

We agree that the results of viewpoint invariance were overemphasized and that a discussion of the intrinsic invariance of CNN models is missing in the previous manuscript. In the revised text, we made a concise summary of this section, including the analysis of invariance discussed above. In this paragraph, we stated that invariant representations, in this case viewpoint invariance, are predicted in our model, as some types of “intrinsic” invariance are a basic property of CNNs. We described how we arrive at the hypothesis that the observed face-selective units may encode invariant representations of the prototype face images, with the introduction of previous studies finding that invariances are an intrinsic property of CNNs. We then explained that the current result shows that the observed invariances can arise from the hierarchical structure without the contribution of structured spatial filters (For details, see our response to Concern 3 Question a).

In addition to the revision of the text, we conducted an additional analysis to explain why our model predicts that viewpoint invariance can arise from untrained random networks, and we also examined how the types of invariance in trained and untrained networks differ from one another. As a result, we found that viewpoint invariance in untrained networks can simply originate from the random wiring of viewpoint-specific units in previous layers and that the actual number of invariant units observed is nearly identical to the prediction by our random wiring model. We also found that the number of viewpoint-invariant units is greater in pre-trained networks than in untrained networks, implying that the initial, primitive viewpoint invariance may be strengthened and refined by visual training.

In detail, we hypothesized that viewpoint-invariant units can arise solely from random wiring (i.e., randomized convolutional weights) of viewpoint-specific units in untrained networks. To validate this idea, we investigated the chance that viewpoint-invariant units in Conv5 arise from the random connections of viewpoint-specific units in Conv4. The detailed process of this analysis is presented below.

1. The numbers of face units and non-face units are counted in Conv4, and the numbers of viewpoint-specific units were counted separately.
2. To model the random wiring from Conv4 to Conv5, random weight values ($n=1,078$; matching the size of the convolutional filter) were sampled and applied between randomly sampled Conv4 units and a model unit in Conv5. This process was repeated to generate $N=43,264$ model neurons in Conv5.
3. The connectivity from the units in Conv4 to each model unit in Conv5 was examined. Conv5 units that receive stronger connections from face units in Conv4 compared to those from non-face units (**c**, Rank-sum test, $p < 0.01$) were labeled as face-selective units in Conv5 (**a-d**). The number of these face-selective units was counted in Conv5.
4. Units in Conv5 that receive projections from viewpoint-specific units in Conv4 with uniformly distributed weights (**f**, ANOVA with a Bonferroni post-hoc test, $p > 0.05$) were labeled as viewpoint-invariant (**e-h**).
5. These processes were repeated 20 times in each of 100 randomly initialized networks.

As a result, we found that the number of emerging face-selective units by random wiring is consistent with that observed in untrained networks (**d**, right, Rank-sum test, n.s., $p = 0.36$). In addition, we observed that the number of emerging viewpoint-invariant units in the model also matches that observed in untrained networks (**h**, right, Rank-sum test, n.s., $p = 0.36$). These results suggest that viewpoint invariance in untrained networks can simply originate from the random wiring of viewpoint-specific units in previous layers.

We also found that the number of viewpoint-invariant units is greater in pre-trained networks than in

untrained networks (**h**, Signed-rank test, $p = 1.98 \times 10^{-18}$), implying that the initial, primitive viewpoint invariance may be strengthened and refined by visual training. We added these results in the revised manuscript.

Revised text (Supplementary Fig. 9 caption)

a. Face-selective units observed in Conv4 and Conv5 of untrained networks. **b.** Face-selective units in Conv5 of untrained networks are observed to receive strong projections from face-selective units in Conv4. **c.** A random

wiring model in which face-selective units in Conv5 emerge from the random wiring of face-selective units in Conv4: The numbers of face units and non-face units are counted in Conv4, and the numbers of viewpoint-specific units were counted separately. To model the random wiring from Conv4 to Conv5, random weight values ($n=1,078$; matching the size of the convolutional filter) were sampled and applied between randomly sampled Conv4 units and a model unit in Conv5. This process was repeated to generate $N=43,264$ model neurons in Conv5. **d.** The connectivity from units in Conv4 to each model unit in Conv5 was examined. Conv5 units that receive stronger connections from face units in Conv4 compared to those from non-face units (Rank-sum test, $p < 0.01$) were labeled as face-selective units in Conv5 (left). The number of these face-selective units was counted in Conv5. The number of emerging face-selective units by random wiring is consistent with that observed in untrained networks (right, $n_{Net} = 100$, Rank-sum test, n.s., $p = 0.36$). **e.** Viewpoint-specific and viewpoint-invariant face units in Conv4 and Conv5 of untrained networks. **f.** Units in Conv5 that receive projections from viewpoint-specific units in Conv4 with uniformly distributed weights (ANOVA with Bonferroni post-hoc test, $p > 0.05$) were labeled as viewpoint-invariant in the model. **g.** The random wiring model predicts that viewpoint-invariant units in Conv5 can originate from the random projections from viewpoint-specific units in Conv4. **h.** The number of emerging viewpoint-invariant units in the model matches that observed in untrained networks (right, Rank-sum test, n.s., $p = 0.36$). Notably, the number of viewpoint-invariant units is greater in pre-trained networks than in untrained networks (Signed-rank test, $p = 1.98 \times 10^{-18}$).

c. The authors should elaborate on how they chose filters for display on their figures and how they performed “back-projection.” They should make sure to explain how they know that these specific inputs matter more than the others. This should include the context of the Gaussian distribution from which weights are pulled, and the context of weaker inputs, which may all become active with the same image and thus matter more than the few strong connections.

We apologize that detailed information with regard to these issues is missing in the previous manuscript, and we thoroughly revised the analysis. We found that the previous analysis using the selected “source - target” back-projection relationship was not the best choice because it ignores the contribution from many other unit projections (**a**, top) and also raises issues regarding the choice of analysis conditions, as the reviewer mentioned. In the revised analysis, we examined the entire weight distribution to each target unit (**a**, bottom) such that a complete weight profile of each unit could be investigated. From this analysis, we confirmed that our conclusion is consistent with that in the previous analysis (**b-d**): We observed that face-selective units in Conv5 receive dominant inputs from Conv4 face-selective units. Specifically, viewpoint-specific units in Conv5 are connected to viewpoint-specific units in Conv4 with the same preferred viewpoint angles (**b** and **c**). On the other hand, viewpoint-invariant units in Conv5 are uniformly connected to a group of viewpoint-specific units with different preferred viewpoint angles (**d**).

In detail, we initially examined viewpoint-specific units in Conv5, finding that they are strongly connected to viewpoint-specific units of the same preferred viewpoint angle in Conv4. First, we found that viewpoint-specific units in Conv5 receive stronger projections from face units than from other non-face units in Conv4 (**b**, center, $n_{Net} = 100$, Rank-sum test, $**p = 2.55 \times 10^{-24}$, Signed-rank test, $*p = 2.55 \times 10^{-24}$). In addition, these viewpoint-specific units in Conv5 receive dominant projections from viewpoint-specific units with the same preferred viewpoint in

Conv4 (**b**, right; Rank-sum test, $**p < 4.14 \times 10^{-3}$). Notably, the average weight value of the projections between the viewpoint-specific units in Conv4 and Conv5 is higher than that of all projections in the network (**b**, right; Signed-rank test, $*p < 7.82 \times 10^{-3}$). We confirmed that this relationship was observed in all viewpoint-specific units in Conv5 (**c**).

Next, we examined viewpoint-invariant units in Conv5, finding that the viewpoint-invariant units in Conv5 receive uniform projections from viewpoint-specific units of various viewpoint angles in Conv4. We found that the average weight value of the projections between viewpoint-specific units in Conv4 and viewpoint-invariant units in Conv5 is higher than that of all projections in the network (**d**, center; Signed-rank test, $*p < 7.82 \times 10^{-3}$) and that the weight values of projections from Conv4 units with a different preferred viewpoint relative to that of a viewpoint-invariant Conv5 unit were not significantly different from each other (**d**, right; one-way ANOVA with a Bonferroni post hoc test, n.s., $p = 0.33$). Overall, these results suggest that the viewpoint-specific and viewpoint-invariant face tunings may originate from the connectivity bias in the random wirings of bottom-up projections and that statistical variance in the random network is sufficient to generate such connectivity bias, as stated in the response to question (b) above.

Revised text (Supplementary Fig. 8 caption)

c. To validate this scenario, we backtracked projections of the units from the source layer (Conv4) to the target layer (Conv5) and examined the weights of connected viewpoint-specific units. First, we confirmed that viewpoint-specific ($n = 136 \pm 30$) and viewpoint-invariant face-selective units ($n = 163 \pm 35$) exist in Conv4, as well as in Conv5 (viewpoint-specific, $n = 96 \pm 27$, viewpoint-invariant, $n = 127 \pm 36$). We found that the viewpoint-specific units in

Conv5 receive inputs from Conv4 units strongly biased to a particular viewpoint angle (right, Rank-sum test, $**p < 4.2 \times 10^{-3}$). The panel presents the weight values of the projections from Conv4 units to a viewpoint-specific unit in the Conv5 target layer. **d.** However, the viewpoint-invariant units in Conv5 receive input from Conv4 units with a fairly homogeneous distribution of viewpoint angles (right, one-way ANOVA with a Bonferroni post hoc test, n.s., $p = 0.33$). The panel presents the weight values of the projections from Conv4 units to a viewpoint-invariant unit in Conv5. The PFI of viewpoint-specific units in Conv4 connected to viewpoint-invariant units in Conv5 (as shown (b)) also show whole face-configuration at each viewpoint angle. These results suggest that the viewpoint-invariant units in Conv5 as well as viewpoint-specific units connected to them are selective to whole faces rather than face parts.

d. The authors should characterize the contributions of weights within the same kernel to weights in different kernels and thus show whether their data evinces convolution within one channel or combinations of semi-redundant channels.

First, for a discussion of this issue, please note **a** that we defined “units” at different spatial locations of the same filter — i.e., units at different locations are to be considered separately, as selectivity appears to be different across spatial positions. We explained this issue in detail in our response to your Concern 2.

From our additional analysis of the connection weights of each unit as described below, we found that the contribution of the weights is not noticeably biased by one channel or by any particular filter. We found that invariant units do not dominantly originate from any single channel or filter. Instead, we confirmed that they emerge from the balanced projections of viewpoint-specific units in the previous layer.

Thus, our result suggests that we must consider viewpoint-invariant units as stemming from combinations of units (not channels) that are selective for each viewpoint angle.

To investigate the contributions of weights within the same kernel to weights in different kernels, we estimated the weight values of the connections to viewpoint-invariant units (a). We labeled these weight values according to the output channel index (a, j^{th} channel in Conv5). Then, we labeled these values again by the slice index corresponding to the input channel index (a, i^{th} channel in Conv4). As a result, we obtained

a two-dimensional matrix of weights in which the average weight value of each unit is represented. Then, we found that weights in the matrix are not biased across either input or output channel indices (b, two-way ANOVA with Bonferroni post-hoc test, $n_{\text{weight}} = 331,776$, Conv4, n.s., $p = 0.14$, Conv5, n.s., $p = 0.16$). This result suggests that the viewpoint-invariant unit is not generated in a single channel or by any particular filter.

Next, considering our previous observations that face-selectivity varies across units in the same filter and face-selective units are not dominantly observed in some particular filter (as we stated in our response to Concern 2), we examined the weight distribution of Conv4 units connected to a viewpoint-invariant unit and to all other units in Conv5. As a result, we initially found that weights from Conv4 face units to Conv5 face units show significantly higher values than the average weight value of the network (c, center; $n_{\text{Net}} = 100$, Signed-rank test, $*p = 1.98 \times 10^{-18}$), whereas the connections from non-face units do not stray from the average weight values (c, center; Signed-rank test, n.s., $p = 1$). Second, we confirmed that the connection weights from viewpoint-specific units in Conv4 to a viewpoint-invariant unit in Conv5 are well balanced relative to each other (c, right; one-way ANOVA with a Bonferroni post hoc test, $p = 0.33$), as we reported in the previous analysis. These results suggest that viewpoint-invariant units stem from combinations of units (not channels) that are selective for each viewpoint angle.

e. Lines 309 through 321 are confusing. The images are padded with fixed integers prior to convolution. This means that variability among pixels along the periphery is less able to contribute to detections (i.e., they are washed out). This gets exacerbated with each subsequent convolution and explains why CNNs show a centrality preference — this is not related to face patches. One could challenge this objection if one of the 39 highly selected for categories showed an edge preference. However, given the convolutional nature of CNNs selectivity in convolutional layers is always expected to show some kind of spatial regularity. Thus, CNNs may not be useful for discussing the spontaneous emergence of patches with respect to the visual field. These lines should probably be removed.

We appreciate this helpful comment. We found that our previous results that face-selective units are more biased to the center of the visual field were not appropriate to support the development of a “face-patch.” In the revised manuscript, we removed this description and mentioned the limitation of explaining the development of the “face-patch” using CNN models.

f. Line 285 –286 “We found that the level of invariance increased...” Again, we know a-priori that invariance will increase through hierarchy due to repeated convolution operators. It is supposed to do that because engineers were inspired by the brain to make CNNs that way.

Again, we agree that the results of invariance were overemphasized here, as described in our responses to the previous questions above. In the revised manuscript, we clarified that our hypothesis in which face-selective units may encode invariant representations is based on the notion that various types of invariance can be expected in CNNs. Accordingly, we introduced previous studies focusing on the invariance implemented by key design factors such as convolution, pooling, and the hierarchical structure in CNN models.

Revised text (Results, Line 185-207)

Next, we hypothesized that the observed face-selective units may encode invariant representations of the prototype face images (**Fig. 2**), as some types of “intrinsic” invariance are a basic property of CNNs. A number of previous studies suggested that various types of invariance (e.g., translation, scaling, rotation) over a wide range of image transformations can be implemented in a CNN^{65–69}, mostly due to three key components — the convolutional layer, the pooling layer, and the hierarchical structure — in CNN models. Thus, to investigate whether the observed face-selective units show invariant representations of face images regardless of the corresponding image condition, we measured the responses of face-selective units to face and non-face object images with various positions, sizes, and rotation angles. First, we observed that single face units show constant face tuning under a fairly wide range of size/position/rotation variations and that range was comparable with those of face-selective neurons in IT⁷⁰ (**Supplementary Figs. 5, 6**). Notably, we found that our model units also show the “inversion effect” observed in monkeys^{5,27,71}. We found that the responses of face-selective units to inverted face images are significantly lower than those to upright faces (**Supplementary Fig. 7**).

Similarly, we found viewpoint-invariant face-selective units (**Supplementary Figs. 8, 9**) and mirror-symmetric viewpoint-specific units (**Supplementary Fig. 10**) in a random network. Interestingly, the number of viewpoint-invariant face-selective units increased along the network hierarchy, similar to previous observations in the brain⁷. These results show that the observed invariances can arise from the hierarchical structure with convolutional filtering without the contribution of structured spatial filters. Notably, the current result also suggests a possible scenario through which to understand how viewpoint invariant selectivity can arise in infant animals. From the similarities between the CNN and the biological brain models, i.e., that the fundamentals of the both CNNs and sensory cortices are based on the hierarchical feedforward structure and that the process of convolution via weight sharing in CNNs can be approximated by a biological model^{72–75} of periodic functional maps with hypercolumns in the visual cortex, our result may inspire new insight into how “innate” invariance can arise in infant animals.

Concern 4: Face selectivity across layers

It is reported that conv2 contains around 60-70% as many face-selective units as deeper layers. The receptive fields of conv2 are about 51 x 51 pixels. All the shown stimuli contain faces that appear to occupy > 65 % of the 227 x 227-pixel image. How can a such a small RF be selective for a face as well as deeper layers given the same RF? Would it give rise to a different face configuration per PFI, or to a face part? These early-layer responses cannot be for whole faces, only for face fragments – which again, the authors actively disregard as an explanation for the face selectivity. Please clarify the PFIs attained by layer, as well as the activations attained during the PFI generation process.

a. Explain whether units (especially in early layers) require whole faces or just face parts. If they require whole face explain how that is possible.

We deeply appreciate this timely comment, which enabled us to investigate face-selectivity for both whole faces and face fragments. In brief, we found that the observed units in Conv2 are selective merely to local face fragments such as eyes or foreheads. To distinguish this type of face-selectivity for local features from that for whole faces, we performed additional tests using scrambled face stimulus images, in which selectivity for local face features is separately observable from that for whole faces. As a result, we found that previously observed units in Conv2 are not selective to whole faces but are to local face fragments and thus can be filtered out by our new test. Details of our revised analysis are presented below.

To investigate whether the previously observed face-selective units in Conv2 are selective to whole faces or to face fragments, first we estimated the receptive fields (RF) of the face-selective units in each layer, after which we generated preferred feature images (PFIs) using the GAN trained to ImageNet data and cropped these images according to the size of the corresponding RF area of the target unit (**a-b**). We found that the RFs of face-selective units in Conv2 can contain only local components of face images, such as an eye or a forehead, whereas those of units in Conv3 to 5 contain a whole, global face profile (**b**). Furthermore, we found that the face-selectivity index values (FSI) of the PFI of the face-selective units in Conv2 were significantly smaller than those of the face units in Conv3 to 5 (**c**, Rank-sum test, $n_{\text{Net}} = 100$, $*p < 6.01 \times 10^{-9}$). As expected, the FSI of the face-selective units in Conv3 to 5 show no significant differences (**c**, one-way ANOVA with a Bonferroni post hoc test, $n_{\text{Net}} = 100$ n.s., $p = 0.12$). These results imply that the observed face-selectivity in Conv2 is not for whole faces and thus need to be

distinguished from that for whole faces in Conv3 to 5.

Next, to filter out the units selective to local face components only, we performed a revised test using scrambled face images. In fact, we found that our previous analysis for selecting face-selective units using scrambled faces was performed with inappropriate parameters — it could not successfully distinguish units selective to whole faces and those selective only to face parts, as the patch size for scrambling was too small to contain face compartments such as the eyes or mouth. As a result, units selective to face parts were grouped together with units selective to whole faces (**d-f**). In the revised test, (i) we increased the patch size for scrambling, and (ii) randomized the positions of only 80% of the patches so that 20% of the patches would be in the original position (**g**) to increase the chance of unit activation selective to face parts at a particular position.

As a result, we found that the number of face-selective units in Conv2 is negligibly small, indicating that units selective to whole faces are not observed in Conv2 (**h**). In the revised manuscript, we present our revised analysis results with a description of the revision method.

Revised text (Results, Line 104-119)

Surprisingly, we observed a group of “face-selective” units ($n = 250 \pm 63$ in 100 random networks, mean \pm s.d.) that show significantly higher responses to face images than to non-face images emerges in the untrained networks (**Fig. 1d**). Here, a “unit” is defined as a unit component at each position of the channel in an activation map of the network. For example, there are 43,264 “units” ($=13 \times 13 \times 256$, $N_{x\text{-position}} \times N_{y\text{-position}} \times N_{\text{channel}}$) in Conv5. We considered each unit (of the same filter) at different spatial locations as different ones, as the selectivity of units at

different locations appears to be distinct despite the fact that they share the same filter (**Supplementary Figs. 2a-b**). We also investigated the layer-specific emergence of face-selective units in untrained networks. We found that face-selective units are also observed in earlier layers, Conv3 to 5 but are scarcely found in Conv1 and 2 (**Fig. 1e**, Conv1: $n = 0.008 \pm 0.002\%$, Conv2: $n = 0.047 \pm 0.009\%$, Conv3: $n = 0.491 \pm 0.089\%$, Conv4: $n = 0.534 \pm 0.103\%$, Conv5: $n = 0.579 \pm 0.146\%$). We found that the number of face-selective units and the face-selectivity index of each unit increased through the layer hierarchy (**Fig. 1e-f**). Notably, the number of face-selective units did not show significant differences across the convolutional group or filters within each layer (**Supplementary Figs. 2c, d**). This suggests that face-selective units are not dominantly generated by a particular filter. The number of observed face-selective units was highest in the mid- and high-level layers, similar to observations in the ventral visual pathway of monkeys^{5,6}. These results suggest that the development of face-selectivity requires a hierarchical structure of the network along with random feedforward weights, which enables multiple linear-nonlinear computations.

b. Clarify how viewpoint invariant units (agglomerated via “back-projection”) relate to face parts vs whole faces.

We found that both viewpoint-invariant and viewpoint-specific units are selective to whole face profiles. To investigate this issue, we estimated the PFIs of viewpoint-invariant units in Conv5 and those of viewpoint-specific units in Conv4. First, we found that the PFIs of the viewpoint-invariant units show the face configuration of the whole face containing both the hair and face contour. Similarly, the PFIs of the viewpoint-specific units in Conv4 connected to the viewpoint-invariant units in Conv5 (as shown in Concern3-b) also show the whole face configuration at each viewpoint angle. These results suggest that the viewpoint-invariant units in Conv5 as well as the viewpoint-specific units connected to them are selective to whole faces rather than face parts.

Revised text (Supplementary Fig. 8 caption)

b. The average PFI of these viewpoint-specific units selective for each viewpoint reveals the representative feature of a face rotated at a specific angle: an asymmetric hairline with only one visible eye for 45-degree specific face units and a symmetric hairline with clearly shown eyes and nose for center-specific face units. The panel shows the PFIs of viewpoint-specific and viewpoint-invariant units in Conv5 in the untrained networks. Based on this result, we hypothesized that viewpoint-invariant units may arise from the projection of multiple viewpoint-specific units preferring various viewpoints in the previous layer. **c.** To validate this scenario, we backtracked projections of the units from the source layer (Conv4) to the target layer (Conv5) and examined the weights of connected viewpoint-

specific units. First, we confirmed that viewpoint-specific ($n = 136 \pm 30$) and viewpoint-invariant face-selective units ($n = 163 \pm 35$) exist in Conv4, as well as in Conv5 (viewpoint-specific, $n = 96 \pm 27$, viewpoint-invariant, $n = 127 \pm 36$). We found that the viewpoint-specific units in Conv5 receive inputs from Conv4 units strongly biased to a particular viewpoint angle (right, Rank-sum test, $**p < 4.2 \times 10^{-3}$). The panel presents the weight values of the projections from Conv4 units to a viewpoint-specific unit in the Conv5 target layer. **d.** However, the viewpoint-invariant units in Conv5 receive input from Conv4 units with a fairly homogeneous distribution of viewpoint angles (right, one-way ANOVA with a Bonferroni post hoc test, n.s., $p = 0.33$). The panel presents the weight values of the projections from Conv4 units to a viewpoint-invariant unit in Conv5. The PFI of viewpoint-specific units in Conv4 connected to viewpoint-invariant units in Conv5 (as shown (b)) also show whole face-configuration at each viewpoint angle. These results suggest that the viewpoint-invariant units in Conv5 as well as viewpoint-specific units connected to them are selective to whole faces rather than face parts.

Concern 5: Statistical tests and reporting

There are no reports of adjustments for multiple comparisons. When looking for selectivity among a small set of classes Bonferroni correction is one option. When looking for single units with a high FSI among hundreds of thousands of units, another method such as step-down minP is appropriate. Correction is necessary because of the lottery-ticket hypothesis the idea is that there is some chance in your extended joint-distribution of gaussians to get a filter with the properties needed to detect a series of blobs in a roughly face-like spatial arrangement. Since all weights come from the same distribution these serendipitous filters are still in the tail of the distribution, not true outliers, thus a properly executed significance test may not

give them very low p-values. Despite their possibly high p-values they are still interesting and consistent with the lottery ticket hypothesis. So, one might seek another criterion for deciding which units are selective or not (such as the receiver-operator characteristic).

Another issue with statistical reporting is that the Mann-Whitney U test seems to be misapplied. Lines 212 and 213 show a Mann-Whitney U test that is used to show “constant face tuning”. This is confusing for two reasons: 1) it is not said what has gone into the analysis, this test detects differences between two populations according to a single metric. These populations are not named, the metric is not named. Because the name Mann-Whitney U is often conflated with the Wilcoxon signed-ranked test we do not know if they are matched (signed-rank) or un-matched (rank-sum). 2) The Mann-Whitney U test cannot detect similarity. It is inappropriate to say $p > 0.05$ therefore these two populations are the same. All populations are different, one must state the resolution at which their differences matter.

Other miscellaneous statistical reporting problems: What is *P? We found no explanation in the main text. There are no effect sizes. There are many to choose from but the one which is most interpretable and consistent across datasets is the simple difference formula 11, another is the rank-biserial correlation. We also recommend using the term “signed-rank” or “rank-sum” rather than “Mann-Whitney U”. Signed-rank and rank-sum are clear terms that describe how the test is conducted in their own names and make it obvious to the reader whether the test is matched or unmatched.

The use of the Kolmogorov-Smirnov test needs to be justified for the face detection task. This test is only designed to tell if two data sets are pulled from the same distribution. It cannot detect if a metric is generally higher for one group than the other (except in cases where the rank-sum test is also diagnostic).

Actionable points:

a. Please include multiple comparison adjustments or justify for why they are not needed.

We sincerely appreciate your timely comments on this critical issue, which was not given enough attention previously. We fully agree with that multiple comparison adjustments are needed. As suggested, in the revised manuscript, we performed multiple comparison adjustments (e.g., the Bonferroni post hoc test) for every test. We describe these in the text.

Revised text (Supplementary Fig. 8 caption)

a. To investigate whether these viewpoint-invariant characteristics can also be observed in an untrained network,

we measured the responses of Conv5 in an untrained AlexNet while face images at different viewpoints were provided to the network. We found that there is a viewpoint-invariant population whose responses are not significantly different across all viewpoint classes ($p > 0.05$, one-way ANOVA with a Bonferroni post hoc test) and a viewpoint-specific population whose responses are significantly different ($p < 0.05$, one-way ANOVA with a Bonferroni post hoc test), as observed in the monkey IT². The panel shows face images with various viewpoints (Left), viewpoint-invariant and the viewpoint-specific responses of face-selective units (Right).

b. Do not use the rank-sum or signed-rank test alone for detecting whether two groups are not distinct from each other. You should use follow up tests (such as KS), a higher P-value threshold (e.g., $P > 0.1$ is more believable) and state the resolution of the test (the statistical power given the population size, or estimate).

Thank you for this helpful comment. In the revised manuscript, we additionally conducted a Kolmogorov–Smirnov test with a higher P-value threshold (n.s., $p > 0.1$), as needed, when comparing two groups in terms of whether they are distinct from each other. In addition, we included details of the statistical power for every statistical test in the revised text.

Revised text (Results, Lines 220-225)

We confirmed that the SVM trained with multiple face-selective units shows noticeably better performance than that trained with the same number of non-selective units as the number of units used in each condition was varied from $n = 1$ to 465 (total number of face units in untrained networks) (Fig. 3c, Rank-sum test, Face vs. Non-selective units, $n_{\text{trial}} = 100$, $*p < 1.4 \times 10^{-33}$). We also found that the SVM using face units ($n = 465$) nearly matches the performance of the SVM using all units in the final layer ($n = 43,264$) (Fig. 3d, Rank-sum test, Face vs. All units, $n_{\text{trial}} = 100$, n.s., $p = 0.19$; Kolmogorov-Smirnov test, n.s., $p = 0.19$).

c. Justify the use of the KS test in the face-detection task. (methods or main text)

After careful consideration with your helpful comment, we found that the Kolmogorov–Smirnov (KS) test may be inappropriate for the face-detection task, as it is used for continuous probability distributions, which is not the case for face detection here. Instead, in the revised manuscript, we use the rank-sum test to investigate the difference in each data point in the face-detecting performance.

d. Report effect sizes alongside all P-values

In the revised manuscript, we added Supplemental Table 3 to include the effect size alongside all P-values for each statistical test: the rank-biserial correlation value (Cureton, 1956) for the rank-sum test, Cohen's f^2 (Cohen, 1988) for the ANOVA test, and the dissimilarity value (Vermeesch, 2013) for the Kolmogorov–Smirnov test.

e. State exactly what groups and what metrics are used for each reporting of P-values. (supplemental is OK so long as you give some explanation in methods or main text)

We appreciate your careful comment on this missing information. In our new Supplemental Table 3, precise information about the tested group, sample size, power, and effect size of each statistical test is summarized.

Revised text (Supplementary Table 3)

Panel #	Group (n)		Test	p value	Effect size	Power
Figure 1						
f	FSI of face units (Conv2, n = 94)	FSI of shuffled response (Conv2, n = 94)	One-sided Rank-sum test	1	-0.5819	0.0204
	FSI of face units (Conv3, n = 365)	FSI of shuffled response (Conv3, n = 365)	One-sided Rank-sum test	3.9100×10^{-10}	0.2889	0.9998
	FSI of face units (Conv4, n = 444)	FSI of shuffled response (Conv4, n = 444)	One-sided Rank-sum test	2.0743×10^{-18}	0.3470	0.8010
	FSI of face units (Conv5, n = 465)	FSI of shuffled response (Conv5, n = 465)	One-sided Rank-sum test	1.4890×10^{-25}	0.5085	0.9999
	FSI of face units (Conv5, n = 465)	FSI of face neuron (Tsao 2010, n = 158)	Two-sided Rank-sum test	0.0769	0.0925	0.0504
g	Response to face (n = 200)	Response to scrambled face (n = 200)	One-sided Rank-sum test	1.7082×10^{-52}	0.7686	0.9999
	Response to face (n = 200)	Response to texform face (n = 100)	One-sided Rank-sum test	4.1201×10^{-30}	0.6559	0.9999
	Response to non-face (n = 200)	Response to scrambled face (n = 100)	One-sided Rank-sum test	1	-0.2424	1.1118×10^{-19}
	Response to non-face (n = 200)	Response to texform face (n = 100)	One-sided Rank-sum test	0.9402	-0.0900	3.4802×10^{-7}

▪
▪
▪

Concern 6: Missing important related works

The manuscript is missing an obvious relation to neural encoding paradigms. We regret not mentioning this in our first review and commend the other reviewers for the astute recommendation to invoke the lottery ticket hypothesis as a significant and relevant body of work. However, reservoir computing 12 is more closely related to theories of brain encoding (here is a starting point for your research 13 14 15 16). This is an area of research called “reservoir computing”. Two theoretical models of neural

computation are “echo state networks” and “liquid state machines”. These approaches seek to explain how to quickly train a large, complicated network while resolve some of the ways that populations of spiking neurons behave inconsistently with older concepts of neural code. Arguably, this paper accidentally implements and tests a simple version of a reservoir computer.

a. Accurately place the work in the context of reservoir computing, one or two sentences in the discussion alongside “lottery ticket hypothesis” is fine.

We appreciate this helpful comment. As suggested, we introduced additional works regarding the context of reservoir computing in the revised text. Accordingly, we provide a more in-depth discussion of the similarities and differences between the reservoir computer model and our current result, which may provide insight leading to a deeper understanding of our results by readers.

It is true that the current study shares a core idea with the reservoir computing model in that the circuits required for higher order cognitive functions, such as image classification, can already exist in random neural networks — the theory of “reservoir computing” suggests that a high dimensional space generated by a random “subnetwork” can perform various tasks without learning such that higher order cognitive functions can be achieved only by training the read-out network, as suggested by the lottery ticket hypothesis (Bellec et al., 2020; Verstraeten et al., 2007). However, it must be also noted that our results demonstrate that the functional tuning of “single units” (comparable with neuronal tuning in biological brains) can arise in random networks, even without any further training of the read-out process. This is clearly distinguished from the main idea of the reservoir computing model. We added further discussions of this issue in the revised manuscript.

Revised text (Discussion, Lines 334-341)

Similarly, the theory of “reservoir computing” suggests that the circuits required for higher order cognitive functions, such as image classification, can already exist in untrained, random recurrent neural networks. In this scenario, higher order cognitive functions can be achieved only by training the read-out network, as suggested by the lottery ticket hypothesis (Bellec et al., 2020; Verstraeten et al., 2007). While these models focus on the innate function of networks in that a high dimensional space generated by a random recurrent network can perform various tasks without learning, our current results demonstrate that the functional tuning of “single units” (comparable with neuronal tuning in biological brains) can arise in random networks without any further training of the read-out process, which is distinguished from the main idea of the reservoir computing model.

Concern 7: Continued conflation of the terms “innate” and “spontaneous”

There continues to be some misunderstanding between the concepts of “innate” and “spontaneous.” Innate is a broader term (one way to be innate is to be spontaneous), another way to be innate is to be programmed by evolution. Proponents of innateness for faces seek to explain why face preferring neurons consistently show up in the same locations across individuals of the same species. Spontaneousness cannot account for that. However, compelling evidence for spontaneous emergence is helpful for the innateness argument as it reduces the burden from explaining how face preferences can exist at all and why in

the same locations, to just explaining why it shows such regularity. On this point there is a schism between the introduction, results, and discussion. The introduction is still beleaguering a strong innateness argument whereas the discussion seems to acknowledge that the evidence is incomplete.

a. The introduction and abstract should be rewritten to clarify the two aspects of the innateness argument and to orient the reader as to which aspect the paper has evidence for.

We regret that our previous manuscript did not provide a careful description of this issue. In the revised manuscript, we used the terms “innate” and “spontaneous” distinctively and appropriately to specify different concepts of innateness, specifically in the abstract, introduction, and discussion. We explicitly demonstrated that our results imply innate face-selectivity that arises spontaneously from feedforward wirings, but this requires further evidence and examinations before one can argue that this mechanism can be indeed considered spontaneous.

Specifically, our model suggests that random weights in CNNs (comparable with random feedforward projections in biological networks) can generate selective responses of units to face images. Then, considering that the development of random weights does not require any training or experience, this could be considered as “innate” face-selectivity. Importantly, these results imply a possible scenario in which the random feedforward connections that develop in young animals may be sufficient for initializing primitive face-selectivity. Accordingly, we argue that this face-selectivity can be considered to arise “spontaneously” under the assumption that random feedforward wirings can arise without a complicated process. In this scenario, the emergence of face-selectivity may not require genetically programmed innateness but may simply originate from the statistical complexity of random wirings.

However, it must be noted that the alternative interpretation that the observed face-selectivity is innate but may not be considered spontaneous because the initial development of random feedforward wiring requires a certain mechanism programmed in genes is also possible. Particularly, regarding the question of why face-selective neurons are observed consistently in the same brain regions, the role of programmed genes may be critical — face-selective neurons can simply originate from random feedforward wirings in hierarchical networks, but this neuronal selectivity can only be refined and reserved in a particular face-patch region in the brain, most likely controlled by a blueprint of the brain circuitry programmed in genes. Detailed arguments pertaining to this issue may be possible when further evidence of the developmental mechanism of random wirings and the corresponding dependence on gene coding becomes available. In the revised manuscript, we describe these issues of the innateness and spontaneousness of observed selectivity, and we rewrote the abstract, introduction, and discussion accordingly.

Revised text (Introduction, Line 19-86)

The ability to identify and recognize faces is a crucial function for social behavior, and this ability is thought to originate from neuronal tuning at the single or multi-neuronal level^{1–20}. Neurons that selectively respond to faces (face-selective neurons) are observed in various species^{21–23}, and they have been considered as the building blocks of face detection¹⁸. The observation of this type of intriguing neuronal tuning in the brain has inspired neuroscientists, raising important questions about its developmental mechanism — whether face-selective neurons can arise innately in the brain or require visual experience, and whether neuronal tuning to faces is a “special” type of function

distinctive from tunings to other visual objects.

Regarding the emergence of neuronal face-selectivity, previous studies have suggested a scenario in which visual experience develops face-selective neurons^{24–26}. The experience-dependent characteristics of face-selective neurons imply that visual experience plays a critical role in developing face-selectivity in the brain. It was observed that the preferred feature images of face-selective neurons in adult monkeys are those that resemble animals or familiar people depending on individual experiences²⁵. Another study of the inferior temporal cortex (IT) in monkeys reported that robust tuning of face-selective neurons is not observed until one year after birth⁶ and that face-selectivity relies on experience during the early infant years. It was also reported that monkeys raised without face exposure did not develop normal face-selective domains²⁶. However, another view suggests that face-selectivity can innately arise without visual experience^{27–34}. Although visual experience is critical for refining the development of face-selective neurons, several lines of research have demonstrated that primitive face-selectivity is observed even before visual experience^{27–31}. Primate infants behaviorally prefer to look at face-like objects as opposed to non-face objects^{32–34}, implying that face-encoding units may already exist in infants. Moreover, visual category-selective domains, including the face, are observed in the ventral visual stream in adult humans who have been blind since birth^{28,29}. Furthermore, a recent study reported that face-selective neurons are observed in infant animals and that the spatial organization of such early face-selective regions appeared similar to that observed in adults⁶. These results taken together imply that face-selective neurons can arise before visual experience, in contradiction to the first scenario.

There has been another important debate as to whether face-selectivity is a special type of visual function distinguished from other processes of object recognition, the developmental mechanism of which needs to be considered and examined distinctively from other visual neural tunings. After early observations of the face-selective responses of single neurons in the IT, face detection has been considered one of the most important visual functions necessary for the survival of social animals^{35–38}. Observation of the fusiform face area (FFA), which is specialized for face-recognition, also reinforced the idea that face-selectivity is a specialized neuronal tuning, which may develop differentially from cognition of other general visual objects^{12,39–42}. However, more recent studies have reported that selectivity to objects such as a car or a bird can also develop in the FFA from visual experience^{43,44}, implying that faces may not be a special, distinct type of object class for visual function and that neuronal tuning to various visual objects can also arise similarly to selectivity to faces.

The argument concerning these issues, which reveals our incomplete understanding of face-selectivity, likely stems from limitations regarding the control of the experimental conditions, as it is impossible to control the amount of visual experience for a particular category, such as face, in individual subjects. Even if the subjects are visually deprived such that they are prevented from having visual experience, the portion of category-selective neurons and their degree of tuning may vary across subjects and cannot easily be predicted. These various factors make it difficult to investigate the developmental mechanism of face- and other object-selective neurons in the brain.

A model study using biologically inspired artificial neural networks, such as deep neural networks (DNNs)^{45,46}, may offer an effective approach to the problem in this case^{47–50}. Recently, DNNs, a stack of biologically inspired

feedforward projections with a linear-nonlinear neural motif, have provided insight into the underlying mechanisms of brain functions, particularly with regard to the development of various functions for visual perception^{47,48,51}. For example, a recent model study reported that the neural response of the monkey IT cortex could not only be predicted by the responses of DNNs trained to natural images^{47,48} but could also be controlled by the preferred feature image generated by the DNN model⁵². Notably, previous studies using random hierarchical networks provide important clues about the origin of innate face-selectivity in untrained neural networks. It was reported that untrained feedforward networks can initiate various cognitive functions with random weights and that a random network can perform image classification tasks in that way as well^{53–56}. It was also reported that a randomly initialized convolutional neural network could reconstruct corrupted images without any training, which implies that a random network can provide a priori information about the low-level statistics in natural images⁵⁷. Overall, such observations suggest the possibility of the emergence of innate cognitive functions, such as primitive face-selectivity in untrained, random hierarchical networks. However, the details of how this innate function emerges in untrained neural networks are not yet understood.

Herein, we show that face-selective units (model neurons) can arise in completely untrained hierarchical neural networks. Using AlexNet⁴⁵, a model reproducing the structure of the ventral stream of the visual cortex, we found that face-selectivity can emerge robustly across different conditions of randomly initialized deep neural networks. We found that their face-selectivity indices (FSI) are comparable to those observed with face-selective neurons in the brain. The preferred feature images obtained from the reverse-correlation method and the generative adversarial network show that face-selective units are selective for a face-like configuration, distinct from units with no selectivity. Furthermore, we found that face-selective units enable the network to perform face detection. Intriguingly, we found that units selective to various non-face objects can also arise innately in untrained neural networks, implying that face-selectivity may not be a special type of visual tuning and that selectivity to various object classes can arise innately in untrained deep neural networks, spontaneously from random feedforward wirings. Overall, our results imply a possible scenario in which the random feedforward connections that develop in early, untrained networks may be sufficient for initializing primitive face-selectivity as well as selectivity to other visual objects in general.

Revised text (Discussion, Lines 355-373)

Importantly, our results imply that innate face-selectivity may arise spontaneously from feedforward wirings, but this requires further evidence and examinations before one can argue whether the mechanism can indeed be considered spontaneous. Specifically, our model suggests that the random weights in CNNs (comparable with random feedforward projections in biological networks) can generate selective responses of units to face images. Then, considering that the development of random weights does not require any training or experience, this could be considered as “innate” face-selectivity. These results imply a possible scenario in which the random feedforward connections that develop in young animals may be sufficient for initializing primitive face-selectivity. Arguably, this face-selectivity can be considered to arise “spontaneously” under the assumption that random feedforward wirings can arise without a complicated process. In this scenario, the emergence of face-selectivity may not require

genetically programmed innateness but may simply originate from the statistical complexity of random wirings. However, it must be noted that an alternative interpretation is also possible: the observed face-selectivity is innate but may not be considered spontaneous because the initial development of random feedforward wiring requires a certain mechanism programmed in one's genes. Particularly, regarding the question of why face-selective neurons are observed consistently in the same brain regions, the role of programmed genes may be critical — face-selective neurons can simply originate from random feedforward wirings in hierarchical networks, but this neuronal selectivity can only be refined and reserved in a particular face-patch region in the brain, most likely controlled by a blueprint of the brain circuitry programmed in genes. Detailed arguments pertaining to this issue may become possible when further evidence of the developmental mechanism of random wirings and the corresponding dependence on gene coding become available.

Minor issues

a. Fig. 3 Analysis: the double-dipping argument from Reviewer #2 was only partially addressed: this information should be included in the Figure legend.

We apologize for the incomplete description of the figure legend. To avoid confusion, we revised the figure legend and now described how the data for training and for testing were sampled exclusively and thus do not share any single images in common.

Revised text (Fig. 3 caption, Line 769-774)

a. Design of the face detection task and SVM classifier using the responses of the untrained AlexNet. During this task, face or non-face images were randomly presented to the networks and the observed response of the final layer was used to train a support vector machine (SVM) to classify whether the given image was a face or not. Among 60 images from each class (face, hand, horn, flower, chair, and scrambled face) that were not used for face unit selection, 40 images were randomly sampled for the training of the SVM and the other 20 images were used for testing.

b. Figure R2Q3 should be in the main text

As suggested, we included materials in R2Q3 in the main figure and text.

Revised text (Results, Line 279-303)

Emergence of selectivity to various objects in untrained deep neural networks

Lastly, we investigated the possibility that units selective to various objects other than faces also emerge similarly in untrained neural networks. For this, we measured the responses of units in random networks to a stimulus dataset of ImageNet containing 1,000 classes of objects (**Fig. 5a**). As a result, we found that selective units are observed in 39 classes among the 1,000 classes (**Figs. 5b, c**). From the analysis using “scrambled” and “texform” control images, we confirmed that these object-selective units are not particularly selective to local image parts but are selective to a whole object, similar with units selective to faces (**Fig. 5b**, inset). This result suggests that units selective innately to various objects such as faces can emerge in untrained neural networks.

Next, to investigate the emergent condition of units selective to each object further, we sorted those 39 classes according to the number of selective units observed and computed the PFI using the reverse correlation method (**Figs. 5c, d**). In general, we observed a tendency in which the PFIs of the “large number” group showed a relatively simple configuration of each preferred object class that was visually observable (**Fig. 5d**, top), whereas those of the “small number” classes represented a more complicated structure of the PFI (**Fig. 5d**, bottom). From this result, we hypothesized that objects with a simple configuration, such as faces, can induce stronger clustering in the latent space representation than those of other object classes and therefore may have a greater likelihood of generating units selective to them. To validate our hypothesis, we used the dimension-reduction method (PCA⁷⁷) to compare a clustered representation of each object class in terms of the raw pixel values and in the responses of Conv5. For quantification of the representational clustering of each class, we measured the Silhouette index⁷⁸ to estimate the consistency of data clustering. We found that there is a strong correlation between the Silhouette index in the Conv5 latent space and the number of selective units observed (**Figs. 5e-g**, Pearson correlation coefficient, $n_{\text{Net}} = 50$, $r = 0.58$, $p = 7.5 \times 10^{-5}$). This result demonstrates that objects with a simple profile, readily distinguishable from those of other objects statistically, lead to a strong clustering of abstracted responses in the DNN and are more likely to generate units selective to it. Thus, this implies that the observed face-selectivity may not be a special case of tuning, whereas selectivity to other visual objects can also arise in random networks simply due to the relatively simple configuration of the corresponding geometric components.

c. Line 352, “face-deprive” is missing a d

Thank you for pointing this out. We fixed this typo.

d. Supp. Figure 12. y-label - ratio is not a percent (%)

We revised the figure and corrected the y-label accordingly. All values are in ratio in the revised figure.

Revised figure

e. Line 301: face units are “scarcely found in Conv1”; conv1 can only implement linear filters. This comment may suggest to readers that compound feature detectors may exist in conv1 when Conv1 is actually severely limited in what it can do for reasons that are completely known.

We appreciate this comment and regret that this description was incorrect and misleading. From our additional analysis (for details, please see our response to your question concern 4a), we confirmed that face-selective units are not observed in Conv1. Following this result, we removed this sentence from the revised text.

f. Line 13 is ungrammatical. It should read: “across different random initializations of the network” (delete the “conditions of” and pluralize “initialization”). There are other definite article and subject-verb agreement mistakes throughout.

In the revised manuscript, we revised the abstract and removed the term accordingly.

References

- Bellec, G., Scherr, F., Subramoney, A., Hajek, E., Salaj, D., Legenstein, R., and Maass, W. (2020). A solution to the learning dilemma for recurrent networks of spiking neurons. *Nat. Commun.* *11*, 1–15.
- Chidester, B., Do, M.N., and Ma, J. (2018). Rotation Equivariance and Invariance in Convolutional Neural Networks.
- Cohen, J. (1988). *Statistical power for the behavioural sciences*. Hillsdale, NY Lawrence Erlbaum.
- Cureton, E.E. (1956). Rank-biserial correlation. *Psychometrika* *21*, 287–290.
- Fukushima, K., and Miyake, S. (1982). Neocognitron: A new algorithm for pattern recognition tolerant of deformations and shifts in position. *Pattern Recognit.* *15*, 455–469.
- Gauthier, I., Skudlarski, P., Gore, J.C., and Anderson, A.W. (2000). Expertise for cars and birds recruits brain areas involved in face recognition. *Nat. Neurosci.* *3*, 191–197.
- Kanwisher, N. (2000). Domain specificity in face perception. *Nat. Neurosci.* *3*, 759–763.
- Kanwisher, N., McDermott, J., and Chun, M.M. (1997). The fusiform face area: a module in human extrastriate cortex specialized for face perception. *J. Neurosci.* *17*, 4302–4311.
- Kanwisher, N., Tong, F., and Nakayama, K. (1998). The effect of face inversion on the human fusiform face area. *Cognition* *68*, 1–11.
- Kavukcuoglu, K., Sermanet, P., Boureau, Y.L., Gregor, K., Mathieu, M., and LeCun, Y. (2010). Learning convolutional feature hierarchies for visual recognition. *Adv. Neural Inf. Process. Syst.* *23* 24th Annu. Conf. Neural Inf. Process. Syst. 2010, NIPS 2010 1–9.
- LeCun, Y. (2012). Learning invariant feature hierarchies. In *European Conference on Computer Vision*, (Springer), pp. 496–505.
- LeCun, Y., and others (1989). Generalization and network design strategies. *Connect. Perspect.* 143–155.
- LeCun, Y., Fu Jie Huang, and Bottou, L. (2004). Learning methods for generic object recognition with invariance to pose and lighting. In *Proceedings of the 2004 IEEE Computer Society Conference on Computer Vision and Pattern Recognition, 2004. CVPR 2004.*, (IEEE), pp. 97–104.
- Maaten, L. van der, and Hinton, G. (2008). Visualizing data using t-SNE. *J. Mach. Learn. Res.* *9*, 2579–2605.
- Rhodes, G., Byatt, G., Michie, P.T., and Puce, A. (2004). Is the Fusiform Face Area Specialized for Faces, Individuation, or Expert Individuation? *J. Cogn. Neurosci.* *16*, 189–203.
- Srivastava, M., and Grill-Spector, K. (2018). The Effect of Learning Strategy versus Inherent Architecture Properties on the Ability of Convolutional Neural Networks to Develop Transformation Invariance.
- Tarr, M.J., and Gauthier, I. (2000). FFA: A flexible fusiform area for subordinate-level visual processing automatized by expertise. *Nat. Neurosci.* *3*, 764–769.
- Tsao, D.Y., and Livingstone, M.S. (2008). Mechanisms of Face Perception. *Annu. Rev. Neurosci.* *31*, 411–437.
- Vermeesch, P. (2013). Multi-sample comparison of detrital age distributions. *Chem. Geol.* *341*, 140–146.
- Verstraeten, D., Schrauwen, B., D'Haene, M., and Stroobandt, D. (2007). An experimental unification of reservoir computing methods. *Neural Networks* *20*, 391–403.
- Zeiler, M.D., and Fergus, R. (2014). Visualizing and understanding convolutional networks. In *European Conference on Computer Vision*, (Springer), pp. 818–833.

Reviewers' Comments:

Reviewer #3:

Remarks to the Author:

It was great to review this manuscript again. We thought the revision was vastly improved, showing a confident grasp of deep neural networks and the debate behind face selectivity in the brain, along with thoughtful restraint on the overall claims of the paper (without taking away its impact). Congratulations to the authors for this excellent work.

We believe this manuscript might be cited a lot, and with that in mind, we want to make sure that the paper holds as much benefit to the field as possible, and that is as clear and correct as it can be. In that spirit, we really think the manuscript needs further work along these points: (1) there is a glaring conceptual gap about the authors' definition of a neuron, its associated receptive field and its relationship to face tuning; (2) some of the statistical reporting and analyses are unacceptable, (3) some previous reviewer comments were not fully addressed, while other issues were claimed to be fixed but the revised text presented in the response to reviewers did not appear in the actual revised manuscript. Finally, there remains an insight gap about CNNs and will need to be addressed -- although again, this was a great overall improvement in the manuscript.

The revised manuscript does not introduce any new paradigms, and many of the points requiring additional work are partial responses to our earlier requests. Therefore, we think that the most effective format for this review is to respond in place, with the same bullet-point numbering as in our previous report but only for the concerns that continue to need clarification.

****Major concern 1: unmerited elevation of faces as a category with special properties.**

1.c.

Previously we asked that the authors use an "unconstrained" GAN (unmodified, no retraining with faces, no retraining at all, with the GAN pre-trained as downloaded). Instead, the authors retrained a GAN with a face-reduced data set. While we recognize that this was an impressive handicap for the authors to adopt, any retraining presents the opportunity for training mistakes. The authors should report and compare XDream PFIs using GANs with no additional training or modification. How many PFIs did the authors generate, and what the rationale for this number (note: this can be addressed along with Sections 2.a,b)?

1.e.

We respect the amount of work that went into validating the t-SNE results before ultimately adopting the PCA analysis. However, there are three persistent problems with the way the silhouette index analysis is carried out.

Persistent problem 1: In the last submission supplemental figure 6.f showed the silhouetted index for a number of categories. An analogous figure using t-SNE is in the new results, we've copy -and- pasted the figures into the accompanied PDF (Figure R1e). According to the description, little has changed in the analysis (both plots are t-SNE on the top 39 categories + 1, the top and bottom categories are the same, etc.). In the revised figure, all are positive while in the original all but one were negative. Why has the silhouette index for all the categories changed so drastically? Please account for this change.

Persistent problem 2: In the last review we wrote that silhouette index analysis should be carried out with "moderate dimensional data". This means using higher dimensionality than the scatter plot and is a compulsory component of any embedding and dimensionality reduction analysis. Plots are limited to 1-3 dimensions; analysis needs to be done with an embedding that retains the maximum amount of relevant information. It appears that the silhouette index analysis was only carried out on the first two principal components. Only 32% of the variance is contained in the first 2 dimensions, thus **most** information is excluded!

Please set a (less-arbitrary) high threshold on the percentage of variance contained (e.g. 75%)

and repeat the silhouette analysis with the minimum number of dimensions necessary to capture that variance. Alternatively, one can use a completely non-arbitrary way to set the number of dimensions (such as the broken-stick <https://blogs.sas.com/content/iml/2017/08/02/retain-principal-components.html>). If the silhouette analysis is in total disagreement with the previous results, then lower the dimensionality until results agree and report both findings.

Persistent problem 3: Why is it significant that the silhouette index correlates with number of units (figure 5.g)? The authors go through lengths to point out this correlation but don't offer much interpretation. Don't we already know more units implies better encoding and thus better silhouette index? The correlation is either important and the authors need to explain why, or it is not necessary to mention at all.

****Major concern 2: Definition of "unit"**

2.a & 2.b.

Why do units with eccentric receptive fields (i.e., RFs located at the edges) have face selectivity in the center of the image? In our last communication, we expressed concern that the authors defined face-selective units as depending on spatial position, even though each spatial position in the image is convolved with the same filter ("weight sharing"). In response, the authors showed that selectivity was not intrinsic to a given filter, but to the interaction of the filter with the image spatial position (Supplementary Figure 2). This is where it gets confusing: in Supp. Fig. 2, the example face-selective unit is near the upper right corner of the image. Units in conv5 have receptive fields. They are large, but they do not cover the full image (the authors illustrate RF sizes in the 'rebuttal' document, page 36). This unit's RF is centered in the upper right, yet it appears it was selected based on its responses to a stimulus set where faces appear most frequently in center of the picture. Also, all shown PFIs appear to generate faces in the center of the picture (Fig. 2c). This makes no sense to us. If the units are only face-selective at specific positions, then their PFIs should appear within their RFs (unless they are encoding not global face configurations, only face parts).

The same filter at two spatial locations is *exactly identical*, if it is face-selective at one location but not another then there are only two possible explanations: 1) The stimulus set did not show all possible faces at all possible locations thus the difference is due to the choice of stimuli. 2) When filters are applied to edges the integer padding of the images impacts filter output regardless of the choice of stimuli. This suggests that some results may still be over-interpreted. Particularly lines 105-121: "This suggests that face-selective units are not dominantly generated by a particular filter". Doesn't it actually suggest that the authors' face stimuli do not guarantee that a face is located at each filter location? How can a filter near the edge (which is padding-impacted) be *more* face selective (supplementary figure 2) than an identical filter at the center? Please clarify these questions.

Please add RF sizes to the Supplemental Table, especially since in Supplemental Figure 5, the x-axis units for translation are in rRF.

Supp. Fig. 2.b shows that channel 247 of conv 5 is selective for faces at only a couple locations. Please show a distribution of face-selective unit locations across all channels. This could be another heatmap or a scatter plot over x and y.

Please show PFIs corresponding to face-selective units in different locations. Show a version of these PFIs with and without the units' RF superimposed over the PFI.

Please add in the main manuscript an explanation of the effect of RF position on the selectivity and/or of the unit. For example, when units had eccentric RFs, why did they have selectivity to image sets where the faces were mostly at the center? Did these have as much selectivity (pick your effect measure) as those in the center?

Only one face PFI is shown throughout the paper. It is customary to show more than one, either in

a supplementary figure where multiple examples are shown for each category, or to exploit the fact that the paper has multiple figures and to choose different PFIs for each figure. Please do so.

****Major Concern 3: Invariance**

3.a.

The authors did a good job summarizing what is known about invariance in CNNs and elucidating the role of spatial structures in the filters. No changes to the manuscript nor additional works are needed. However, we offer insights the authors may find useful in the future. Sometimes "effective range" can mean the average *minimum* distance between *local* maxima and minima of filter outputs, or it can mean what it does in this analysis: the average degradation at a given distance. Both are justifiable depending on whether one studies smoothness of filter weights or of their outputs. A sinusoid filter has high spatial structure and is smooth, but its output is not smooth since translating a strong edge by $\frac{1}{2}$ the period will produce the opposite response. The banding effect is related to theories about why adversarial attacks are possible and CNN invariance is less than hoped for (wrong *type* of spatial structure), but the authors choice of smoothness and effective range are justified and consistent with their needs.

3.b.

We admire the effort that went into properly contextualizing viewpoint invariance and the result that viewpoint invariance is greater in trained networks provides quality context on the role of training. However, there are two persistent problems.

Persistent problem 1: The definition of "viewpoint invariant unit" is confusing, and possibly circular. Supplemental Figure 9 defines a viewpoint invariant neuron as "Units in Conv5 that receive projections from viewpoint-specific units in Conv4 with uniformly distributed weights". Since your hypothesis is that weights cause viewpoint invariance, one would expect viewpoint invariance to be defined with respect to neural output, not the weight distribution. Is this definition circular? Panel "e" shows a strong difference in responses shouldn't that be used instead?

Persistent problem 2: How is it possible that Supplemental Figure 9 panel h and f are identical? Furthermore, how come they are not significant in f (asterisked) but significant in h. The "n.s." bars seem misapplied throughout the manuscript with inconsistent asterisking. Most important, double check and explain why the indicated figure panels are perfect matches and correct the asterisking and n.s. labeling ****throughout the manuscript**** (we pasted the problematic Figure in the attached PDF, Figure R3b).

****Major Concern 5: Statistical tests and reporting**

5.a.

The p-value reporting in the captions and main text remains unclear, and much of the reporting is inaccurate, not satisfying the submission checklist. The submission checklist indicates that the authors' provided p values and effect sizes for all hypothesis tests. They attempted this by providing a table. First, the effect size name is missing from the table header. Reporting is incomplete and there is nothing to indicate or link to the table with the text reporting which is still the old way (e.g. $p < 0.05$) (see <https://www.vox.com/science-and-health/2017/7/31/16021654/p-values-statistical-significance-redefine-0005> and <http://verso.mat.uam.es/~pablo.fernandez/Fisherb.pdf>, about why this is discouraged). Thus, it is difficult to quickly know whether it is reported. For example, Supplemental Figure 8 panel a cites two one-way ANOVAs but zero one-way ANOVAs are reported in table S4 for figure S8 (also there is no panel a in the table). Additionally, it is not clear what divisor is used for Bonferroni correction, sometimes there is "n" and also "groups". Is "n" the number of experiments (e.g. number of times the test was used) or is "n" the number of units? Units are expected to number in the hundreds of thousands! We struggled to sort out what Bonferroni divisor was used. The usual practice is to cram all that information from the table into the parenthetical phrase (e.g. "rrbc=0.773 and $p=0.003$ for ranksum test, Bonferroni significance with $n=5$ is $p < 0.01$ " per the

examples at the bottom of https://my.ilstu.edu/~mshesso/apa_stats.htm). A novel solution that is more readable than the cluttered standard reporting method is to label each row in the table and referencing the table and row in your parenthetical phrase, (e.g. "p<0.05 Bonferroni adjustment n=5, see Table S4.R82" or simply "p<0.05/5, see Table S4.R82"). Editors or other reviewers may not like that, in which case it should follow examples from the link.

Please scrutinize your tables for further errors as it appears not to satisfy the submission checklist. Clearly state what the Bonferroni adjustment is (e.g. 0.05/10) for every p-value reported. Either reference specific rows of the table within each parenthetical phrase or compactly report Bonferroni adjustment N values, as well as p values and effect sizes to 2-3 significant figures in each parenthetical phrase (which was the intent of our original request).

As previously requested, the authors must explain why some p values have asterisks to their left (e.g. *p = 1.5 × 10⁻² on line 126 but **p < 2.7 × 10⁻² on line 183) and be consistent about equality vs inequality symbols. Please explain all of the non-standard notations and asterisking.

When the authors define a "face-selective unit" they cite a paper that does permutation testing, yet the authors here do not. They use a ranksum test with no correction for multiple comparisons. We believe we can rationalize this choice (see AUC, https://en.wikipedia.org/wiki/Receiver_operating_characteristic#Area_under_the_curve), but the authors do not disclose how and why they deviate from the cited work, nor justify their thresholding in the definition of a face-selective unit. Please clarify these points.

It would also be helpful to see the mean/median FSI per convolutional layer, as a way to characterize the emergence of selectivity (but this is a minor point).

5.b.

The Kolmogorov-Smirnov tests are not reported in the main text as claimed. Lines 220-225 do not match the response to the reviewers. Furthermore, we cannot find the values in the supplementary tables.

Please use the KS test in addition to either ANOVA, the signed-rank, or the rank sum test any time the manuscript claims that two populations are not distinct, or justify that only the difference in medians matters not the consistency/reliability. Also match all statistics reported in the text to a table or just complete the reporting in the parenthetical phrases.

To clarify our request, let us elaborate about why we asked for two tests every time there is a claim that two populations are the same. A rank-sum or signed-rank test can only detect differences in medians. Two random variables, one that is unimodal, and one that is bi-modal may appear the same to a signed-rank or rank-sum test, whereas a KS test would be more likely to tell them apart. Conversely if two datasets are identical except the second has had an infinitesimal value added to it, then the KS test would fail to distinguish them, but a signed-rank test would be able to distinguish them. This is why, one needs two tests any time they claim two populations are not distinct. In summary: the rank-sum or signed-rank tests capture differences in median performance, the KS test captures differences in consistency, it is possible to have the same median performance but much less consistency.

5.c.

It is good that the authors switched to a rank-sum test and it also shows a strong difference, but why are the p values and effect sizes reported differently for figure 3 panel c in the supplemental table. What are the inequality signs signifying?

Additionally, we will caution that the reason a KS test is inappropriate is not that it is for continuous distributions. Classification performance should be continuous anyway. The reason KS tests are inappropriate to judge difference in *performance* is that KS is sensitive to factors other than median performance. For example, if SVMs trained with face selective units had much higher

variability, but the median value was indistinguishable, the KS test would tell one they were different but a rank-sum test would tell one they are not. Since the median, and not the variance is the most important factor then the KS test is not appropriate.

5.d.

Standard practice is to introduce these effect sizes early in the manuscript and give them symbols e.g. r_{bcb} for rank-biserial correlation, and f^2 for Cohen's f^2 , etc. Then one reports them in the parenthetical phrase where each p-value listed. We are aware that the style of " $p > 0.05$ " used to be widespread and not everyone has updated their practices but there have been a number of big problems and failures after re-analysis that justify the push of topflight journals to have authors abide by those practices (see https://my.ilstu.edu/~mshesso/apa_stats.htm). Unfortunately, it falls on reviewers to actually check that authors have followed the check list. Given that we've found a number of inconsistencies it appears that more work is needed to bring the necessary attention to details to the statistical reporting.

The authors have done good work on the rest of the manuscript and we only need explanations, or small reanalysis for the other parts. Now we ask that particularly close attention be paid to the statistical reporting and choice of tests.

****Major Concern 6: Missing important related works**

6.a.

This is satisfactory, but there are some points to clarify. The readout of a reservoir network can be arbitrary or chosen in advance based on knowledge of the system. So, selecting "activation maximization" as the readout mechanism in advance is consistent with the reservoir computing concept. More fundamentally, the authors train an SVM coupled with a feature engineering mechanism (SVM on algorithmically identified face units). This SVM work was what connects it to reservoir computing most strongly, maybe other readers will notice that.

Minor.c,

Now it is line 277 but "face-deprive" is still missing that d.

Minor.d,

Some apparent issues with the "n.s." asterisking. It is not the same notation from figure to figure and as pointed out above, some figures are indistinguishable but have different n.s. asterisks. Please check your figures reference the correct data and that asterisking is consistent. For example, figure S5 has single and double asterisks, but these are not explained.

Minor.f

These corrections were not carried out correctly. The sentence is still ungrammatical, with a noun where an adjective should go and a definite article where an adverb should go.

Line 13 should have been changed like so:

"across different conditions of random initialization of the network" -> "across different random initializations of the network"

Instead it was changed like so:

"across different conditions of random initialization of the network" -> "in random initialized networks and these units reproduce the characteristics observed in monkeys."

Now the simplest change is like so:

"in random initialized networks and these units reproduce the characteristics observed in monkeys." -> "in randomly initialized networks and these units reproduce many characteristics observed in monkeys."

Minor.g

the word "translation" is misspelled throughout the manuscript as "translatation"

Reviewer #5:

None

Response to Reviewers

We sincerely appreciate your enthusiasm devoted to reviewing our manuscript. We believe that our manuscript was noticeably improved with your helpful comments and considerate suggestions. Based on the questions and comments provided, we made appropriate revisions with regard to the remaining issues. We are confident that our revised manuscript fully addresses the issues. Please find our responses to each comment below. Thank you very much for your kind consideration.

Reviewer #3

It was great to review this manuscript again. We thought the revision was vastly improved, showing a confident grasp of deep neural networks and the debate behind face selectivity in the brain, along with thoughtful restraint on the overall claims of the paper (without taking away its impact). Congratulations to the authors for this excellent work.

We believe this manuscript might be cited a lot, and with that in mind, we want to make sure that the paper holds as much benefit to the field as possible, and that is as clear and correct as it can be. In that spirit, we really think the manuscript needs further work along these points: (1) there is a glaring conceptual gap about the authors' definition of a neuron, its associated receptive field and its relationship to face tuning; (2) some of the statistical reporting and analyses are unacceptable, (3) some previous reviewer comments were not fully addressed, while other issues were claimed to be fixed but the revised text presented in the response to reviewers did not appear in the actual revised manuscript. Finally, there remains an insight gap about CNNs and will need to be addressed although again, this was a great overall improvement in the manuscript.

The revised manuscript does not introduce any new paradigms, and many of the points requiring additional work are partial responses to our earlier requests. Therefore, we think that the most effective format for this review is to respond in place, with the same bullet-point numbering as in our previous report but only for the concerns that continue to need clarification.

****Major concern 1: unmerited elevation of faces as a category with special properties.**

Q1. (I.c. in previous revision) Previously we asked that the authors use an “unconstrained” GAN (unmodified, no retraining with faces, no retraining at all, with the GAN pre-trained as downloaded). Instead, the authors retrained a GAN with a face-reduced data set. While we recognize that this was an impressive handicap for the authors to adopt, any retraining presents the opportunity for training mistakes. The authors should report and compare XDream PFIs using GANs with no additional training or modification. How many PFIs did the authors generate, and what the rationale for this number (note: this can be addressed along with Sections 2.a,b)?

As suggested, we replaced the preferred feature images (PFIs) with those generated by an “unconstrained” GAN (Dosovitskiy and Brox, 2016) (i.e., unmodified, no retraining with faces, no retraining at all, with the GAN pre-trained as downloaded) throughout the revised manuscript. We confirmed that those new PFIs show face configurations similar to those previously generated by a GAN trained with face-reduced datasets (Q1). The PFIs were generated for all selective units for each category; thus, the number of PFIs is identical to the number of face-selective units

under each condition. In the revised manuscript, we illustrate examples of these new PFIs. Please find our extended discussion on this issue in section Q3 (2a,b in the previous revision) for details.

Q1

Revised text (Methods, Line 496)

We used a GAN (Dosovitskiy and Brox, 2016) pre-trained with natural images (ILSVRC 2010).

(I.e. in previous revision) We respect the amount of work that went into validating the t-SNE results before ultimately adopting the PCA analysis. However, there are three persistent problems with the way the silhouette index analysis is carried out.

Q2-1. Persistent problem 1: In the last submission supplemental figure 6.f showed the silhouetted index for a number of categories. An analogous figure using t-SNE is in the new results, we've copy -and -pasted the figures into the accompanied PDF (Figure R1e). According to the description, little has changed in the analysis (both plots are t-SNE on the top 39 categories + 1, the top and bottom categories are the same, etc.). In the revised figure, all are positive while in the original all but one were negative. Why has the silhouette index for all the categories changed so drastically? Please account for this change.

We apologize for missing detailed information regarding the revised analysis. In the previous manuscript, we revised the definitions of the intra- and inter-class distances, following a more general form for each term (Handl et al., 2005); this caused a rescaling of the range of the silhouette index (e.g., some negative values became positive). In the current manuscript, we clarified the new definitions of the intra- and inter-class distances as well as the silhouette index.

The silhouette index for the i^{th} point was defined as $c_i = (b_i - a_i) / \max(a_i, b_i)$, where a_i and b_i refer to the intra-class distance and the inter-class distance for the i^{th} point, respectively. Previously, due to the unmatched (or arbitrarily chosen) scales of the intra- and inter-class distances (**Q2-1a**), the intra-distance tends to appear much longer than the inter-distance (**b**). This induced bias in the silhouette index toward the negative direction, as found in the previous results (**c**).

Q2-1

In the revised analysis, we used new, scaled definitions of the intra- and inter-class distances, following a more general form for each term (Handl et al., 2005) (d). With these new definitions, the intra- and inter-class distances are properly calibrated to be on a comparable scale (e). As a result, the silhouette index appeared as positive values in most conditions (f). In the current manuscript, we describe these definitions and the references for each term used to calculate the silhouette index in detail.

Revised text (Methods, Line 467-470)

The silhouette index SI (Kaufman and Rousseeuw, 2009) for the i^{th} point is defined as $c_i = (b_i - a_i) / \max(a_i, b_i)$, where a_i and b_i refer to the intra-class distance and the inter-class distance for the i^{th} point, respectively. The intra-class distance was defined as the average distance between the centroid of the class and each data point in the class. The inter-class distance was defined as the average distance between the centroids of each cluster (Handl et al., 2005).

Q2-2. Persistent problem 2: In the last review we wrote that silhouette index analysis should be carried out with “moderate dimensional data”. This means using higher dimensionality than the scatter plot and is a compulsory component of any embedding and dimensionality reduction analysis. Plots are limited to 1-3 dimensions; analysis needs to be done with an embedding that retains the maximum amount of relevant information. It appears that the silhouette index analysis was only carried out on the first two principal components. Only 32% of the variance is contained in the first 2 dimensions, thus *most*

information is excluded! Please set a (less-arbitrary) high threshold on the percentage of variance contained (e.g. 75%) and repeat the silhouette analysis with the minimum number of dimensions necessary to capture that variance. Alternatively, one can use a completely non-arbitrary way to set the number of dimensions (such as the broken-stick <https://blogs.sas.com/content/iml/2017/08/02/retain-principal-components.html>). If the silhouette analysis is in total disagreement with the previous results, then lower the dimensionality until results agree and report both findings.

We apologize for the incomplete descriptions of our previous analysis regarding these issues. Indeed, we already performed our analysis as suggested: the results in Figs. 5f and 5g were gained when using all 3,999 principal components. (We confirmed that 100% of the variance is contained in this case and that 75% of the variance is contained when 140 ± 32 components were used.) However, only two PCs are shown in Fig. 5e for display purposes. In the revised manuscript, we describe the exact number of PCs used for each analysis.

Revised figure (Fig. 5)

Revised text (Fig. 5 caption)

e. Visualization of the PCA⁷⁷ analysis results (only two PCs are shown) using the Conv5 unit responses to each class in untrained networks. The analysis was performed using 3,999 principal components, and the top 140 ± 32 components contained 75% of the variance. **f.** The silhouette index⁷⁸ of the Conv5 unit responses was measured using all principal components to estimate the consistency of data clustering. The error bar indicates the standard deviation of 50 simulations of randomly initialized networks.

Q2-3. Persistent problem 3: Why is it significant that the silhouette index correlates with number of units (figure 5.g)? The authors go through lengths to point out this correlation but don't offer much interpretation. Don't we already know more units implies better encoding and thus better silhouette index? The correlation is either important and the authors need to explain why, or it is not necessary to mention at all.

We regret that our previous manuscript did not provide a sufficient discussion of this issue. In the revised manuscript, we demonstrate that this relationship explains the chance that units selective to a particular object can be observed, both in a computational model study and possibly in experimental studies in neuroscience.

From our results showing that selective units are observed for only 39 objects among 1,000 categories in ImageNet as tested here, an important question arises: what is the condition in which units selective to a particular object can or cannot be observed in a network? The analysis of the relationship between the silhouette index and the number of units shows that the number of units increases as the silhouette index increases, and no selective units can be found below a certain threshold, which is the silhouette 0.036 in our model network (**Q2-3**, black dashed line). This result implies that there may be a threshold of the clustering level in the response embedding space, by which a unit selective to that object class can be defined and observed. In neuroscience, this may provide a possible explanation of why face-selective neurons are observed in various experiments while neurons selective to other objects are not observed as frequently. In the revised manuscript, we describe this issue in the discussion section.

Revised figure (Fig. 5)

Revised text (Results, Lines 313-325)

We found that there is a strong correlation between the silhouette index in the Conv5 latent space and the number of selective units observed (**Figs. 5e-g**, Pearson correlation coefficient, $n_{Net} = 50$, $r = 5.85 \times 10^{-1}$, $P = 7.48 \times 10^{-5}$). This result demonstrates that objects with a simple profile, readily distinguishable from those of other objects statistically, lead to a strong clustering of abstracted responses in the DNN and are more likely to generate units selective to it. Furthermore, the relationship between the silhouette index and the number of units observed shows that the number of units increases as the silhouette index increases, with no selective units observed when the

Q2-3

silhouette index is below 0.036 (**Fig. 5g**, black dashed line). This result implies that there may be a threshold of the clustering level in the response embedding space, by which a unit selective to that object class can be defined and observed. In neuroscience, this may provide a possible explanation of why face-selective neurons are observed in various experiments while neurons selective to other objects are not observed as readily. Thus, the observed face-selectivity may not be a special case of tuning, whereas selectivity to other visual objects can also arise in random networks simply due to the relatively simple configuration of the corresponding geometric components.

****Major concern 2: Definition of “unit”**

Q3. (2.a & 2.b. in previous revision) Why do units with eccentric receptive fields (i.e., RFs located at the edges) have face selectivity in the center of the image? In our last communication, we expressed concern that the authors defined face-selective units as depending on spatial position, even though each spatial position in the image is convolved with the same filter (“weight sharing”). In response, the authors showed that selectivity was not intrinsic to a given filter, but to the interaction of the filter with the image spatial position (Supplementary Figure 2). This is where it gets confusing: in Supp. Fig. 2, the example face-selective unit is near the upper right corner of the image. Units in conv5 have receptive fields. They are large, but they do not cover the full image (the authors illustrate RF sizes in the ‘rebuttal’ document, page 36). This unit’s RF is centered in the upper right, yet it appears it was selected based on its responses to a stimulus set where faces appear most frequently in center of the picture. Also, all shown PFIs appear to generate faces in the center of the picture (Fig. 2c). This makes no sense to us. If the units are only face-selective at specific positions, then their PFIs should appear within their RFs (unless they are encoding not global face configurations, only face parts). The same filter at two spatial locations is *exactly identical*, if it is face-selective at one location but not another then there are only two possible explanations: 1) The stimulus set did not show all possible faces at all possible locations thus the difference is due to the choice of stimuli. 2) When filters are applied to edges the integer padding of the images impacts filter output regardless of the choice of stimuli. This suggests that some results may still be over-interpreted. Particularly lines 105-121: “This suggests that face-selective units are not dominantly generated by a particular filter”. Doesn’t it actually suggest that the authors’ face stimuli do not guarantee that a face is located at each filter location? How can a filter near the edge (which is padding-impacted) be *more* face selective (supplementary figure 2) than an identical filter at the center? Please clarify these questions.

We appreciate this constructive argument. In brief, we confirmed that (1) the face-selective units in the periphery encode faces at an off-centered location in the filter (matching the center of the image), but (2) these “periphery” face-selective units are not observed as frequently as those in the center and also show weaker face-selectivity (i.e., a lower average value of the selectivity index).

Q3

Specifically, units with an eccentric receptive field (RF) can show face-selectivity to a face stimulus in the center because those units encode face features at off-centered locations of the filter (**a-b**). When this filter is combined with a unit in the periphery, the position of the face feature matches that of a center-located face in the stimulus image, thus generating strong responses (**a**). In this case, this filter becomes more face-selective than an identical filter at the center. On the other hand, if the face feature is located in the center of a filter (**b**), this filter would show the highest response when located at the center.

To confirm this scenario, we measured the profile of the RFs of the filters that generate face-selective responses in the periphery (**c**) and in the center locations (**d**). As expected, filters at periphery locations (**c**, left) show a face feature profile that is off-centered (**c**, right), whereas filters at the center (**d**, left) show face features at the center (**d**, right). We also found that face-selective units are observed in various eccentricities (**e**), but mostly at the center (**e**), and that the number of face-selective units decreases with the degree of eccentricity (**f**). Accordingly, we also found that the face-selectivity decreases with the eccentricity (**g**). Noticeably, even a face-selective unit located at the most peripheral position has an RF that covers more than 60% of face images at the center (**f**), which is enough to contain most features of the entire face.

In the revised manuscript, we replaced the “unusual periphery unit” example in Supplementary Fig. S2b with one that shows a “more frequently observed center unit” response to avoid any possible confusion. We also include more diverse RFs of different units of different eccentricities.

Revised figure (Supplementary Fig. S2)

Q4. Please add RF sizes to the Supplemental Table, especially since in Supplemental Figure 5, the x-axis units for translation are in rRF.

As suggested, we defined the receptive field size and the relative sizes to the face stimulus images of each layer in the revised supplemental table.

Revised text (Supplementary Table 1)

Layer	Type	Number of neurons	Kernels	Activations	RF size	RF size / total size (RF size / face size)
Input	Image input	$227 \times 227 \times 3$	Weights $11 \times 11 \times 3 \times 96$ Bias $1 \times 1 \times 96$			

Conv1	Convolution	55 × 55 × 96		ReLU and cross channel normalization	11 × 11	2.35×10^{-3} (7.39×10^{-3})
Pool1	Max pooling	27 × 27 × 96				
Conv2	Convolution	27 × 27 × 256	Weights $5 \times 5 \times 48 \times 256$ Bias $1 \times 1 \times 256$	ReLU and cross channel normalization	51 × 51	5.05×10^{-2} (1.59×10^{-1})
Pool2	Max pooling	13 × 13 × 256				
Conv3	Convolution	13 × 13 × 384	Weights $3 \times 3 \times 256 \times 384$ Bias $1 \times 1 \times 384$	ReLU	99 × 99	1.90×10^{-1} (5.92×10^{-1})
Conv4	Convolution	13 × 13 × 384	Weights $3 \times 3 \times 192 \times 384$ Bias $1 \times 1 \times 384$	ReLU	131 × 131	3.33×10^{-1} (9.14×10^{-1})
Conv5	Convolution	13 × 13 × 256	Weights $3 \times 3 \times 192 \times 256$ Bias $1 \times 1 \times 256$	ReLU	163 × 163	5.16×10^{-1} (1.00×10^0)
Pool5	Max pooling	6 × 6 × 256				
FC6	Fully Connected	1 × 1 × 4096	Weights 4096×9216 Bias 4096×1	ReLU and dropout	227 × 227	1.00 (1.00)

Q5. Supp. Fig. 2.b shows that channel 247 of conv 5 is selective for faces at only a couple locations. Please show a distribution of face-selective unit locations across all channels. This could be another heatmap or a scatter plot over x and y.

In the revised manuscript, we show multiple examples of face-selective unit distribution in each channel and the average distribution of face-selective units across all channels.

Revised figure (Supplementary Fig. S2)

Revised text (Supplementary Fig. S2 caption)

d. (Left) Distribution of face-selective units from different channels, (Right) average distribution of units across channels. e. The number of face-selective units with eccentricity. (Inset) The face-selective unit located at the most peripheral position has an RF that covers more than 60% of face images at the center. f. Face-selectivity index with eccentricity.

Q6. Please show PFIs corresponding to face-selective units in different locations. Show a version of these PFIs with and without the units' RF superimposed over the PFI.

In the revised manuscript, we show various examples of preferred feature images in different locations, both with and without receptive field superimposition.

Revised figure (Supplementary Fig. S4)

Revised text (Supplementary Fig. S4 caption)

Samples of stimulus images (left) and preferred feature images (PFI) obtained using the reverse-correlation method (middle) and X-Dream (right) of face-selective units and units selective to a non-face class. (Inset) PFIs superimposed with the RFs of the corresponding units.

Q7. Please add in the main manuscript an explanation of the effect of RF position on the selectivity and/or of the unit. For example, when units had eccentric RFs, why did they have selectivity to image sets where the faces were mostly at the center? Did these have as much selectivity (pick your effect measure) as those in the center?

In our response to Q3, we showed that units with an eccentric RF outcome can show face-selectivity to a face stimulus in the center because such units encode face features at off-centered locations of the filter. We also confirmed that these “periphery” face-selective units are not observed as frequently as those in the center and that they also show weaker face-selectivity. For details, please refer to our response to Q3.

Q8. Only one face PFI is shown throughout the paper. It is customary to show more than one, either in a supplementary figure where multiple examples are shown for each category, or to exploit the fact that the paper has multiple figures and to choose different PFIs for each figure. Please do so.

Thank you for pointing this out. In the revised manuscript, we added multiple examples of preferred feature images from different units.

Revised figure (Supplementary Fig. S4)

Revised text (Supplementary Fig. S4 caption)

Samples of stimulus images (left) and preferred feature images obtained (PFI) using the reverse-correlation method (middle) and X-Dream (right) of face-selective unit and units selective to a non-face class. (Inset) PFIs superimposed with the RFs of the corresponding units.

**Major Concern 3: Invariance

Q9. (3.a. in previous revision) The authors did a good job summarizing what is known about invariance in CNNs and elucidating the role of spatial structures in the filters. No changes to the manuscript nor additional works are needed. However, we offer insights the authors may find useful in the future. Sometimes “effective range” can mean the average *minimum* distance between *local* maxima and minima of filter outputs, or it can mean what it does in this analysis: the average degradation at a given distance. Both are justifiable depending on whether one studies smoothness of filter weights or of their outputs. A sinusoid filter has high spatial structure and is smooth, but its output is not smooth since translating a strong edge by $\frac{1}{2}$ the period will produce the opposite response. The banding effect is related to theories about why adversarial attacks are possible and CNN invariance is less than hoped for (wrong *type* of spatial structure), but the authors choice of smoothness and effective range are justified and consistent with their needs.

We sincerely appreciate your detailed comment. We believe that this information will be very helpful for our future works.

Q10. (3.b. in previous revision) We admire the effort that went into properly contextualizing viewpoint invariance and the result that viewpoint invariance is greater in trained networks provides quality context on the role of training. However, there are two persistent problems.

Q10-1. Persistent problem 1: The definition of “viewpoint invariant unit” is confusing, and possibly circular. Supplemental Figure 9 defines a viewpoint invariant neuron as “Units in Conv5 that receive projections from viewpoint-specific units in

Conv4 with uniformly distributed weights”. Since your hypothesis is that weights cause viewpoint invariance, one would expect viewpoint invariance to be defined with respect to neural output, not the weight distribution. Is this definition circular? Panel “e” shows a strong difference in responses shouldn’t that be used instead?

We apologize for our unclear descriptions in the previous manuscript. Actually, the viewpoint-invariant unit was not defined by the “weight distribution” but by the neural “response” to the stimulus of each viewpoint following the conventional definition. In the previous manuscript, there were some incorrect descriptions, as the reviewer pointed out, which may cause confusion with regard to this definition. Again, we clarify to allay concerns that the definition is not circular.

In Supplementary Figure 9e, the viewpoint-invariant and viewpoint-specific units are defined by their responses to various viewpoint images: the responses of the viewpoint-invariant units are not significantly different across viewpoint angles (one-way ANOVA, $P > 0.05$, Bonferroni adjustment $n=5$), while those of the viewpoint-specific units to the preferred viewpoint are significantly higher than those for any other angle ($P < 0.05$). Thus, after the classification of these instances of viewpoint-invariance by their responses, we examined the connections of each unit and found a uniform distribution of weights for invariant units, which are projected from Conv4 viewpoint-specific units as shown in Supplementary Figure 9f. This result suggests that the observed viewpoint-invariance may be induced by the weight distribution of the feedforward projections.

Next, to test whether the observed uniform weight distribution can develop by chance in random networks, we simulated a random wiring model (Supplementary Figure 9) in which face-selective units in Conv5 emerge from the random wiring of face-selective units in Conv4. As a result, we found that the number of units with a uniform weight distribution (virtual viewpoint-invariant units) that emerged from random wiring in the model matched that observed in untrained networks (Supplementary Figures 9g-h). In the revised manuscript, we corrected the caption of Supplementary Figure 9 accordingly.

Revised figure (Supplementary Fig. S9)

Revised text (Supplementary Fig. S9 caption)

e. Viewpoint-specific and viewpoint-invariant face units in Conv4 and Conv5 of untrained networks were selected according to their responses to a stimulus at each viewpoint: The responses of viewpoint-invariant units are not significantly different across viewpoint angles (one-way ANOVA, $P > 0.05$, Bonferroni adjustment $n=5$), while those of viewpoint-specific units to the preferred viewpoint are significantly higher than those at any other angle ($P < 0.05$).

f. The weight projections from Conv4 viewpoint-specific units to a Conv5 viewpoint-invariant unit show a uniform distribution across preferred viewpoints (one-way ANOVA, $P > 0.05$, Bonferroni adjustment $n=5$).

g. To test whether the observed uniform weight distribution can develop by chance in random networks, a random wiring model in which face-selective units in Conv5 that emerge from the random wiring of face-selective units in Conv4 was simulated.

h. (Left) A Conv5 unit is virtually considered as viewpoint-invariant if the projected feedforward weights from Conv4 viewpoint-specific units are uniform (one-way ANOVA, $P > 0.05$, Bonferroni adjustment $n=5$). (Right). The number of units with a uniform weight distribution (virtual viewpoint-invariant units) that emerged from random wiring in the model matches that observed in untrained networks ($n_{\text{Net}} = 100$, two-sided rank-sum test, NS, $P = 3.55 \times 10^{-1}$, $r_{\text{rbc}} = -6.56 \times 10^{-2}$, two-sided Kolmogorov–Smirnov test, $P = 3.44 \times 10^{-1}$, $d = 1.84 \times 10^{-2}$).

Q10-2. Persistent problem 2: How is it possible that Supplemental Figure 9 panel h and f are identical? Furthermore, how come they are not significant in f (asterisked) but significant in h. The “n.s.” bars seem misapplied throughout the manuscript with inconsistent asterisking. Most important, double check and explain why the indicated figure panels are perfect matches and correct the asterisking and n.s. labeling ****throughout the manuscript**** (we pasted the problematic Figure in the attached PDF, Figure R3b).

We apologize for our mistakes. We found that an incorrect graph was inserted in Fig. S9f and thus replaced it with a correct version. We also corrected the asterisks and n.s. labels appropriately throughout the manuscript.

Q10-2

Revised figure (Supplementary Fig. S9)

**Major Concern 5: Statistical tests and reporting

Q11. (5.a. in previous revision) The p-value reporting in the captions and main text remains unclear, and much of the reporting is inaccurate, not satisfying the submission checklist. The submission checklist indicates that the authors' provided p values and effect sizes for all hypothesis tests. They attempted this by providing a table. First, the effect size name is missing from the table header. Reporting is incomplete and there is nothing to indicate or link to the table with the text reporting which is still the old way (e.g. $p < 0.05$) (see <https://www.vox.com/science-and-health/2017/7/31/16021654/p-values-statistical-significance-redefine-0005> and <http://verso.mat.uam.es/~pablo.fernandez/Fisherb.pdf>, about why this is discouraged). Thus, it is difficult to quickly know whether it is reported. For example, Supplemental Figure 8a panel cites two one-way ANOVAs but zero one-way ANOVAs are reported in table S4 for figure S8 (also there is no panel a in the table). Additionally, it is not clear what divisor is used for Bonferroni correction, sometimes there is "n" and also "groups". Is "n" the number of experiments (e.g. number of times the test was used) or is "n" the number of units? Units are expected to number in the hundreds of thousands! We struggled to sort out what Bonferroni divisor was used. The usual practice is to cram all that information from the table into the parenthetical phrase (e.g. "rrbc=0.773 and $p=0.003$ for ranksum test, Bonferroni significance with $n=5$ is $p < 0.01$ " per the examples at the bottom of https://my.ilstu.edu/~mshesso/apa_stats.htm). A novel solutions that is more readable than the cluttered standard reporting method is to label each row in the table and referencing the table and row in your parenthetical phrase, (e.g. " $p < 0.05$ Bonferroni adjustment $n=5$, see Table S4.R82" or simply " $p < 0.05/5$, see Table S4.R82"). Editors or other reviewers may not like that, in which case it should follow examples from the link.

Please scrutinize your tables for further errors as it appears not to satisfy the submission checklist. Clearly state what the Bonferroni adjustment is (e.g. 0.05/10) for every p-value reported. Either reference specific rows of the table within each parenthetical phrase or compactly report Bonferroni adjustment N values, as well as p values and effect sizes to 2-3 significant figures in each parenthetical phrase (which was the intent of our original request).

Thank you for this comment. In the revised manuscript, we added all missing information pertaining to the statistical tests in the figures. Following the submission checklist, we also indicated the effect size for all statistical tests in both the main text and the supplementary tables. Furthermore, we showed the “n” values for all Bonferroni adjustments and now indicate what “n” means in each statistical test.

Revised text (Results, Lines 121-126)

We also found that the observed face-selective units in the untrained networks (Conv5) show a value of the averaged face-selectivity index (FSI)^{5,7} comparable to the index associated with monkey IT neurons⁷ (**Fig. 1f**, $n_{\text{untrained}} = 465$, $n_{\text{monkey}} = 158$, NS, two-sided rank-sum test, $P = 7.69 \times 10^{-2}$, $r_{\text{fbc}} = 9.25 \times 10^{-2}$, two-sided Kolmogorov–Smirnov test, $P = 2.49 \times 10^{-4}$, $d = 2.32 \times 10^{-2}$) and a significantly higher value than those measured from a shuffled response (**Fig. 1f**, $n_{\text{untrained}} = 465$, $n_{\text{shuffled}} = 465$, two-sided rank-sum test, $P = 1.49 \times 10^{-25}$, $r_{\text{fbc}} = 5.09 \times 10^{-1}$) for various definitions of the FSI^{5,17,60} (**Supplementary Fig. 3**).

Revised text (Methods, Line 453-455)

Among the face-selective units found, a face viewpoint-invariant unit was defined as a unit for which the response was not significantly different (one-way ANOVA, $P > 0.05$, Bonferroni adjustment $n=5$) among all viewpoint classes.

Revised text (Methods, Lines 514-519)

A rank-sum test was used for all analyses, except for the number of face units across convolutional groups (Kolmogorov-Smirnov test; **Supplementary Fig. S2h**), the face detection task (Kolmogorov-Smirnov test; **Fig. 3d**, **Supplementary Fig. S11b**, **Supplementary Fig. S12b**), the detection of viewpoint-invariant and -specific units (one-way ANOVA with Bonferroni adjustment, **Supplementary Fig. 8a**) and a connectivity analysis (one-way ANOVA with Bonferroni adjustment, **Supplementary Fig. 8c, d**). The divisor of all Bonferroni adjustment was five viewpoint groups ($n = 5$).

Revised text (Supplementary Table 3)

Figure S8					
a	Responses of viewpoint-specific units (0°) to each viewpoint (number of groups = 5, $n = 10$; number of units = 26)	One-way ANOVA	< 0.05	$f^2 \geq 3.82 \times 10^{-2}$	$\geq 4.63 \times 10^{-1}$
		Bonferroni adjustment	< 0.05	$f^2 \geq 5.42 \times 10^{-2}$	$\geq 4.81 \times 10^{-1}$
b	Responses of viewpoint-invariant units to each viewpoint (number of groups = 5, $n = 10$; number of units = 241)	One-way ANOVA	> 0.05	$f^2 \leq 3.62 \times 10^{-2}$	$\leq 4.41 \times 10^{-1}$
		Two-sided Kolmogorov–Smirnov test	$\geq 3.10 \times 10^{-2}$	$d \leq 2.69 \times 10^{-1}$	-
		Bonferroni adjustment	> 0.05	$f^2 \leq 3.13 \times 10^{-2}$	$\leq 4.34 \times 10^{-1}$

Q12. As previously requested, the authors must explain why some p values have asterisks to their left (e.g. $*p = 1.5 \times 10^{-25}$ on line 126 but $**p < 2.7 \times 10^{-2}$ on line 183) and be consistent about equality vs inequality symbols. Please explain all of the non-standard notations and asterisking.

Thank you for pointing this out. In the revised manuscript, we thoroughly revised the non-standard notations of the p value and the asterisks (e.g., exact P-values are shown without asterisks) throughout the manuscript.

Q13. When the authors define a “face-selective unit” they cite a paper that does permutation testing, yet the authors here do not. They use a ranksum test with no correction for multiple comparisons. We believe we can rationalize this choice (see AUC, https://en.wikipedia.org/wiki/Receiver_operating_characteristic#Area_under_the_curve), but the authors do not disclose how and why they deviate from the cited work, nor justify their thresholding in the definition of a face-selective unit. Please clarify these points. It would also be helpful to see the mean/median FSI per convolutional layer, as a way to characterize the emergence of selectivity (but this is a minor point).

We appreciate the constructive comments. Please find that we actually used identical criteria to define face-selective units and the corresponding face-selectivity index (FSI) relative to the previous work cited (Grossman, 2019).

In Grossman (2019), they defined the face-selective region as the location at which the neuronal responses are significantly greater for face stimuli than for other objects. This was done using a rank-sum test ($P < 0.05$) without permutation and/or multiple-comparison processes (Method, Section: Definition criteria of face-selective contacts, Lines 12-16 in Grossman, 2019). The permutation test was applied to validate the FSI of the face-selective region but not to define the face-selective region (Method, Section: Exemplar selectivity index of individual face contacts, Lines 13-16 in Grossman, 2019).

Thus, we also defined the face-selective unit using a rank-sum test without a permutation test or a multiple-comparison process, following the cited paper. In addition, we performed a permutation test to validate the FSI of the face-selective units. We compared the FSI of the face-selective units with those generated by the response with the shuffled labels, as in the cited paper. We confirmed that the FSI outcomes of the face-selective units in Conv3 to 5 show significantly higher values than those calculated from the responses with shuffled labels (**Fig. 1f** in the manuscript, rank-sum test, $n_{\text{Conv3}} = 365$, $n_{\text{Conv4}} = 444$, $n_{\text{Conv5}} = 465$, $P \leq 3.91 \times 10^{-10}$, $r_{\text{fbc}} \geq 2.89 \times 10^{-1}$).

Q14. (5.b in previous revision) The Kolmogorov-Smirnov tests are not reported in the main text as claimed. Lines 220-225 do not match the response to the reviewers. Furthermore, we cannot find the values in the supplementary tables. Please use the KS test in addition to either ANOVA, the signed-rank, or the rank sum test any time the manuscript claims that two populations are not distinct, or justify that only the difference in medians matters not the consistency/reliability. Also match all statistics reported in the text to a table or just complete the reporting in the parenthetical phrases. To clarify our request, let us elaborate about why we asked for two tests every time there is a claim that two populations are the same. A rank-sum or signed-rank test can only detect differences in medians. Two random variables, one that is unimodal, and one that is bi-modal may appear the same to a signed-rank or rank-sum test, whereas a KS test would be more likely to tell them apart. Conversely if two datasets are identical except the second has had an infinitesimal value added to it, then the KS test would fail to distinguish them, but a signed-rank test would be able to distinguish them. This is why, one needs two tests any time they claim two populations are not distinct. In summary: the rank-sum or signed-rank tests capture differences in median performance, the KS test captures differences in consistency, it is possible to have the same median performance but much less consistency.

Thank you for finding these mistakes remaining in the previous manuscript. We have double-checked the text to match all statistics reported in the text to the correct table and to add any missing information throughout the manuscript. We also added the results of the Kolmogorov-Smirnov tests to all statistical assessments that show that two populations are not distinct.

Revised text (Results, Lines 121-126)

We also found that the observed face-selective units in the untrained networks (Conv5) show a comparable value of averaged face-selectivity index (FSI)^{5,7} to the index associated with the monkey IT neurons⁷ (**Fig. 1f**, $n_{\text{untrained}} = 465$, $n_{\text{monkey}} = 158$, NS, two-sided rank-sum test, $P = 7.69 \times 10^{-2}$, $r_{\text{bc}} = 9.25 \times 10^{-2}$, two-sided Kolmogorov–Smirnov test, $P = 2.49 \times 10^{-4}$, $d = 2.32 \times 10^{-2}$) and a significantly higher value than those measured from a shuffled response (**Fig. 1f**, $n_{\text{untrained}} = 465$, $n_{\text{shuffled}} = 465$, two-sided rank-sum test, $P = 1.49 \times 10^{-25}$, $r_{\text{bc}} = 5.09 \times 10^{-1}$) for various definitions of the FSI^{5,17,60} (**Supplementary Fig. 3**).

Q15. (5.c. in previous revision) It is good that the authors switched to a rank-sum test and it also shows a strong difference, but why are the p values and effect sizes reported differently for figure 3 panel c in the supplemental table. What are the inequality signs signifying? Additionally, we will caution that the reason a KS test is inappropriate is not that it is for continuous distributions. Classification performance should be continuous anyway. The reason KS tests are inappropriate to judge difference in *performance* is that KS is sensitive to factors other than median performance. For example, if SVMs trained with face selective units had much higher variability, but the median value was indistinguishable, the KS test would tell one they were different but a rank-sum test would tell one they are not. Since the median, and not the variance is the most important factor then the KS test is not appropriate.

We found that the P-values in Figure 3c in the main text ($P < 1.4 \times 10^{-33}$) and in the supplementary table ($P < 1.4403 \times 10^{-33}$) appeared differently because we rounded the P-value to two significant figures in the text. In the revised manuscript, we corrected all of these numbers so that the p values and effect sizes from the same test always appear identically throughout the text and the tables.

Revised text (Results, Lines 228-232)

We confirmed that the SVM trained with multiple face-selective units shows noticeably better performance than that trained with the same number of non-selective units, as the number of units used in each condition was varied from $n = 1$ to 465 (total number of face units in untrained networks) (**Fig. 3c**, Face vs. Non-selective units, $n_{\text{trial}} = 100$, two-sided rank-sum test, $P \leq 1.45 \times 10^{-33}$, $r_{\text{rbc}} \geq 8.74 \times 10^{-1}$).

Revised text (Results, Lines 272-274)

Next, we found that the number of face-selective units observed was greater in the network trained with face-including image sets compared to that trained to face-reduced images (**Fig. 4c**, Untrained vs. Trained, $n_{\text{Net}} = 10$, two-sided rank-sum test, $P \leq 1.40 \times 10^{-3}$, $r_{\text{rbc}} \geq 5.72 \times 10^{-1}$).

Q16. (5.d. in previous revision) Standard practice is to introduce these effect sizes early in the manuscript and give them symbols e.g. r_{rbc} for rank-biserial correlation, and f^2 for Cohen's f^2 , etc. Then one reports them in the parenthetical phrase where each p-value listed. We are aware that the style of “ $p > 0.05$ ” used to be widespread and not everyone has updated their practices but there have been a number of big problems and failures after re-analysis that justify the push of topflight journals to have authors abide by those practices (see https://my.ilstu.edu/~mshesso/apa_stats.htm). Unfortunately, it falls on reviewers to actually check that authors have followed the check list. Given that we've found a number of inconsistencies it appears that more work is needed to bring the necessary attention to details to the statistical reporting. The authors have done good work on the rest of the manuscript and we only need explanations, or small reanalysis for the other parts. Now we ask that particularly close attention be paid to the statistical reporting and choice of tests.

We appreciate this detailed comment. As suggested, we reported the values of the effect size using symbols for all statistical tests in the main text. We also demonstrated exact P values throughout the revised manuscript.

Revised text (Results, Lines 121-126)

We also found that the observed face-selective units in the untrained networks (Conv5) show a value of the averaged face-selectivity index (FSI)^{5,7} comparable to the index associated with the monkey IT neurons⁷ (**Fig. 1f**, $n_{\text{untrained}} = 465$, $n_{\text{monkey}} = 158$, NS, two-sided rank-sum test, $P = 7.69 \times 10^{-2}$, $r_{\text{rbc}} = 9.25 \times 10^{-2}$, two-sided Kolmogorov–Smirnov test, $P = 2.49 \times 10^{-4}$, $d = 2.32 \times 10^{-2}$) and a significantly higher value than those measured from a shuffled response (**Fig. 1f**, $n_{\text{untrained}} = 465$, $n_{\text{shuffled}} = 465$, two-sided rank-sum test, $P = 1.49 \times 10^{-25}$, $r_{\text{rbc}} = 5.09 \times 10^{-1}$) for various definitions of the FSI^{5,17,60} (**Supplementary Fig. 3**).

****Major Concern 6: Missing important related works**

Q17. (6.a. in previous revision) This is satisfactory, but there are some points to clarify. The readout of a reservoir network can be arbitrary or chosen in advance based on knowledge of the system. So, selecting “activation maximization” as the readout mechanism in advance is consistent with the reservoir computing concept. More fundamentally, the authors train an SVM coupled with a feature engineering mechanism (SVM on algorithmically identified face units). This SVM work was what connects it to reservoir computing most strongly, maybe other readers will notice that.

Thank you for the helpful comments. In the discussion of the revised manuscript, we added sentences that add to the discussion of the possibility that our current model can be consistent with the concept of reservoir computing.

Revised text (Discussion, Lines 356-358)

Similarly, the theory of “reservoir computing” suggests that the circuits required for higher order cognitive functions, such as image classification, may already exist in untrained, random recurrent neural networks. In this scenario, higher order cognitive functions can be achieved only by training a read-out network, as suggested by the lottery ticket hypothesis^{82,83}. Interestingly, our results in the face-detection task performed with the SVM are comparable to the concept of reservoir computing, as the training of the SVM with the responses of untrained networks is consistent with the procedure of training a read-out projection from random networks in reservoir computing. It is notable that recent studies suggest that the random network can perform this task if the read-out units are selected via a prior understanding of the system. For example, object classification can be performed by a random network if read-outs are chosen by a synaptic rule observed in the brain (Weidel 2021, Tetzlaff 2013). While these models focus on the innate functions of networks in that a high dimensional space generated by a random “network” can perform various tasks without learning, our current results demonstrate that the functional tuning of “single units” (comparable to neuronal tuning in biological brains) can arise in random networks without any further training of the read-out process, which is distinguished from the main idea of the reservoir computing model.

**Minor Concerns

Q18. (Minor. c in previous revision) Now it is line 277 but “face-deprive” is still missing that d.

We corrected the typo in the revised manuscript.

Q19. (Minor. d in previous revision) Some apparent issues with the “n.s.” asterisking. It is not the same notation from figure to figure and as pointed out above, some figures are indistinguishable but have different n.s. asterisks. Please check your figures reference the correct data and that asterisking is consistent. For example, figure S5 has single and double asterisks, but these are not explained.

We regret this confusing notation in the previous manuscript and we have corrected all errors and inconsistencies in the statistical notations throughout the revised manuscript; we removed asterisks and reported exact P-values in all figures and text.

Q20. (Minor. f in previous revision) These corrections were not carried out correctly. The sentence is still ungrammatical, with a noun where an adjective should go and a definite article where an adverb should go. Line 13 should have been changed like so: “across different conditions of random initialization of the network” -> “across different random initializations of the network”. Instead it was changed like so: “across different conditions of random initialization of the network” -> “in random initialized networks and these units reproduce the characteristics observed in monkeys.”. Now the simplest change is like so: “in random initialized networks and these units reproduce the characteristics observed in monkeys.” -> “in randomly initialized networks and these units reproduce many characteristics observed in monkeys.”

The sentence on line 13 was revised as suggested.

Revised text (Abstract, Lines 12-13)

We found that units selective to faces emerge robustly in randomly initialized networks and that these units reproduce many characteristics observed in monkeys.

Q21. (Minor. g in previous revision) the word “translation” is misspelled throughout the manuscript as “translatation”.

The typo was fixed.

References

Dosovitskiy, A., and Brox, T. (2016). Generating Images with Perceptual Similarity Metrics based on Deep Networks. *Adv. Neural Inf. Process. Syst.* 658–666.

Grossman, S., Gaziv, G., Yeagle, E.M., Harel, M., Mégevand, P., Groppe, D.M., Khuvis, S., Herrero, J.L., Irani, M., Mehta, A.D., et al. (2019). Convergent evolution of face spaces across human face-selective neuronal groups and deep convolutional networks. *Nat. Commun.* 10, 4934.

Handl, J., Knowles, J., and Kell, D.B. (2005). Computational cluster validation in post-genomic data analysis. *Bioinformatics* 21, 3201–3212.

Kaufman, L., and Rousseeuw, P.J. (2009). *Finding groups in data: an introduction to cluster analysis* (John Wiley & Sons).

Weidel, Philipp, Renato Duarte, and Abigail Morrison. "Unsupervised learning and clustered connectivity enhance reinforcement learning in spiking neural networks." *Frontiers in computational neuroscience* 15 (2021): 18.

Tetzlaff, C., Kolodziejcki, C., Timme, M., Tsodyks, M., and Wörgötter, F. (2013). Synaptic scaling enables dynamically distinct short- and long-term memory formation. *PLoS Comput. Biol.* 9:e1003307.

Reviewers' Comments:

Reviewer #3:

Remarks to the Author:

We appreciate the hard work on this, and commend the authors for a fascinating body of work. The manuscript is much stronger now, and concerns are largely assuaged. Just a couple of minor notes:

-Line 38: "Moreover, visual category-selective domains, including the face, are observed in the ventral visual stream in adult humans who have been blind since birth."

By definition, "visual" domains cannot occur in blind humans. The more accurate statement would read:

"Moreover, category-selective domains, including those for faces, are observed in the ventral stream of adult humans who have been blind since birth."

-Legend for figure 2 uses the word 'generic' instead of 'genetic'

Reviewer #5:

None

1 **Response to Reviewers**

2 Thank you very much for the effort you have devoted to reviewing our manuscript. We revised our manuscript with
3 regard to the remaining issues. Please find our responses to each comment below.

4

5 **Reviewer #3**

6 We appreciate the hard work on this, and commend the authors for a fascinating body of work. The manuscript is much stronger
7 now, and concerns are largely assuaged. Just a couple of minor notes:

8 Q1. Line 38: "Moreover, visual category-selective domains, including the face, are observed in the ventral visual stream in
9 adult humans who have been blind since birth." By definition, "visual" domains cannot occur in blind humans. The more
10 accurate statement would read: "Moreover, category-selective domains, including those for faces, are observed in the ventral
11 stream of adult humans who have been blind since birth."

12 The sentence (line 38) was revised as suggested.

13 **Revised text (Abstract, Lines 37-39)**

14 Moreover, category-selective domains, including those for faces, are observed in the ventral stream of adult humans
15 who have been blind since birth

16 Q2. Legend for figure 2 uses the word 'generic' instead of 'genetic'

17 We corrected the typo in the revised manuscript.